# Theory on Score-Mismatched Diffusion Models and Zero-Shot Conditional Samplers

**Yuchen Liang**[†], **Peizhong Ju**[‡], **Yingbin Liang**[†], **Ness Shroff**[†]
[†]The Ohio State University       [‡]University of Kentucky

## Abstract

The denoising diffusion model has recently emerged as a powerful generative technique, capable of transforming noise into meaningful data. While theoretical convergence guarantees for diffusion models are well established when the target distribution aligns with the training distribution, practical scenarios often present mismatches. One common case is in the zero-shot conditional diffusion sampling, where the target conditional distribution is different from the (unconditional) training distribution. These score-mismatched diffusion models remain largely unexplored from a theoretical perspective. In this paper, we present the first performance guarantee with explicit dimensional dependencies for general score-mismatched diffusion samplers, focusing on target distributions with finite second moments. We show that score mismatches result in an asymptotic distributional bias between the target and sampling distributions, proportional to the accumulated mismatch between the target and training distributions. This result can be directly applied to zero-shot conditional samplers for any conditional model, irrespective of measurement noise. Interestingly, the derived convergence upper bound offers useful guidance for designing a novel bias-optimal zero-shot sampler in linear conditional models that minimizes the asymptotic bias. For such bias-optimal samplers, we further establish convergence guarantees with explicit dependencies on dimension and conditioning, applied to several interesting target distributions, including those with bounded support and Gaussian mixtures. Our findings are supported by numerical studies.

## 1 Introduction

Generative modeling stands as a cornerstone in deep learning, with the goal of producing samples whose distribution emulates that of the training data. Traditional approaches encompass variational autoencoders (VAE) (Kingma & Welling, 2022), generative adversarial networks (GANs) (Goodfellow et al., 2014), normalizing flows (Rezende & Mohamed, 2015), and others. Recently, diffusion models, especially the denoising diffusion probabilistic models (DDPMs) (Sohl-Dickstein et al., 2015; Ho et al., 2020), have emerged as particularly compelling generative models, gaining widespread acclaim for their stable and cutting-edge performance across various tasks, such as image and video generation (Ramesh et al., 2022; Rombach et al., 2022).

In ideal situations, the training and target distributions of generative models match each other. However, this often does not hold in practice, where distributional mismatch between the training and target distributions can occur due to various reasons such as possible privacy constraints, need for computational efficiency, and knowledge gap between training and sampling processes. Specifically for diffusion models, such mismatches exhibit between the scores obtained from the training data and the scores of the target distribution from which we want to generate samples. One common scenario that existing studies primarily focus on is **conditional** diffusion models in image generation tasks (see Croitoru et al. (2023); Li et al. (2023); Moser et al. (2024) for surveys of diffusion models in computer vision). Different from unconditional image generation, conditional image samplers aim to generate images that are consistent with the given information, either be a text-prompt (as in text-to-image synthesis) or a sub-image (as in image super-resolution). For example, in image super-resolution, given the input of a low-resolution image, the goal is not to generate some arbitrary high-resolution image but the one whose corresponding low-resolution part matches the given input. Here the diffusion models are well-trained on the *unconditional* distribution of high-resolution images, whereas the target distribution is the *conditional* distribution given the low-resolution input. If one

uses these well-trained unconditional scores to generate conditional samples, there will be a mismatch at each step of the sampling process.

One class of methods to tackle the conditional sampling problem is to include *extra-guided training*, where a modified score function is trained with the *extra* knowledge of the conditioning information (Dhariwal & Nichol, 2021; Ho & Salimans, 2022). On the theory side, several recent works (Yuan et al., 2023; Wu et al., 2024; Fu et al., 2024) provided performance guarantees for such conditional diffusion samplers, where a score guidance is obtained through *extra* training based on the conditional information. However, the additional guided training in these samplers requires *extra* computations and needs to be conducted for every image conditioning, which may not be efficient in practice.

Alternatively, *zero-shot* conditional image samplers arise as a prevalent approach (e.g., Choi et al. (2021); Chung et al. (2022b;a; 2023); Wang et al. (2023); Song et al. (2023a); Fei et al. (2023)) for *training-free* conditional generation given well-trained unconditional scores. For each conditioned image, zero-shot samplers require no additional training to modify the scores. Instead, they adjust the scores during sampling by calculating rectified scores based on conditional information to mitigate the mismatch between the oracle conditional scores and the approximated ones.[1] Despite their empirical promise, theoretical guarantee on these zero-shot samplers is largely unexplored. In Gupta et al. (2024), the authors provided a super-polynomial lower bound for zero-shot sampling as a converse result. In Xu & Chi (2024), the authors proposed and analyzed a *plug-and-play* conditional sampler. Their analysis relies on the properties of the Markov transition kernel specific to their plug-and-play model, which does not appear to be applicable to several widely used zero-shot samplers, such as Come-Closer-Diffuse-Faster (CCDF) (Chung et al., 2022b) and Denoising Diffusion Null-space Model (DDNM) (Wang et al., 2023). Therefore, there is a need to provide the performance guarantee for those popular zero-shot conditional samplers.

In this paper, we address two key theoretical research gaps in zero-shot score-mismatched diffusion models: (i) We provide performance guarantees for general score-mismatched diffusion models, extending their applicability beyond the primary focus of existing theoretical studies on the special case of conditional image generation. (ii) We analyze zero-shot conditional diffusion models, which are generally applicable to existing zero-shot samplers such as CCDF (Chung et al., 2022b) and DDNM (Wang et al., 2023) to which the analysis in Xu & Chi (2024) is not applicable (as discussed above).

## 1.1 OUR CONTRIBUTIONS

Technically, the main challenge due to mismatched scores is to analyze the expected tilting factor (Liang et al., 2024) under a mean-perturbed Gaussian, providing an upper bound of the asymptotic orders of all Gaussian non-centralized moments. Our detailed contributions are as follows.

**Convergence of General Score-Mismatched DDPM**: We provide the *first* non-asymptotic convergence bound on the KL divergence between the target and generated distributions when there is mismatch between the sampling and target scores in DDPM samplers, for general target distributions having finite second moments. We show that the score mismatch at each diffusion step introduces an asymptotic distributional bias that is proportional to the accumulated mismatch. We also provide the first explicit dimensional dependency when the sixth moment of the target distribution exists. Our result is applicable to general forms of mismatch between the target and training scores, which greatly extend the focus of the existing theoretical research on conditional score-mismatch diffusion models.

We then apply our results to zero-shot conditional DDPM samplers, as long as the conditioning involves certain deterministic or random transformations of the data. This provides the first theoretical guarantees for several existing zero-shot samplers, such as CCDF (Chung et al., 2022b) and DDNM (Wang et al., 2023). Notably, the theory in Xu & Chi (2024) does not apply to these samplers, as their analysis relies on the properties of the Markov transition kernel specific to their plug-and-play model. In contrast, our approach is based on the tilting-factor analysis from Liang et al. (2024), which is applicable to general score-mismatched DDPM models. Moreover, the theory in Xu & Chi (2024) is limited to cases where the measurement log-likelihood function is differentiable and bounded and does not provide explicit dependencies on the data dimension. In contrast, our results do not require

---

[1]Note, however, that some zero-shot methods, such as DPS (Chung et al., 2023) and ΠGDM (Song et al., 2023a), might induce additional computational costs during sampling.

the measurement log-likelihood function to be differentiable or bounded and explicitly characterize the dependencies on the data dimension.

**Novel Design of Bias-Optimal Zero-shot Sampler BO-DDNM**: Inspired by our convergence analysis of score-mismatched DDPM, we design a novel zero-shot conditional sampler, called the BO-DDNM sampler, which minimizes the asymptotic bias for linear conditional models. Such a sampler coincides with the regular DDNM sampler (Wang et al., 2023) when there is no presence of measurement noise, and achieves faster convergence than both the DDNM and DDNM$^+$ samplers under measurement noise, as shown by our theory and numerical simulations.

**Theory for BO-DDNM with Explicit Parameter Dependencies**: We provide the convergence bound for the proposed BO-DDNM sampler with explicit dependencies on the dimension $d$ as well as the conditional information $y$, for various interesting classes of target distributions including those having bounded support and Gaussian mixture. For the case of Gaussian mixture, we further show that three factors positively affect the asymptotic bias: (1) the variance of the measurement noise, (2) the averaged distance between $y$ and the mean of each Gaussian component, and (3) the corresponding correlation coefficient for each component.

## 1.2 RELATED WORK

We provide a summary of works addressing *unconditional* and *score-matched* diffusion models in Appendix B. Below we discuss related works on *conditional* diffusion models which are closely related to our study here.

**Extra-Guided Training:** In order to achieve conditional sampling using DDPM models in practice, one method is to introduce conditional guided training, where one either uses an existing classifier (a.k.a., classifier guidance) (Dhariwal & Nichol, 2021) or jointly trains the unconditional and conditional scores (a.k.a., classifier-free guidance (CFG)) (Ho & Salimans, 2022). Here a guidance term is obtained to "guide" the diffusion sampling process at each step such that the sampling scores correspond to the true conditional scores. On the theory side, Wu et al. (2024) investigates the effect of the guidance strength in CFG on Gaussian mixtures, Bradley & Nakkiran (2024) shows that CFG is an instance of predictor-corrector methods, and Chidambaram et al. (2024) finds that CFG might fail to sample correctly on certain mixture targets. There are other theoretical works that investigate sample complexity bounds for conditional score matching for a variety of target distribution models, including the conditional Ising models (Mei & Wu, 2023), those supported on a low-dimensional linear subspace (Yuan et al., 2023), and Hölder smooth and sub-Gaussian conditional models (Fu et al., 2024). Other than stochastic samplers, a conditional ODE sampler is proposed and studied in Chang et al. (2024), which also requires extra training of the conditional score function.

**Zero-shot Samplers:** To achieve conditional DDPM sampling, a popular method is to use zero-shot conditional samplers, with which one generates a conditional sample using approximated scores. These scores are calculated from the unconditional score estimates and the conditional information using simple (usually linear) functions *without extra-training* (Choi et al., 2021; Chung et al., 2022a;b; 2023; Wang et al., 2023; Song et al., 2023a; Fei et al., 2023). The only theoretical works on the performance of zero-shot DDPM conditional samplers are Xu & Chi (2024); Gupta et al. (2024). In Xu & Chi (2024), a diffusion plug-and-play sampler is proposed which alternates between a diffusion sampling step and a consistency sampling step. The difference of our results from those in Xu & Chi (2024) has been thoroughly discussed in Section 1.1. From an alternative perspective, Gupta et al. (2024) shows that the sampling complexity with zero-shot samplers can take super-polynomial time for some worst-case distribution (among the set of distributions where smooth scores can be efficiently estimated). Instead, our result shows a consistent fact that there exists a non-vanishing asymptotic distributional bias within polynomial time.

## 2 PROBLEM SETUP

In this section, we first provide some background on the score-*matched* DDPMs. Then, we introduce the score-*mismatched* DDPM samplers and, as a special example, the *conditional* sampling problem and zero-shot samplers.

### 2.1 BACKGROUND OF SCORE-MATCHED DDPMS

The goal of the score-matched sampling problem is to generate a sample whose distribution is close to the data distribution. To this end, the DDPM algorithm (Ho et al., 2020) is widely used, which

consists of a forward process and a reverse process of latent variables. Let $x_0 \in \mathbb{R}^d$ be the initial data, and let $x_t \in \mathbb{R}^d, \forall 1 \leq t \leq T$ be the latent variables. Let $Q_0$ be the data distribution, and let $Q_t$ (resp., $Q_{t,t-1}$) be the marginal (resp., joint) latent distribution for all $1 \leq t \leq T$.

**Forward Process:** In the forward process, white Gaussian noise is gradually added to the data: $x_t = \sqrt{1-\beta_t}x_{t-1} + \sqrt{\beta_t}w_t, \forall 1 \leq t \leq T$, where $w_t \stackrel{i.i.d.}{\sim} \mathcal{N}(0, I_d)$. Equivalently, this can be expressed as:

$$Q_{t|t-1}(x_t|x_{t-1}) = \mathcal{N}(x_t; \sqrt{1-\beta_t}x_{t-1}, \beta_t I_d), \tag{1}$$

which means that under $Q$, the Markov chain $X_0 \to X_1 \to \cdots \to X_T$ holds. Define $\alpha_t := 1 - \beta_t$ and $\bar{\alpha}_t := \prod_{i=1}^{t} \alpha_i$ for all $1 \leq t \leq T$. An immediate result by accumulating the steps is that $Q_{t|0}(x_t|x_0) = \mathcal{N}(x_t; \sqrt{\bar{\alpha}_t}x_0, (1-\bar{\alpha}_t)I_d)$, or, written equivalently, $x_t = \sqrt{\bar{\alpha}_t}x_0 + \sqrt{1-\bar{\alpha}_t}\bar{w}_t, \forall 1 \leq t \leq T$, where $\bar{w}_t \sim \mathcal{N}(0, I_d)$ denotes the *aggregated* noise at time $t$ and is independent of $x_0$. Finally, since each $\bar{w}_t$ is Gaussian, each $Q_t$ ($t \geq 1$) is absolutely continuous w.r.t. the Lebesgue measure. Let the p.d.f. of each $Q_t$ be $q_t$, and $q_{t,t-1}$, $q_{t|t-1}$, and $q_{t-1|t}$ for $t \geq 1$ are similarly defined.

**Reverse Process:** In the reverse process, the latent variable at time $T$ is first drawn from a standard Gaussian distribution: $x_T \sim \mathcal{N}(0, I_d) =: P_T$. Then, each forward step is approximated by a reverse sampling step. At each time $t = T, T-1, \dots, 1$, define the *true* reverse process as $x_{t-1} = \mu_t(x_t) + \sigma_t z_t$, where $z \sim \mathcal{N}(0, I_d)$. Here $\sigma_t^2 := \frac{1-\alpha_t}{\alpha_t}$. For the typical DDPM sampling process, $\mu_t(x_t) = \frac{1}{\sqrt{\alpha_t}}(x_t + (1-\alpha_t)\nabla \log q_t(x_t))$. Equivalently, $P_{t-1|t} = \mathcal{N}(x_{t-1}; \mu_t(x_t), \sigma_t^2 I_d)$. Here $\nabla \log q_t(x)$ is called the *score* of $q_t$, and $\mu_t(x_t)$ is a function of the score. Let $P_t$ be the marginal distributions of $x_t$ in the true reverse process, and let $p_t$ be its corresponding p.d.f. w.r.t. the Lebesgue measure. Define $p_{t-1|t}$ and $p_{t|t-1}$ in a way similar to the forward process.

In practice, one does not have access to $\nabla \log q_t(x_t)$ and thus $\mu_t(x_t)$. Instead, an estimate of $\nabla \log q_t(x_t)$, denoted as $s_t(x_t)$, is used, which results in an estimated $\hat{\mu}_t(x_t)$ and the *estimated* reverse process: $x_{t-1} = \hat{\mu}_t(x_t) + \sigma_t z$. Let $\hat{P}_t$ be the marginal distributions of $x_t$ in the estimated reverse process with the corresponding p.d.f. $\hat{p}_t$. Note that $\hat{P}_{t-1|t} = \mathcal{N}(x_{t-1}; \hat{\mu}_t(x_t), \sigma_t^2 I_d)$ and $\hat{P}_T = P_T$. Hence, under $P$ and $\hat{P}$, $X_T \to X_{T-1} \to \cdots \to X_0$ holds.

**Performance Metrics:** In the case where $Q_0$ is absolutely continuous w.r.t. the Lebesgue measure, we are interested in measuring the sampling performance through the KL divergence between $Q_0$ and $\hat{P}_0$, defined as

$$\mathrm{KL}(Q\|P) := \int \log \frac{\mathrm{d}Q}{\mathrm{d}P}\mathrm{d}Q = \mathbb{E}_{X \sim Q}\left[\log \frac{q(X)}{p(X)}\right] \geq 0.$$

Indeed, from Pinsker's inequality, the total-variation (TV) distance can be upper bounded as $\mathrm{TV}(Q_0, \hat{P}_0)^2 \leq \frac{1}{2}\mathrm{KL}(Q_0\|\hat{P}_0)$. When $q_0$ does not exist, we use the Wasserstein-2 distance to measure the one-step perturbed performance, which is defined as

$$W_2(Q, P) := \left\{\min_{\Gamma \in \Pi(Q,P)} \int_{\mathbb{R}^d \times \mathbb{R}^d} \|x - y\|^2 \, \mathrm{d}\Gamma(x, y)\right\}^{1/2},$$

where $\Pi(Q, P)$ is the set of all joint probability measures on $\mathbb{R}^d \times \mathbb{R}^d$ with marginal distributions $Q$ and $P$, respectively. Both metrics are widely adopted (Chen et al., 2023a; Benton et al., 2024a).

## 2.2 Score-Mismatched DDPMs

Differently from the score-*matched* sampling problem, the goal of the score-*mismatched* problem is to sample from a **different** target distribution from the training distribution with which we estimate the scores. Thus, there will be a mismatch between the target score and the estimated score at each diffusion step. Let $Q_t$ ($t \geq 0$) be the *training* distributions used for training the score. Let $\tilde{Q}_0$ be the *target* distribution that one hopes to generate samples from, and let $\tilde{Q}_t$ ($t \geq 1$) be its Gaussian-perturbed distributions according to the forward process in (1). Define the posterior mean under the target distributions as $m_t(\tilde{x}_t) := \mathbb{E}_{\tilde{X}_{t-1} \sim \tilde{Q}_{t-1|t}}[\tilde{X}_{t-1}|\tilde{x}_t]$. Note that by Tweedie's formula (Efron, 2011), $m_t(\tilde{x}_t) = \frac{1}{\sqrt{\alpha_t}}(\tilde{x}_t + (1-\alpha_t)\nabla \log \tilde{q}_t(\tilde{x}_t))$. Recall that $P_t$ and $\hat{P}_t$ are the *sampling* distributions of the true and estimated reverse process, respectively. For general score-mismatched DDPMs, we leave the generic definition of $\mu_t(x_t)$ without providing any particular expression. An

example of $\mu_t(x_t)$ is given later in (6), yet our general analysis does not require any particular form for $\mu_t$. With these notations, the *score mismatch* at each step $t \geq 1$ can be defined as

$$\Delta_t(x_t) := \frac{\sqrt{\alpha_t}}{1-\alpha_t} \left( \mathbb{E}_{X_{t-1} \sim \tilde{Q}_{t-1|t}}[X_{t-1}|x_t] - \mathbb{E}_{X_{t-1} \sim P_{t-1|t}}[X_{t-1}|x_t] \right) = \frac{\sqrt{\alpha_t}}{1-\alpha_t}(m_t(x_t) - \mu_t(x_t)). \tag{2}$$

The goal, then, is to provide an upper bound on the distributional dissimilarity between the target distribution $\tilde{Q}_0$ and the sampling distribution $\widehat{P}_0$. We use the same metrics as those defined in Section 2.1 to evaluate the performance of the score-mismatched DDPM.

## 2.3 ZERO-SHOT CONDITIONAL DDPMs

One interesting example of score mismatch is the zero-shot conditional sampling problem. Differently from the unconditional counterpart, the conditional sampling problem aims to obtain a sample that aligns in particular with the provided conditioning. Define $y \in \mathbb{R}^p$ to be the conditioned information about $x_0$. Specifically, let $y = h(x_0)$, where $h(\cdot)$ is some arbitrary (deterministic or random) function of only $x_0$ (apart from independent noise). Note that general score-mismatched DDPMs can be specialized to zero-shot conditional samplers with the following notations:

$$\tilde{Q}_t = Q_{t|y}, \quad m_t = m_{t,y}, \quad \mu_t = \mu_{t,y}, \quad \text{and } \Delta_t = \Delta_{t,y}. \tag{3}$$

**Linear Conditional Models:** In practice, one commonly adopted model is the linear conditional model (Jalal et al., 2021; Wang et al., 2023; Song et al., 2023a), defined as

$$y := Hx_0 + n, \tag{4}$$

where $H \in \mathbb{R}^{p \times d}$ ($p \leq d$) is a deterministic matrix and $n \sim \mathcal{N}(0, \sigma_y^2 I_p)$ is the measurement noise, which is assumed to be Gaussian and independent of $x_0$. For the case where there is no measurement noise, let $\sigma_y^2 = 0$ and thus $n = 0$ almost surely. In applications like image super-resolution and inpainting (Wang et al., 2023), $H$ admits a simple form of a 0-1 diagonal matrix, where the 1's occur only on the diagonal and at those locations corresponding to the provided pixels. In these scenarios, both $H$ and $y$ are fixed and given. The linear conditional model is studied in Section 4.

**Conditional Forward Process for Linear Models:** Write the Moore–Penrose pseudo-inverse of $H$ as $H^\dagger$, and note that $H^\dagger H$ is an orthogonal projection matrix. With this notation, under (4), we can re-express the forward process in (1) as

$$x_t = \sqrt{\bar{\alpha}_t}(I_d - H^\dagger H)x_0 + \sqrt{\bar{\alpha}_t}H^\dagger y - \sqrt{\bar{\alpha}_t}H^\dagger n + \sqrt{1 - \bar{\alpha}_t}\bar{w}_t.$$

Here, since $n$ is independent of $\bar{w}_t$, for fixed $x_0$ and $y$, we have that, for all $t \geq 1$,

$$Q_{t|0,y}(x_t|x_0, y) = \mathcal{N}(x_t; \sqrt{\bar{\alpha}_t}(I_d - H^\dagger H)x_0 + \sqrt{\bar{\alpha}_t}H^\dagger y, \bar{\alpha}_t \sigma_y^2 H^\dagger (H^\dagger)^\intercal + (1 - \bar{\alpha}_t)I_d). \tag{5}$$

Also, since the forward process is a Markov chain, we have that $Q_{t|t-1,y} = Q_{t|t-1}$ for all $t \geq 1$.

**Zero-shot Conditional Sampler for Linear Models:** We employ the *zero-shot* conditional sampler for linear conditional models in the following form: $x_{t-1} = \mu_{t,y}(x_t) + \sigma_t z_t$, where

$$\mu_{t,y}(x_t) = \frac{1}{\sqrt{\alpha_t}}\left(x_t + (1 - \alpha_t)g_{t,y}(x_t)\right), \quad g_{t,y} := (I_d - H^\dagger H)\nabla \log q_t(x_t) + f_{t,y}(x_t). \tag{6}$$

Here $f_{t,y}(x_t)$ is a simple function of $y$ and $x_t$ computable *without extra training* and such that $(I_d - H^\dagger H)f_{t,y}(x) \equiv 0$ for all $x \in \mathbb{R}^d$. Intuitively, $f_{t,y}$ characterizes the score rectification in the range space of $H^\dagger H$. Indeed, many zero-shot samplers in the literature have such $f_{t,y}(x_t)$'s that satisfy (6) (see Appendix D). Now, with the linear model in (4) and the zero-shot conditional sampler in (6), the score mismatch at each time $t \geq 1$ is equal to

$$\Delta_{t,y}(x_t) = (I_d - H^\dagger H)(\nabla \log q_{t|y}(x_t) - \nabla \log q_t(x_t)) + (H^\dagger H)\nabla \log q_{t|y}(x_t) - f_{t,y}(x_t). \tag{7}$$

## 3 DDPM UNDER GENERAL SCORE MISMATCH

In this section, we provide convergence guarantees for general score-mismatched DDPM samplers under a general target distribution $\tilde{Q}_0$. Throughout this section we keep the generic definition for score mismatch $\Delta_t$ as in (2), without assuming any particular expression for $\mu_t$.

## 3.1 TECHNICAL ASSUMPTIONS

We will analyze general score-mismatched DDPMs under the following technical assumptions.

**Assumption 1** (Finite Second Moment). There exists a constant $M_2 < \infty$ (that does not depend on $d$ and $T$) such that $\mathbb{E}_{X_0 \sim \tilde{Q}_0} \|X_0\|^2 \leq d M_2$.

The first Assumption 1 is commonly adopted in the analyses of score-matched DDPM samplers (Chen et al., 2023a;d; Liang et al., 2024).

**Assumption 2** (Posterior Mean Estimation). The estimated posterior mean $\widehat{\mu}_t$ at $t = 1, \ldots, T$ satisfy

$$\frac{1}{T} \sum_{t=1}^{T} \frac{\alpha_t}{(1-\alpha_t)^2} \mathbb{E}_{X_t \sim \tilde{Q}_t} \|\widehat{\mu}_t(X_t) - \mu_t(X_t)\|^2 \leq \varepsilon^2, \text{ where } \varepsilon^2 = \tilde{O}(T^{-2}).$$

The above Assumption 2 is made for the score estimation error for the general mismatched setting, where we leave generic definitions of $\mu_t$ and $\widehat{\mu}_t$. While the expectation is over $\tilde{Q}_t$, Assumption 2 is very likely to hold when $\tilde{Q}_t$ is close to $Q_t$, i.e., when the score mismatches are moderate. For zero-shot conditional samplers in linear models, this assumption is weaker than that for the estimation error for unconditional scores (see (9)). Compared with the score-matched case, the estimation error needs to be achieved at a higher accuracy because of the extra error term when there is score mismatch (Lemma 2). Such a higher level of estimation accuracy also occurs in previous theoretical studies for accelerated DDPM samplers (Li et al., 2024a).

**Assumption 3** (Regular Derivatives). For all $t \geq 1$ where $\tilde{q}_{t-1}$ exists, $\ell \geq 1$, and $\boldsymbol{a} \in [d]^p$ where $|\boldsymbol{a}| = p \geq 1$,

$$\mathbb{E}_{X_t \sim \tilde{Q}_t} |\partial_{\boldsymbol{a}}^p \log \tilde{q}_t(X_t)|^\ell = O(1), \quad \mathbb{E}_{X_t \sim \tilde{Q}_t} |\partial_{\boldsymbol{a}}^p \log \tilde{q}_{t-1}(m_t(X_t))|^\ell = O(1).$$

The above Assumption 3 is useful for our tilting-factor based analysis, which guarantees that all (higher-order) Taylor polynomials of $\log \tilde{q}_t$ are well controlled in expectation. It is rather soft, and it can be verified when $\tilde{Q}_0$ has finite variance (under early-stopping) (Liang et al., 2024).

**Assumption 4** (Bounded Mismatch). For all $t \geq 1$ where $\tilde{q}_{t-1}$ exists, and $\ell \geq 2$,

$$\mathbb{E}_{X_t \sim \tilde{Q}_t} \|\Delta_t(X_t)\|^\ell = O(\bar{\alpha}_t).$$

The above Assumption 4 is used to characterize the amount of mismatch at each time $t \geq 1$. The $\bar{\alpha}_t := \prod_{i=1}^{t} \alpha_i$ is necessary for the overall bias to be bounded.

In the paper, Assumptions 3 and 4 have been established in two cases of zero-shot conditional sampling: (i) where $Q_0$ has bounded support for any $H$, using a special $\alpha_t$ in (8) (see the proof of Theorem 4); and (ii) where $Q_0$ is Gaussian mixture and $H = (I_p \quad 0)$ (see Lemma 8). For Case (i), the assumption that $Q_0$ has bounded support has wide applicability in practice (e.g., images (Ho et al., 2020; Wang et al., 2023)) and is commonly made in many theoretical investigations of the score-matched DDPM (Li et al., 2024a;c). Finally, note that when $\tilde{q}_0$ does not exist (e.g., for images (Ho et al., 2020; Wang et al., 2023)), Assumptions 3 and 4 are required only for $t \geq 2$.

## 3.2 CONVERGENCE BOUND

Before presenting the main result, we first define a set of noise schedule as follows.

**Definition 1** (Noise Schedule). For all sufficiently large $T$, set the step size $\alpha_t$'s to satisfy

$$1 - \alpha_t \lesssim \frac{\log T}{T}, \; \forall 1 \leq t \leq T, \quad \text{and} \quad \bar{\alpha}_T := \prod_{t=1}^{T} \alpha_t = o\left(\frac{1}{T}\right).$$

An example of $\alpha_t$ that satisfies Definition 1 is $1 - \alpha_t \equiv \frac{c \log T}{T}, \; \forall t \geq 1$ with $c > 1$. Then, $\bar{\alpha}_T = \left(1 - \frac{c \log T}{T}\right)^T = \exp\left(T \log\left(1 - \frac{c \log T}{T}\right)\right) = O\left(e^{T \frac{-c \log T}{T}}\right) = o\left(T^{-1}\right).$

The following Theorem 1 provides an upper bound on the KL-divergence between the target distribution $\tilde{Q}_0$ and the sampling distribution $\widehat{P}_0$, as a function of (general) score-mismatch $\Delta_t$ at each time $t \geq 1$. Theorem 1 is the *first* convergence result for score-mismatched DDPM samplers for any smooth $\tilde{Q}_0$ that has finite second moment (along with some mild regularity conditions).

**Theorem 1** (DDPM under Score Mismatch). *Suppose that $\tilde{Q}_0$ has a p.d.f. $\tilde{q}_0$ which is analytic, and suppose that Assumptions 1 to 4 are satisfied. Then, with the $\alpha_t$ chosen to satisfy Definition 1, the distribution $\widehat{P}_0$ from the score-mismatched DDPM satisfies*

$$\mathrm{KL}(\tilde{Q}_0 \| \widehat{P}_0) \lesssim \mathcal{W}_{oracle} + \mathcal{W}_{bias} + \mathcal{W}_{vanish}, \quad where$$

$$\mathcal{W}_{oracle} = \sum_{t=1}^{T} \frac{(1-\alpha_t)^2}{2\alpha_t} \mathbb{E}_{X_t \sim \tilde{Q}_t} \left[ \mathrm{Tr}\left( \nabla^2 \log \tilde{q}_{t-1}(m_t(X_t)) \nabla^2 \log \tilde{q}_t(X_t) \right) \right] + (\log T)\varepsilon^2$$

$$\mathcal{W}_{bias} = \sum_{t=1}^{T} (1-\alpha_t) \mathbb{E}_{X_t \sim \tilde{Q}_t} \|\Delta_t(X_t)\|^2$$

$$\mathcal{W}_{vanish} = \sum_{t=1}^{T} \frac{1-\alpha_t}{\sqrt{\alpha_t}} \mathbb{E}_{X_t \sim \tilde{Q}_t} \left[ (\nabla \log \tilde{q}_{t-1}(m_t(X_t)) - \sqrt{\alpha_t} \nabla \log \tilde{q}_t(X_t))^\intercal \Delta_t(X_t) \right]$$

$$- \sum_{t=1}^{T} \frac{(1-\alpha_t)^2}{2\alpha_t} \mathbb{E}_{X_t \sim \tilde{Q}_t} \left[ \Delta_t(X_t)^\intercal \nabla^2 \log \tilde{q}_{t-1}(m_t(X_t)) \Delta_t(X_t) \right]$$

$$+ \sum_{t=1}^{T} \frac{(1-\alpha_t)^2}{3! \alpha_t^{3/2}} \mathbb{E}_{X_t \sim \tilde{Q}_t} \left[ 3 \sum_{i=1}^{d} \partial_{iii}^3 \log \tilde{q}_{t-1}(m_t(X_t)) \Delta_t(X_t)^i \right.$$

$$\left. + \sum_{\substack{i,j=1 \\ i \neq j}}^{d} \partial_{iij}^3 \log \tilde{q}_{t-1}(m_t(X_t)) \Delta_t(X_t)^j \right] + \max_{t \geq 1} \sqrt{\mathbb{E}_{X_t \sim \tilde{Q}_t} \|\Delta_t(X_t)\|^2} (\log T)\varepsilon.$$

When $\tilde{Q}_0$ does not have a p.d.f., a similar upper bound is applied to $\mathrm{KL}(\tilde{Q}_1 \| \widehat{P}_1)$ such that $\mathrm{W}_2(\tilde{Q}_1, \tilde{Q}_0)^2 \lesssim (1-\alpha_1)d$ (see Corollary 1 in Appendix F.5).

To explain the three error terms in Theorem 1, $\mathcal{W}_{oracle}$ captures the error assuming that one has access to (a close estimate of) $\nabla \log \tilde{q}_t, \forall t \geq 1$. This error is independent of the score mismatch $\Delta_t$, and it decays as $\tilde{O}(T^{-1})$ under Assumption 3 (Liang et al., 2024, Theorem 1). The remaining two error terms $\mathcal{W}_{bias}$ and $\mathcal{W}_{vanish}$ arise from the mismatched sampling process. Both terms become zero if $\Delta_t \equiv 0$ for all $t \geq 1$, which corresponds to the score-matched case. Under Assumptions 3 and 4, $\mathcal{W}_{vanish}$ decays as $\tilde{O}(T^{-1})$ under an additional mild condition (see Lemma 5 in Appendix G), and $\mathcal{W}_{bias}$ asymptotically approaches a constant. Combining all three terms, score mismatch causes an asymptotic distributional bias between $\tilde{Q}_0$ and $\widehat{P}_0$.

To further understand $\mathcal{W}_{bias}$, note that $1 - \alpha_t$ is usually summable under Assumption 4 (cf. Lemmas 7 and 10). Thus, the bias can be further upper-bounded by the maximum step-wise mismatch $\max_{t \geq 1} \mathbb{E}_{X_t \sim \tilde{Q}_t} \|\Delta_t(X_t)\|^2$. In case that $\mu_t(x_t) = \frac{1}{\sqrt{\alpha_t}} (x_t + (1-\alpha_t)g_t(x_t))$ (as for the zero-shot sampler in (6)), define a measure $\tilde{P}_t$ such that $g_t(x_t) = \nabla \log \tilde{p}_t(x_t)$. Then, from (2),

$$\mathbb{E}_{X_t \sim \tilde{Q}_t} \|\Delta_t(X_t)\|^2 = \mathbb{E}_{X_t \sim \tilde{Q}_t} \left\| \nabla \log \frac{\tilde{q}_t(X_t)}{\tilde{p}_t(X_t)} \right\|^2 =: \mathcal{F}(\tilde{Q}_t \| \tilde{P}_t).$$

where $\mathcal{F}(Q \| P)$ denotes the *Fisher divergence* (or called relative Fisher information) between $Q$ and $P$. In Section 4, this distributional bias $\mathcal{W}_{bias}$ inspires us to design a novel zero-shot DDPM sampler, the BO-DDNM sampler, that minimizes the asymptotic bias.

Next we provide an upper bound with explicit dimensional dependency, for any $Q_0$ that has finite sixth moment such as Gaussian mixture $Q_0$'s and those $Q_0$'s having bounded support. To this end, we consider a special noise schedule first proposed in Li et al. (2024c):

$$1 - \alpha_1 = \delta, \quad 1 - \alpha_t = \frac{c \log T}{T} \min \left\{ \delta \left( 1 + \frac{c \log T}{T} \right)^t, 1 \right\}, \forall 2 \leq t \leq T \tag{8}$$

for any constants $(c, \delta)$ such that $c > 1$ and $\delta e^c > 1$. Note that this noise schedule corresponds to early-stopping in the literature (Chen et al., 2023a; Benton et al., 2024a). With the $\alpha_t$ in (8), we can show that the regularity condition Assumption 3 holds for a quite general set of distributions (see Lemma 5 in Appendix G).

**Theorem 2.** *Suppose that $\mathbb{E}_{X_0 \sim \tilde{Q}_0} \|X_0\|^6 \lesssim d^3$. Further, suppose that $\Delta_t$ satisfies that $\mathbb{E}_{X_t \sim \tilde{Q}_t} \|\Delta_t(X_t)\|^4 \lesssim \frac{\bar{\alpha}_t^2}{(1-\bar{\alpha}_t)^{2r}} d^{2\gamma}$ with some $\gamma, r \geq 1$ for all $t \geq 2$. Then, if the estimation error satisfies Assumption 2 and if $\Delta_t$ satisfies Assumption 4, with the $\alpha_t$ in (8) such that $\delta \ll 1$ and*

$c \asymp \log(1/\delta)$, we have, for some $\tilde{Q}_1$ such that $\mathrm{W}_2(\tilde{Q}_1, \tilde{Q}_0)^2 \lesssim \delta d$,

$$
\begin{aligned}
\mathrm{KL}(\tilde{Q}_1 \| \widehat{P}_1) \lesssim\ & d^\gamma \delta^{-r} \left(1 - \tfrac{2\log(1/\delta)\log T}{T}\right) \\
& + \max\{d^{(3+\gamma)/2}\delta^{-\frac{r+2}{2}}, d^{1+\gamma}\delta^{-(r-1)}\}\tfrac{(\log T)^2}{T} + d^{\gamma/2}\delta^{-r/2}(\log T)\varepsilon.
\end{aligned}
$$

Note that Theorem 2 provides the *first* performance guarantee with *explicit* dimensional dependence for general score-mismatched DDPMs. Here the finite sixth moment is a technical condition to guarantee small expected difference of the first-order Taylor polynomial in case of mismatched scores (see Lemma 5 in Appendix G). Later, Theorem 2 will be useful to provide guarantees for zero-shot conditional samplers under linear models (Theorem 4).

## 4   ZERO-SHOT CONDITIONAL DDPM SAMPLERS

As we discuss before, an important scenario of score-mismatched diffusion models is the zero-shot conditional problem, where certain information $y$ is given. In this section, we apply our general results for score-mismatch DDPMs in Section 3 to studying zero-shot conditional DDPM samplers. In the following, we are particularly interested in the linear conditional model in (4). We take the same Assumptions 1, 3 and 4 (albeit with changed notations), and further adopt the following common assumption on the *unconditional* score estimation (Chen et al., 2023a;d; Liang et al., 2024).

**Assumption 5** (Estimation Error of Unconditional Score). Suppose that $s_t$ satisfies

$$
\tfrac{1}{T} \sum_{t=1}^T \mathbb{E}_{X_t \sim Q_{t|y}} \left\| s_t(X_t) - \nabla \log q_t(X_t) \right\|^2 \le \varepsilon^2, \text{ where } \varepsilon^2 = \tilde{O}(T^{-2}).
$$

Note that, with the zero-shot sampler defined in (6), since $\left\| I_d - H^\dagger H \right\| = 1$, we have, $\forall x \in \mathbb{R}^d$,

$$
\|\widehat{\mu}_{t,y} - \mu_{t,y}\|^2 = \frac{(1-\alpha_t)^2}{\alpha_t} \left\| (I_d - H^\dagger H)(s_t - \nabla \log q_t) \right\|^2 \le \frac{(1-\alpha_t)^2}{\alpha_t} \left\| s_t - \nabla \log q_t \right\|^2. \quad (9)
$$

Therefore, Assumption 5 directly implies Assumption 2, and thus Theorem 1 (as well as Corollary 1) still holds under Assumptions 1 and 3 to 5.

### 4.1   A NOVEL BIAS-OPTIMAL ZERO-SHOT SAMPLER

Guided by the performance guarantee characterized in Theorem 1, we will propose a novel *optimized* zero-shot condition sampler. With the zero-shot sampler defined in (6), the goal is to choose the $f_{t,y}$ function that minimizes the convergence error for each $y \in \mathbb{R}^p$ and $t \ge 1$.

Specifically, it is observed in Theorem 1 that the convergence error in terms of the KL-divergence will have an asymptotic distributional bias given by $\mathcal{W}_{\mathrm{bias}}$. As follows, we characterize an optimal $f_{t,y}$ that minimizes $\mathcal{W}_{\mathrm{bias}}$, which thus yields a corresponding bias-optimal zero-shot sampler.

**Theorem 3.** *Define* $\Sigma_{t|0,y} := \bar{\alpha}_t \sigma_y^2 H^\dagger (H^\dagger)^\intercal + (1 - \bar{\alpha}_t) I_d$. *For any $Q_0$ and $t \ge 1$, we have*

$$
\nabla \log q_{t|y}(x_t|y) = \Sigma_{t|0,y}^{-1}(\sqrt{\bar{\alpha}_t} H^\dagger y - x_t) + \tfrac{\sqrt{\bar{\alpha}_t}}{1 - \bar{\alpha}_t}(I_d - H^\dagger H)\mathbb{E}_{Q_{0|t,y}}[X_0|x_t, y].
$$

*Also, recall the sampler in* (6) *and define* $f_{t,y}^*$ *as*

$$
f_{t,y}^*(x_t) := \Sigma_{t|0,y}^{-1}\left(\sqrt{\bar{\alpha}_t} H^\dagger y - H^\dagger H x_t\right). \quad (10)
$$

*Also recall $\Delta_{t,y}$ from* (7). *Then, $f_{t,y}^*$ satisfies that, for all $t \ge 1$ and fixed $y \in \mathbb{R}^p$,*

$$
f_{t,y}^* \in \underset{f_{t,y}:(I_d - H^\dagger H)f_{t,y} \equiv 0}{\arg\min} \|\Delta_{t,y}\|^2, \quad Q_{t|y}\text{–almost surely.}
$$

The sampler $f_{t,y}^*(x_t)$ defined in (10) provides a bias-optimal zero-shot conditional DDPM sampler. In the case with $\sigma_y = 0$, such an optimal sampler coincides with the regular DDNM sampler in Wang et al. (2023) (see Appendix D). Thus, we call this sampler as **Bias-Optimal (BO) DDNM** sampler. With (10), we can also calculate the minimum step-wise mismatch as

$$
\underset{f_{t,y}:(I_d - H^\dagger H)f_{t,y} \equiv 0}{\min} \mathbb{E}_{X_t \sim Q_{t|y}} \|\Delta_{t,y}\|^2 = \mathbb{E}_{X_t \sim Q_{t|y}} \left\| \nabla \log \frac{q_{t|y}(X_t)}{q_t(X_t)} \right\|_{(I_d - H^\dagger H)}^2,
$$

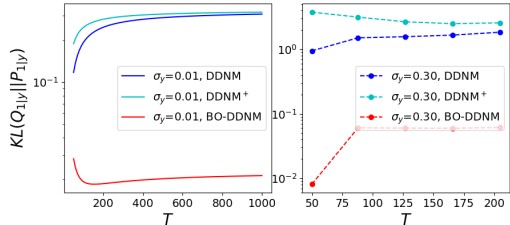 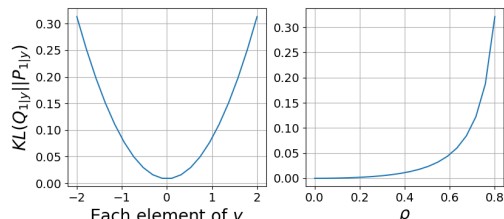

Figure 1: Comparison of BO-DDNM, DDNM and DDNM$^+$ for Gaussian (left) and Gaussian mixture (right) $Q_0$ under measurement noise.

Figure 2: Distributional bias as a function of the conditioning $y$ (left) and the correlation coefficient $\rho$ (right) for Gaussian $Q_0$.

which is the projected Fisher divergence between $Q_{t|y}$ and $Q_t$ on $\mathrm{range}(I_d - H^\dagger H)$.

In the following lemma, we provide the performance bound for BO-DDNM when $Q_0$ has bounded support. For comparison, we also derive the theoretical performance of vanilla DDNM, denoted as $f_{t,y}^N$.

**Theorem 4** (BO-DDNM vs. DDNM). *Suppose that $\|X_0\|^2 \le R^2 d$ a.s. under $Q_0$. Suppose that Assumptions 1 and 5 hold. Then, with the conditional sampler $f_{t,y}^*$ in (10), Theorem 2 holds with $\gamma = 1$ and $r = 2$. Also, with the conditional sampler $f_{t,y}^N := (1 - \bar{\alpha}_t)^{-1} \left( \sqrt{\bar{\alpha}_t} H^\dagger y - H^\dagger H x_t \right)$, if further $\|H^\dagger\| \lesssim 1$, then Theorem 2 holds with $\gamma = 1$ and $r = 4$.*

Theorem 4 establishes the *first* result applicable to DDNM-type zero-shot conditional samplers for any linear conditional models on those target distributions having bounded support.

**Advantage of BO-DDNM over DDNM and DDNM$^+$:** When there is positive measurement noise, Theorem 4 indicates that our BO-DDNM sampler that uses $f_{t,y} = f_{t,y}^*$ enjoys a smaller asymptotic bias than DDNM that uses $f_{t,y}^N$ with the $\alpha_t$ in (8) ($\delta^{-2}$ vs. $\delta^{-4}$). Note that the DDNM sampler corresponds to $f_{t,y} = f_{t,y}^N$ (see Appendix D). Such an advantage is also demonstrated by our numerical experiment. In Figure 1, we numerically compared modified conditional zero-shot sampler (as given in (10)) with the DDNM and DDNM$^+$ sampler for both Gaussian and Gaussian mixture $Q_0$'s at different levels of measurement noise. It is observed that the optimal BO-DDNM sampler achieves a much lower bias than both the DDNM and the DDNM$^+$ samplers numerically, especially when $\sigma_y^2$ becomes large.

## 4.2 BO-DDNM Sampler for Gaussian Mixture $Q_0$

In this section, we focus on the convergence dependency on other system parameters of the BO-DDNM sampler, including the chosen $y$. In particular, we restrict our attention to Gaussian mixture $Q_0$'s and to a special conditional model, where $H = (I_p \quad 0)$. This choice can be seen in many applications, such as image super-resolution and inpainting (after reorganizing the pixels), where $I_p$ corresponds to the locations of the given pixels (Wang et al., 2023; Song et al., 2023a). We assume positive measurement noise. We introduce the notation $[\Sigma_0]_{ab}$ to denote the variance components that correspond to the space of $a \times b$ where $a, b \in \{y, \bar{y}\}$.

The following Proposition 1 gives an upper bound on the asymptotic bias for Gaussian mixture $Q_0$.

**Proposition 1.** *Suppose that $Q_0$ is Gaussian mixture with equal variance, whose p.d.f. is given by $q_0(x_0) = \sum_{n=1}^N \pi_n q_{0,n}(x_0)$, where $q_{0,n}$ is the p.d.f. of $\mathcal{N}(\mu_{0,n}, \Sigma_0)$ and $\pi_n \in [0,1]$ is the mixing coefficient with $\sum_{n=1}^N \pi_n = 1$. Suppose that $H = (I_p \quad 0)$, and adopt $f_{t,y}^*$ in (10) and $\alpha_t$ in Definition 1. Write $\lambda_1 \ge \cdots \ge \lambda_d > 0$ and $\tilde{\lambda}_1 \ge \cdots \ge \tilde{\lambda}_{d-p} > 0$ as the eigenvalues of $\Sigma_0$ and $[\Sigma_0]_{\bar{y}\bar{y}}$, respectively. Then,*

$$\mathbb{E}_{X_t \sim Q_{t|y}} \|\Delta_{t,y}(X_t)\|^2$$
$$\lesssim \bar{\alpha}_t d + \bar{\alpha}_t^2 \frac{\|[\Sigma_0]_{y\bar{y}}\|^2}{\min\{\tilde{\lambda}_{d-p}, 1\}^2 \min\{\lambda_d, 1\}^2} \max \left\{ d(\lambda_1 + \sigma_y^2) + \sum_{n=1}^N \pi_n \left\| H^\dagger y - H^\dagger H \mu_{0,n} \right\|^2, d \right\}$$
$$\lesssim \bar{\alpha}_t \left( d + \sum_{n=1}^N \pi_n \left\| H^\dagger y - H^\dagger H \mu_{0,n} \right\|^2 \right).$$

Proposition 1 indicates that three factors affect (an upper bound on) the asymptotic bias. (i) The measurement noise variance $\sigma_y^2$ determined by the system nature has an increasing effect on the bias. (ii) The averaged distance $\sum_{n=1}^{N} \pi_n \left\| H^\dagger y - H^\dagger H \mu_{0,n} \right\|^2$ between $H^\dagger y$ and $H^\dagger H \mu_{0,n}$ captures the quadratic dependency in $y$, as illustrated in the left plot of Figure 2. (iii) The correlation between $Hx_0$ and $(I_d - H^\dagger H)x_0$ of each mixture component contributes positively to the bias, which is contained in the factor $\frac{\|[\Sigma_0]_{y\bar{y}}\|^2}{\min\{\lambda_d,1\}^2}$. To see this, consider $\sigma_y^2 = 0$ and a specific Gaussian example with $d = 2$, $p = 1$, and $\Sigma_0 = \begin{pmatrix} \sigma_{11}^2 & \rho\sigma_{11}\sigma_{22} \\ \rho\sigma_{11}\sigma_{22} & \sigma_{22}^2 \end{pmatrix}$. As the correlation coefficient $\rho$ increases, $\Sigma_0$ becomes closer to be singular, and thus $\lambda_d$ decreases to 0. Also, $\|[\Sigma_0]_{y\bar{y}}\|^2 = \rho^2 \sigma_{11}^2 \sigma_{22}^2$ increases quadratically with $\rho$. Hence, this factor $\frac{\|[\Sigma_0]_{y\bar{y}}\|^2}{\min\{\lambda_d,1\}^2}$ grows unboundedly as $\rho \to 1$, as does $\mathbb{E}_{X_t \sim Q_{t|y}} \|\Delta_{t,y}(X_t)\|^2$. Such dependency on the correlation is illustrated numerically in the right plot of Figure 2.

The following theorem characterizes the conditional KL divergence when $Q_0$ is mixture Gaussian. In particular, we can show Assumption 4 holds with any $\alpha_t$ that satisfies Definition 1 when $Q_0$ is Gaussian mixture (see Lemma 8 in Appendix I.5).

**Theorem 5.** *Suppose the same conditions in Proposition 1 hold and $\sigma_y^2 > 0$. Suppose that Assumption 5 holds. Take $f_{t,y}^*$ in (10) and $\alpha_t$ that further satisfies $\sum_{t=1}^{T}(1 - \alpha_t)\bar{\alpha}_t = 1 + o(1)$. Then,*

$$\mathrm{KL}(Q_{0|y} \| \widehat{P}_{0|y}) \lesssim \left( d + \sum_{n=1}^{N} \pi_n \left\| H^\dagger y - H^\dagger H \mu_{0,n} \right\|^2 \right) +$$

$$\left( d^2 + \sum_{n=1}^{N} \pi_n \left\| H^\dagger y - H^\dagger H \mu_{0,n} \right\|^4 \right) \frac{(\log T)^2}{T} + \sqrt{d + \sum_{n=1}^{N} \pi_n \left\| H^\dagger y - H^\dagger H \mu_{0,n} \right\|^2} (\log T)\varepsilon.$$

Although Proposition 1 and Theorem 5 assume $H = (I_p \quad 0)$, extension to general $H$ is straightforward by modifying the proof of Lemma 8 and using the fact that $\left\| H^\dagger H \right\| = \left\| I_d - H^\dagger H \right\| = 1$.

This is the *first* convergence result for zero-shot samplers where explicit dependency on the conditioning $y$ is derived for Gaussian mixture targets. Note that the extra condition on $\alpha_t$ can be verified for both constant $\alpha_t$ (Lemma 10) and that in (8) (Lemma 7). Among the three terms in Theorem 5, the first term is the asymptotic bias analyzed in Proposition 1. Since the last two terms decrease to zero as $T \to \infty$, the asymptotic KL divergence will also approach some non-zero limit of order $d$.

The proof of Theorem 5 is non-trivial because from Theorem 1 we need to figure out the dependency on $y$ in all first three orders of partial derivatives of a Gaussian mixture density, which is generally hard to express. To this end, we restrict focus to a particular linear model where explicit dependency can be sought. The result can be extended to the case of $\sigma_y^2 = 0$ with the $\alpha_t$ in (8) (see Remark 2).

## 5 CONCLUSION

In this paper, we have provided convergence guarantees for the general score-mismatched diffusion models, which are specialized to zero-shot conditional samplers. For linear conditional models, we also designed an optimal BO-DDNM sampler that minimizes the asymptotic bias, for which we showed the dependencies on the system parameters. One future direction is to explore zero-shot samplers that use higher-order derivatives of the log-densities, which might achieve better convergence results.

## ACKNOWLEDGMENTS

This work has been supported in part by the U.S. National Science Foundation under the grants: DMS-2134145, CCF-1900145, NSF AI Institute (AI-EDGE) 2112471, CNS-2312836, CNS-2225561, ONR grant N000142412729, and was sponsored by the Army Research Laboratory under Cooperative Agreement Number W911NF-23-2-0225. The views and conclusions contained in this document are those of the authors and should not be interpreted as representing the official policies, either expressed or implied, of the Army Research Laboratory or the U.S. Government. The U.S. Government is authorized to reproduce and distribute reprints for Government purposes notwithstanding any copyright notation herein.

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

# Appendix

## A   FULL LIST OF NOTATIONS

For any two functions $f(d, \delta, T)$ and $g(d, \delta, T)$, we write $f(d, \delta, T) \lesssim g(d, \delta, T)$ (resp. $f(d, \delta, T) \gtrsim g(d, \delta, T)$) for some universal constant (not depending on $\delta$, $d$ or $T$) $L < \infty$ (resp. $L > 0$) if $\limsup_{T \to \infty} |f(d, \delta, T)/ g(d, \delta, T)| \leq L$ (resp. $\liminf_{T \to \infty} |f(d, \delta, T)/g(d, \delta, T)| \geq L$). We write $f(d, \delta, T) \asymp g(d, \delta, T)$ when both $f(d, \delta, T) \lesssim g(d, \delta, T)$ and $f(d, \delta, T) \gtrsim g(d, \delta, T)$ hold. Note that the dependence on $\delta$ and $d$ is retained with $\lesssim, \gtrsim, \asymp$. We write $f(d, \delta, T) = O(g(T))$ (resp. $f(d, \delta, T) = \Omega(g(T))$) if $f(d, \delta, T) \lesssim L(d, \delta)g(T)$ (resp. $f(d, \delta, T) \gtrsim L(d, \delta)g(T)$) holds for some $L(d, \delta)$ (possibly depending on $\delta$ and $d$). We write $f(d, \delta, T) = o(g(T))$ if $\limsup_{T \to \infty} |f(d, \delta, T) /g(T)| = 0$. We write $f(d, \delta, T) = \tilde{O}(g(T))$ if $f(d, \delta, T) = O(g(T)(\log g(T))^k)$ for some constant $k$. Note that the big-$O$ notation omits the dependence on $\delta$ and $d$. In the asymptotic when $\varepsilon^{-1} \to \infty$, we write $f(d, \varepsilon^{-1}) = \mathcal{O}(g(d, \varepsilon^{-1}))$ if $f(d, \delta, \varepsilon^{-1}) \lesssim g(d, \delta, \varepsilon^{-1})(\log g(\varepsilon^{-1}))^k$ for some constant $k$. Unless otherwise specified, we write $x^i (1 \leq i \leq d)$ as the $i$-th element of a vector $x \in \mathbb{R}^d$ and $[A]^{ij}$ as the $(i, j)$-th element of a matrix $A$. For a function $f(x) : \mathbb{R}^d \to \mathbb{R}$, we write $\partial_i f(z)$ as a shorthand for $\frac{\partial}{\partial x^i} f(x)\big|_{x=z}$, and similarly for higher moments. For a vector (resp. matrix), all norms, if not explicitly specified, are referred to 2-norm (resp. spectral norm). For a vector $x$ and matrix $P$, define $\|x\|_P := \sqrt{a^\intercal P a}$. For matrices $A, B$, $\mathrm{Tr}(A)$ is the trace of $A$, and $A \preceq B$ means that $B - A$ is positive semi-definite. For a positive integer $n$, $[n] := \{1, \dots, n\}$.

## B   RELATED WORKS ON UNCONDITIONAL DDPM SAMPLERS

Given time-averaged $L^2$ unconditional score estimation error (Hyvärinen, 2005), polynomial-time convergence guarantees have been established for wide families of target distributions (De Bortoli et al., 2021; Chen et al., 2023d; Lee et al., 2023; Chen et al., 2023a; Benton et al., 2024a; Pedrotti et al., 2023; Conforti et al., 2023). For all target distributions with finite second moment, under $L^2$ score estimation error, $\mathcal{O}(d \log(1/\delta)^2/\varepsilon^2)$ number of steps are sufficient to achieve $\varepsilon^2$ KL divergence between the $\delta$-perturbed target distribution and the generated distribution using the specially designed exponential-decay-then-constant step-sizes (Benton et al., 2024a; Conforti et al., 2023). The analysis usually involves applying the Girsanov change-of-measure framework and the Fokker-Plank equation (Chen et al., 2023d;a) to either the original SDE diffusion process or some transformed process (Benton et al., 2024a; Conforti et al., 2023), followed by an analysis of the discretization of the continuous-time process. More recently, similar convergence guarantees have been established using non-SDE-type techniques, such as with typical sets (Li et al., 2024c) and with tilting factor representations (Liang et al., 2024). Here the new analysis introduced in Liang et al. (2024) is applicable to a larger set of step-sizes (equivalently, noise schedules) than the ones commonly used in previous analyses (Chen et al., 2023a; Benton et al., 2024a; Conforti et al., 2023). In this paper, we employ the same analytical framework as in Liang et al. (2024).

Some other works analyzed sampling errors using a different measure (the Wasserstein-2 distance) (Bruno et al., 2023; Gao et al., 2023; Gao & Zhu, 2024). Beyond stochastic samplers, another line of studies provided theoretical guarantees for the deterministic sampler corresponding to DDPM (Chen et al., 2023e;c; Huang et al., 2024). Besides, Cheng et al. (2023); Benton et al. (2024b); Jiao et al. (2024); Gao et al. (2024) provided guarantees for the closely-related flow-matching model, which learns a deterministic coupling between any two distributions. Also, Lyu et al. (2024); Li et al. (2024b) provided convergence guarantees for the closely-related consistency models (Song et al., 2023b). Finally, in order to achieve an end-to-end analysis, several works also developed sample complexity bounds to achieve the $L^2$ score estimation error for a variety of distributions (Oko et al., 2023; Shah et al., 2023; Gatmiry et al., 2024; Chen et al., 2024; Cole & Lu, 2024; Zhang et al., 2024; Mei & Wu, 2023; Chen et al., 2023b).

## C    DETAILS OF NUMERICAL SIMULATIONS

In Figure 1, we compared the performances of our optimal BO-DDNM sampler (with the $f_{t,y}^*$ in (10)) against the DDNM and DDNM$^+$ samplers (Wang et al., 2023) at different levels of $\sigma_y^2$. For Gaussian, we use $\mu_0 = 0$, $d = 4$, $p = 2$, and $y = (0.5 \quad 0.5)$. We first randomly generate a positive definite matrix $\Sigma_0$ and uniformly sample $\rho \in [0.4, 0.7)$, and then this correlation coefficient is enforced for any $[\Sigma_0]^{ij}$ where $i \in [p]$ and $j \in \{p+1, \ldots, d\}$. We use the noise schedule in (8) with $c = 3$ and $\delta = 0.0001$ for Gaussian $Q_0$. For Gaussian mixture, we use $N = 2$, $d = 2$, $p = 1$, and $y = 1$. We set $\pi_n = (0.4 \quad 0.6)$, $\text{diag}(\Sigma_0) = (0.1 \quad 1)$, and $\rho = 0.6$. We further uniformly sample $\{\mu_{0,n}\}_{n=1}^N$ in the space $[-1, 1) \times [-1, 1)$. We use the noise schedule in (8) with $c = 4$ and $\delta = 0.02$ for Gaussian mixture $Q_0$. We use 150000 samples to estimate the divergence when $Q_0$ is Gaussian mixture.

In Figure 2, we numerically verify the exact bias in KL divergence as a function of $y$ and $\rho$ for Gaussian $Q_0$. Here $Q_0 = \mathcal{N}(0, \Sigma_0)$, $d = 4$ and $p = 2$. Suppose that $\sigma_y^2 = 0$. We assume that each element of $y$ has equal values. The correlation coefficient $\rho$ is enforced for any pair of $x^i$ and $x^j$ where $i \in [p]$ and $j \in \{p+1, \ldots, d\}$. We first randomly generate a positive definite matrix $\Sigma_0$ and then enforce the correlation condition for any $x^i$ and $x^j$ where $i \in [p]$ and $j \in \{p+1, \ldots, d\}$. We use a sufficiently large number of steps $T = 20000$. The conditional sampler is set as $f_{t,y} = f_{t,y}^*$ given in (10). The noise schedule in (8) with $c = 3$ and $\delta = 0.0001$ is used.

## D    DERIVATION OF SCORE BIAS FOR EXISTING ZERO-SHOT DDPM SAMPLERS

In this section we show some examples of zero-shot conditional samplers proposed in the literature and in particular how they are related to the formulation of interest in (6). We recall the notations $H$, $y$, and $\sigma_t$ from Section 2.3. Also denote

$$\mu_t := \frac{1}{\sqrt{\alpha_t}} x_t + \frac{1 - \alpha_t}{\sqrt{\alpha_t}} \nabla \log q_t(x_t) = \mathbb{E}_{X_{t-1} \sim Q_{t-1|t}(\cdot|x_t)}[X_{t-1}|x_t]$$

which is the mean of the unconditional reverse-step at time $t \geq 1$.

### D.1    COME-CLOSER-DIFFUSE-FASTER (CCDF)

We first examine the Come-Closer-Diffuse-Faster (CCDF) algorithm (Chung et al., 2022b). The CCDF algorithm using DDPM samplers gives that

$$x'_{t-1} = \mu_t + \sigma_t z_{t,1},$$
$$x_{t-1} = (I - H^\dagger H)x'_{t-1} + \sqrt{\bar{\alpha}_t} H^\dagger y + \sqrt{1 - \bar{\alpha}_t} z_{t,2}$$

where $z_{t,1}, z_{t,2} \overset{i.i.d.}{\sim} \mathcal{N}(0, I_d)$ are standard Gaussian random variables. Thus, the conditional mean of the update is

$$\mu_{t,y} = (I - H^\dagger H)\mu_t + \sqrt{\bar{\alpha}_{t-1}} H^\dagger y$$
$$= \frac{1}{\sqrt{\alpha_t}} x_t + \frac{1 - \alpha_t}{\sqrt{\alpha_t}}(I - H^\dagger H)\nabla \log q_t(x_t) + \sqrt{\bar{\alpha}_t} H^\dagger y - \frac{1}{\sqrt{\alpha_t}} H^\dagger H x_t$$

in which

$$f_{t,y}(x_t) = \frac{1}{1 - \alpha_t}\left(\sqrt{\alpha_t}\sqrt{\bar{\alpha}_t} H^\dagger y - H^\dagger H x_t\right).$$

### D.2    DDNM AND DDNM$^+$

Next, we examine the DDNM algorithm and its modified version DDNM$^+$ (Wang et al., 2023). We first note that the unconditional DDPM satisfies that (cf. (Ho et al., 2020, Equations (7) and (11))),

$$\mu_t := \frac{\sqrt{\bar{\alpha}_{t-1}}(1 - \alpha_t)}{1 - \bar{\alpha}_t} x_{0|t} + \frac{\sqrt{\alpha_t}(1 - \bar{\alpha}_{t-1})}{1 - \bar{\alpha}_t} x_t,$$

$$x_{0|t} := \frac{1}{\sqrt{\bar{\alpha}_t}} x_t + \frac{1 - \bar{\alpha}_t}{\sqrt{\bar{\alpha}_t}} \nabla \log q_t(x_t) = \mathbb{E}_{X_0 \sim Q_{0|t}(\cdot|x_t)}[X_0|x_t]. \tag{11}$$

Combining these two lines, we have $\mu_t = \frac{1}{\sqrt{\alpha_t}}(x_t + (1 - \alpha_t)\nabla \log q_t(x_t))$. In DDNM, $x_{0|t}$ is projected along the direction of the given $y$, which yields

$$x_{0|t,y} := H^\dagger y + (I_d - H^\dagger H)x_{0|t},$$

and the corresponding conditional mean of the update becomes

$$\mu_{t,y} = \frac{\sqrt{\bar{\alpha}_{t-1}}(1 - \alpha_t)}{1 - \bar{\alpha}_t}x_{0|t,y} + \frac{\sqrt{\alpha_t}(1 - \bar{\alpha}_{t-1})}{1 - \bar{\alpha}_t}x_t.$$

Thus,

$$\begin{aligned}
\mu_{t,y} &= \frac{\sqrt{\bar{\alpha}_{t-1}}(1 - \alpha_t)}{1 - \bar{\alpha}_t}\left(H^\dagger y + (I_d - H^\dagger H)x_{0|t}\right) + \frac{\sqrt{\alpha_t}(1 - \bar{\alpha}_{t-1})}{1 - \bar{\alpha}_t}x_t \\
&\overset{(i)}{=} (I_d - H^\dagger H)\mu_t + H^\dagger\left(\frac{\sqrt{\bar{\alpha}_{t-1}}(1 - \alpha_t)}{1 - \bar{\alpha}_t}y + \frac{\sqrt{\alpha_t}(1 - \bar{\alpha}_{t-1})}{1 - \bar{\alpha}_t}Hx_t\right) \\
&= \frac{1}{\sqrt{\alpha_t}}x_t + \frac{1 - \alpha_t}{\sqrt{\alpha_t}}(I_d - H^\dagger H)\nabla \log q_t(x_t) \\
&\quad + \left(\frac{\sqrt{\bar{\alpha}_{t-1}}(1 - \alpha_t)}{1 - \bar{\alpha}_t}H^\dagger y + \frac{\sqrt{\alpha_t}(1 - \bar{\alpha}_{t-1})}{1 - \bar{\alpha}_t}H^\dagger Hx_t - \frac{1}{\sqrt{\alpha_t}}H^\dagger Hx_t\right)
\end{aligned}$$

where $(i)$ follows from (11). Thus, to express this conditional mean in the form of (6),

$$\begin{aligned}
f_{t,y}(x_t) &= \frac{\sqrt{\bar{\alpha}_t}}{1 - \bar{\alpha}_t}H^\dagger y + \frac{1}{1 - \alpha_t}\left(\frac{\alpha_t(1 - \bar{\alpha}_{t-1})}{1 - \bar{\alpha}_t} - 1\right)H^\dagger Hx_t \\
&= \frac{1}{1 - \bar{\alpha}_t}\left(\sqrt{\bar{\alpha}_t}H^\dagger y - H^\dagger Hx_t\right).
\end{aligned}$$

Here note that $f_{t,y}(x_t)$ is supported on $\text{range}(H^\dagger H)$. Also note that for DDNM, $f_{t,y} = f^*_{t,y}$, which is the BO-DDNM sampler defined in (10), when there is no measurement noise (i.e., $\sigma_y^2 = 0$).

Next we investigate its modified version, DDNM$^+$, in particular when $H = (I_p \quad 0)$. To relate the notations of (Wang et al., 2023, Section 3.3 and Appendix I) with ours, note that $\Sigma = A = H$, $U = I_p$, $V = I_d$, $s_1, \ldots, s_p = 1$, $s_{p+1}, \ldots, s_d = 0$, and $a = \frac{\sqrt{\bar{\alpha}_{t-1}}(1-\alpha_t)}{1-\bar{\alpha}_t}$. If $\sigma_t \geq a\sigma_y$, we have

$$\Sigma_t = I_d, \quad \Phi_t = \begin{pmatrix} (\sigma_t^2 - a^2\sigma_y^2)I_p & 0 \\ 0 & \sigma_t^2 I_{d-p} \end{pmatrix}.$$

Otherwise, if $\sigma_t < a\sigma_y$, we have

$$\Sigma_t = \begin{pmatrix} \frac{\sigma_t}{a\sigma_y}I_p & 0 \\ 0 & I_{d-p} \end{pmatrix}, \quad \Phi_t = \begin{pmatrix} 0 & 0 \\ 0 & \sigma_t^2 I_{d-p} \end{pmatrix}.$$

Observe that the only difference is on the space that supports $H^\dagger H$.

From (Wang et al., 2023, Equations (17) and (18)), we can write

$$\hat{x}_{0|t,y} := (I_d - H^\dagger H)x_{0|t} + \underbrace{\Sigma_t H^\dagger y + (I_d - \Sigma_t)H^\dagger Hx_{0|t}}_{\text{supported on } \text{range}(H^\dagger H)},$$

Thus, with similar arguments above,

$$\begin{aligned}
\mu_{t,y} &= \frac{\sqrt{\bar{\alpha}_{t-1}}(1 - \alpha_t)}{1 - \bar{\alpha}_t}\hat{x}_{0|t,y} + \frac{\sqrt{\alpha_t}(1 - \bar{\alpha}_{t-1})}{1 - \bar{\alpha}_t}x_t \\
&= \frac{1}{\sqrt{\alpha_t}}x_t + \frac{1 - \alpha_t}{\sqrt{\alpha_t}}(I_d - H^\dagger H)\nabla \log q_t(x_t) \\
&\quad + \frac{1 - \alpha_t}{\sqrt{\alpha_t}(1 - \bar{\alpha}_t)}\underbrace{\left(\sqrt{\bar{\alpha}_t}(\Sigma_t H^\dagger y + (I_d - \Sigma_t)H^\dagger Hx_{0|t}) - H^\dagger Hx_t\right)}_{\text{supported on } \text{range}(H^\dagger H)}
\end{aligned}$$

where $f_{t,y}$ is again supported on $\text{range}(H^\dagger H)$.

### D.3 Samplers Using Higher-Order Derivatives

Before we end this section, we note that the formulation in (6) only uses (estimates of) first-order derivatives of (unconditional) log-p.d.f.s (a.k.a. unconditional score functions). This might not correspond to the optimal zero-shot sampler, and in practice there have been methods that use both first- and second-order derivatives (namely, in $\partial x_{0|t}(x_t)/\partial x_t$) to achieve better zero-shot sampling performance (Chung et al., 2023; Song et al., 2023a). Nevertheless, the second-order derivatives might be hard to obtain, which require extra machine time and memory in the calculation. We leave investigations to use second-order derivatives in zero-shot conditional samplers as future work.

## E Proof Sketch of Theorem 1

We now provide a proof sketch of Theorem 1 to describe the idea of our analysis approach. The main technical challenge due to mismatched scores is to analyze the expected tilting factor under a mean-perturbed Gaussian, providing an upper bound of the asymptotic orders of all Gaussian non-centralized moments. See the full proof in Appendix F.

To begin, with Lemma 1, we decompose the total error as $\mathrm{KL}(\tilde{Q}_0\|\widehat{P}_0) \le \mathbb{E}_{X_T\sim\tilde{Q}_T}\left[\log\frac{\tilde{q}_T(X_T)}{\widehat{p}_T(X_T)}\right] +$ $\sum_{t=1}^{T}\mathbb{E}_{X_t,X_{t-1}\sim\tilde{Q}_{t,t-1}}\left[\log\frac{p_{t-1|t}(X_{t-1}|X_t)}{\widehat{p}_{t-1|t}(X_{t-1}|X_t)}\right]+\sum_{t=1}^{T}\mathbb{E}_{X_t,X_{t-1}\sim\tilde{Q}_{t,t-1}}\left[\log\frac{p_{t-1|t}(X_{t-1}|X_t)}{\widehat{p}_{t-1|t}(X_{t-1}|X_t)}\right]$. These three terms correspond respectively to the *initialization error*, *estimation error*, and *reverse-step error*. The initialization error can be bounded by $\bar{\alpha}_T d$ in order using (Liang et al., 2024, Lemma 3) under Assumption 1. Below we focus on the remaining two terms.

**Step 1: Bounding estimation error under mismatch (Lemma 2).** At each time $t = 1,\dots,T$, $\log(p_{t-1|t}(x_{t-1}|x_t)/\widehat{p}_{t-1|t}(x_{t-1}|x_t))$ has an explicit expression since they are conditional Gaussians with the same variance. However, differently from the typical matched case, the mean of $P_{t-1|t}$ (i.e., $\mu_t(x_t)$) is no longer equal to the posterior mean of $\tilde{Q}_{t-1|t}$ (i.e., $m_t(x_t)$). Their difference is contained in $\Delta_t(x_t)$, whose asymptotic order needs to be upper-bounded in light of Assumption 2.

**Step 2: Decomposing reverse-step error under mismatch (Equation (15)).** First we decompose the tilting factor as $\tilde{\zeta}_{t,t-1}(x_t, x_{t-1}) = \tilde{\zeta}_{\mathrm{mis}}(x_t, x_{t-1}) + \tilde{\zeta}_{\mathrm{van}}(x_t, x_{t-1})$, where

$$\tilde{\zeta}_{\mathrm{mis}} := \sqrt{\alpha_t}\Delta_t(x_t)^\mathsf{T}(x_{t-1} - m_t(x_t))$$

$$\tilde{\zeta}_{\mathrm{van}} := (\nabla\log\tilde{q}_{t-1}(m_t(x_t)) - \sqrt{\alpha_t}\nabla\log\tilde{q}_t(x_t))^\mathsf{T}(x_{t-1} - m_t(x_t)) + \sum_{p=2}^{\infty} T_p(\log\tilde{q}_{t-1}, x_t, m_t(x_t)).$$

Here $\tilde{\zeta}_{\mathrm{mis}}$ captures the factor that contributes to the total bias within $\tilde{\zeta}_{t,t-1}$. Define the oracle sampling process as $\tilde{P}_{t-1|t} = \mathcal{N}(m_{t,y}, \sigma_t^2 I_d)$. Then, the reverse-step error can be decomposed as

$$\mathbb{E}_{\tilde{Q}_{t,t-1}}[\tilde{\zeta}_{t,t-1}] - \mathbb{E}_{\tilde{Q}_t\times P_{t-1|t}}[\tilde{\zeta}_{t,t-1}] = \underbrace{\left(\mathbb{E}_{\tilde{Q}_{t,t-1}}[\tilde{\zeta}_{t,t-1}] - \mathbb{E}_{\tilde{Q}_t\times P_{t-1|t}}[\tilde{\zeta}_{t,t-1}]\right)}_{\mathcal{W}_{\mathrm{oracle,\,rev\text{-}step}}}$$

$$+ \underbrace{\left(\mathbb{E}_{\tilde{Q}_t\times\tilde{P}_{t-1|t}}[\tilde{\zeta}_{\mathrm{mis}}] - \mathbb{E}_{\tilde{Q}_t\times P_{t-1|t}}[\tilde{\zeta}_{\mathrm{mis}}]\right)}_{\mathcal{W}_{\mathrm{bias,\,rev\text{-}step}}} + \underbrace{\left(\mathbb{E}_{\tilde{Q}_t\times\tilde{P}_{t-1|t}}[\tilde{\zeta}_{\mathrm{van}}] - \mathbb{E}_{\tilde{Q}_t\times P_{t-1|t}}[\tilde{\zeta}_{\mathrm{van}}]\right)}_{\mathcal{W}_{\mathrm{vanish,\,rev\text{-}step}}}.$$

**Step 3: Bounding $\mathcal{W}_{\mathbf{oracle,\,rev\text{-}step}}$ and $\mathcal{W}_{\mathbf{bias,\,rev\text{-}step}}$ (Equations (16) and (17)).** Under Assumption 3, the dominant term of $\mathcal{W}_{\mathrm{oracle,\,rev\text{-}step}}$ is given by (Liang et al., 2024, Theorem 1). Also, the calculation of $\mathcal{W}_{\mathrm{bias,\,rev\text{-}step}}$ is reduced to the difference in conditional mean, which is proportional to $\|\Delta_t(x_t)\|^2$.

**Step 4: Bounding $\mathcal{W}_{\mathbf{vanish,\,rev\text{-}step}}$ (Lemmas 3 and 4).** To upper-bound $\mathcal{W}_{\mathrm{vanish,\,rev\text{-}step}}$, with results on the matched case in Liang et al. (2024), we need only to characterize the mean of $\tilde{\zeta}_{\mathrm{van}}$ under the mismatched posterior $P_{t-1|t}$. We determine the dominant order in the expected values of all Taylor polynomials, which includes calculating all non-centralized moments. We first calculate the first three non-centralized moments (Lemma 3) and then determine the asymptotic order of all higher moments (Lemma 4). With these, we can finally locate the terms of dominating order in $\mathcal{W}_{\mathrm{vanish,\,rev\text{-}step}}$.

## F  PROOF OF THEOREM 1 AND COROLLARY 1

Overall, the structure of the proof of Theorem 1 is similar to that for (Liang et al., 2024, Theorem 1). To start, we note that with similar arguments in (Liang et al., 2024, Equation 13), an upper bound on $\mathrm{KL}(\tilde{Q}_0 \| \widehat{P}_0)$ is given by

$$
\begin{aligned}
&\mathrm{KL}(\tilde{Q}_0 \| \widehat{P}_0)\\
&= \mathrm{KL}(\tilde{Q}_T \| \widehat{P}_T) + \sum_{t=1}^{T} \mathbb{E}_{X_t \sim \tilde{Q}_t} \left[ \mathrm{KL}(\tilde{Q}_{t-1|t}(\cdot|X_t) \| \widehat{P}_{t-1|t}(\cdot|X_t)) \right]\\
&\quad - \sum_{t=1}^{T} \mathbb{E}_{X_{t-1} \sim \tilde{Q}_{t-1}} \left[ \mathrm{KL}(\tilde{Q}_{t|t-1}(\cdot|X_{t-1}) \| \widehat{P}_{t|t-1}(\cdot|X_{t-1})) \right]\\
&\leq \mathrm{KL}(\tilde{Q}_T \| \widehat{P}_T) + \sum_{t=1}^{T} \mathbb{E}_{X_t \sim \tilde{Q}_t} \left[ \mathrm{KL}(\tilde{Q}_{t-1|t}(\cdot|X_t) \| \widehat{P}_{t-1|t}(\cdot|X_t)) \right]\\
&= \underbrace{\mathbb{E}_{X_T \sim \tilde{Q}_T} \left[ \log \frac{\tilde{q}_T(X_T)}{\widehat{p}_T(X_T)} \right]}_{\text{Term 1: initialization error}} + \underbrace{\sum_{t=1}^{T} \mathbb{E}_{X_t, X_{t-1} \sim \tilde{Q}_{t,t-1}} \left[ \log \frac{p_{t-1|t}(X_{t-1}|X_t)}{\widehat{p}_{t-1|t}(X_{t-1}|X_t)} \right]}_{\text{Term 2: estimation error}}\\
&\quad + \underbrace{\sum_{t=1}^{T} \mathbb{E}_{X_t, X_{t-1} \sim \tilde{Q}_{t,t-1}} \left[ \log \frac{\tilde{q}_{t-1|t}(X_{t-1}|X_t)}{p_{t-1|t}(X_{t-1}|X_t)} \right]}_{\text{Term 3: reverse-step error}}.
\end{aligned}
\tag{12}
$$

The last equality holds because $\widehat{p}_T = p_T$. Now, we provide an upper bound for the reverse-step error that is ready for further analysis. In the following lemma, we show that the mismatched $\tilde{q}_{t-1|t}$ is an exponentially tilted form of $p_{t-1|t}$.

**Lemma 1.** *Fixed* $t \geq 1$. *For any fixed* $x_t \in \mathbb{R}^d$, *as long as* $\tilde{q}_{t-1}$ *exists, we have*

$$
\tilde{q}_{t-1|t}(x_{t-1}|x_t) = \frac{p_{t-1|t}(x_{t-1}|x_t) e^{\tilde{\zeta}_{t,t-1}(x_t, x_{t-1})}}{\mathbb{E}_{X_{t-1} \sim P_{t-1|t}}[e^{\tilde{\zeta}_{t,t-1}(x_t, X_{t-1})}]}
$$

*where*

$$
\begin{aligned}
&\tilde{\zeta}_{t,t-1}(x_t, x_{t-1})\\
&:= \sqrt{\alpha_t} \Delta_t(x_t)^\mathsf{T}(x_{t-1} - m_t(x_t)) + (\nabla \log \tilde{q}_{t-1}(m_t(x_t)) - \sqrt{\alpha_t} \nabla \log \tilde{q}_t(x_t))^\mathsf{T}(x_{t-1} - m_t(x_t))\\
&\quad + \sum_{p=2}^{\infty} T_p(\log \tilde{q}_{t-1}, x_{t-1}, m_t(x_t)).
\end{aligned}
$$

*Here we define the* p-*th order term in the Taylor expansion of* $f(x)$ *around* $\mu$ *as*

$$
T_p(f, x, \mu) := \frac{1}{p!} \sum_{\gamma \in \mathbb{N}^d : \sum_i \gamma^i = p} \partial_{\boldsymbol{a}}^p f(\mu) \prod_{i=1}^{d} (x^i - \mu^i)^{\gamma^i}
$$

*where* $\boldsymbol{a} \in [d]^p$ *are the indices of differentiation in which the multiplicity of* $i \in [d]$ *is* $\gamma^i$.

*Proof.* See Appendix H.1.  □

We abbreviate $\tilde{\zeta}_{t,t-1} = \tilde{\zeta}_{t,t-1}(x_t, x_{t-1})$. Given the expression of $\tilde{\zeta}_{t,t-1}$, the conditional reverse-step error can be upper-bounded for any fixed $x_t$ as

$$
\begin{aligned}
&\mathbb{E}_{X_{t-1} \sim \tilde{Q}_{t-1|t}} \left[ \log \frac{\tilde{q}_{t-1|t}(X_{t-1}|x_t)}{p_{t-1|t}(X_{t-1}|x_t)} \right]\\
&= \mathbb{E}_{X_{t-1} \sim \tilde{Q}_{t-1|t}} \left[ \tilde{\zeta}_{t,t-1} - \log \mathbb{E}_{X_{t-1} \sim P_{t-1|t}}[e^{\tilde{\zeta}_{t,t-1}}] \right]
\end{aligned}
$$

$$\overset{(i)}{\leq} \mathbb{E}_{X_{t-1}\sim\tilde{Q}_{t-1|t}}\left[\tilde{\zeta}_{t,t-1}\right] + \mathbb{E}_{X_{t-1}\sim P_{t-1|t}}\left[-\log e^{\tilde{\zeta}_{t,t-1}}\right]$$

$$= \mathbb{E}_{X_{t-1}\sim\tilde{Q}_{t-1|t}}[\tilde{\zeta}_{t,t-1}] - \mathbb{E}_{X_{t-1}\sim P_{t-1|t}}[\tilde{\zeta}_{t,t-1}] \tag{13}$$

where in $(i)$ we use Jensen's inequality and note that $-\log(\cdot)$ is convex. Thus, from (12), we have an upper bound as

$$\mathrm{KL}(\tilde{Q}_0\|\widehat{P}_0) \leq \underbrace{\mathbb{E}_{X_T\sim\tilde{Q}_T}\left[\log\frac{\tilde{q}_T(X_T)}{\widehat{p}_T(X_T)}\right]}_{\text{Term 1: initialization error}} + \underbrace{\sum_{t=1}^{T}\mathbb{E}_{X_t,X_{t-1}\sim\tilde{Q}_{t,t-1}}\left[\log\frac{p_{t-1|t}(X_{t-1}|X_t)}{\widehat{p}_{t-1|t}(X_{t-1}|X_t)}\right]}_{\text{Term 2: estimation error}}$$

$$+ \underbrace{\sum_{t=1}^{T}\mathbb{E}_{X_{t-1}\sim\tilde{Q}_{t-1|t}}[\tilde{\zeta}_{t,t-1}] - \mathbb{E}_{X_{t-1}\sim P_{t-1|t}}[\tilde{\zeta}_{t,t-1}]}_{\text{Term 3: reverse-step error}}.$$

Here, using (Liang et al., 2024, Lemma 3), the initialization error can be upper-bounded as, when $T\to\infty$,

$$\mathbb{E}_{X_T\sim\tilde{Q}_T}\left[\log\frac{\tilde{q}_T(X_T)}{\widehat{p}_T(X_T)}\right] \leq \frac{1}{2}\mathbb{E}_{X_0\sim\tilde{Q}_0}\|X_0\|^2\,\bar{\alpha}_T + O\left(\bar{\alpha}_T^2\right).$$

This implies that, under Assumption 1 and if $c > 1$,

$$\mathbb{E}_{X_T\sim\tilde{Q}_T}\left[\log\frac{\tilde{q}_T(X_T)}{p_T(X_T)}\right] = o(T^{-1}).$$

Also, under Assumption 3, the higher-order Taylor polynomials enjoy exponential rate of decay in expectation, which is contained in powers of $(1-\alpha_t)$. Thus, we are allowed to exchange the limit (of Taylor expansion) and the expectation operators (cf. (Liang et al., 2024, Lemma 11)).

Now, we upper-bound the estimation error and reverse-step error under score mismatch separately.

### F.1 STEP 1: BOUNDING ESTIMATION ERROR UNDER MISMATCH

The following lemma provides an upper bound for the estimation error under score mismatch.

**Lemma 2.** *Under Assumptions 2 and 4, with the $\alpha_t$ satisfying Definition 1, we have*

$$\sum_{t=1}^{T}\mathbb{E}_{X_t,X_{t-1}\sim\tilde{Q}_{t,t-1}}\left[\log\frac{p_{t-1|t}(X_{t-1}|X_t)}{\widehat{p}_{t-1|t}(X_{t-1}|X_t)}\right] \lesssim \max_{t\geq 1}\sqrt{\mathbb{E}_{X_t\sim\tilde{Q}_t}\|\Delta_t(X_t)\|^2}(\log T)\varepsilon + (\log T)\varepsilon^2.$$

*Proof.* See Appendix H.2. $\qquad\qquad\square$

### F.2 STEP 2: DECOMPOSING REVERSE-STEP ERROR UNDER MISMATCH

Now, we decompose $\tilde{\zeta}_{t,t-1}(x_t,x_{t-1}) = \tilde{\zeta}_{\text{mis}} + \tilde{\zeta}_{\text{van}}$ where

$$\tilde{\zeta}_{\text{mis}} := \sqrt{\alpha_t}\Delta_t(x_t)^{\intercal}(x_{t-1}-m_t(x_t)),$$

$$\tilde{\zeta}_{\text{van}} := (\nabla\log\tilde{q}_{t-1}(m_t(x_t)) - \sqrt{\alpha_t}\nabla\log\tilde{q}_t(x_t))^{\intercal}(x_{t-1}-m_t(x_t)) + \sum_{p=2}^{\infty}T_p(\log\tilde{q}_{t-1},x_{t-1},m_t(x_t)).$$

$$\tag{14}$$

Here $\tilde{\zeta}_{\text{van}}$ is the same tilting factor without score bias (cf. Liang et al. (2024)). Also, define an auxiliary conditional probability $\tilde{P}_{t-1|t}$ such that

$$\tilde{P}_{t-1|t} := \mathcal{N}\left(m_t(x_t), \frac{1-\alpha_t}{\alpha_t}I_d\right),$$

which corresponds to the oracle reverse process that knows the true scores of the perturbed target distributions. Thus, we can decompose the expected value of (13) in the following way:

$$\mathbb{E}_{X_t\sim\tilde{Q}_t}\left(\mathbb{E}_{X_{t-1}\sim\tilde{Q}_{t-1|t}} - \mathbb{E}_{X_{t-1}\sim P_{t-1|t}}\right)[\tilde{\zeta}_{t,t-1}]$$

$$= \underbrace{\mathbb{E}_{X_t \sim \tilde{Q}_t} \left( \mathbb{E}_{X_{t-1} \sim \tilde{Q}_{t-1|t}} - \mathbb{E}_{X_{t-1} \sim \tilde{P}_{-1|t}} \right) [\tilde{\zeta}_{t,t-1}]}_{\mathcal{W}_{\text{oracle, rev-step}}}$$

$$+ \underbrace{\mathbb{E}_{X_t \sim \tilde{Q}_t} \left( \mathbb{E}_{X_{t-1} \sim \tilde{P}_{-1|t}} - \mathbb{E}_{X_{t-1} \sim P_{t-1|t}} \right) [\tilde{\zeta}_{\text{mis}}]}_{\mathcal{W}_{\text{bias, rev-step}}}$$

$$+ \underbrace{\mathbb{E}_{X_t \sim \tilde{Q}_t} \left( \mathbb{E}_{X_{t-1} \sim \tilde{P}_{-1|t}} - \mathbb{E}_{X_{t-1} \sim P_{t-1|t}} \right) [\tilde{\zeta}_{\text{van}}]}_{\mathcal{W}_{\text{vanish, rev-step}}}. \tag{15}$$

### F.3 STEP 3: BOUNDING $\mathcal{W}_{\text{ORACLE, REV-STEP}}$ AND $\mathcal{W}_{\text{BIAS, REV-STEP}}$

Among the terms above, (Liang et al., 2024, Theorem 1) shows that, under Assumption 3 and using the $\alpha_t$ in Definition 1,

$$\mathcal{W}_{\text{oracle, rev-step}} \lesssim \sum_{t=1}^{T} (1 - \alpha_t)^2 \mathbb{E}_{X_t \sim \tilde{Q}_t} \left[ \text{Tr} \left( \nabla^2 \log \tilde{q}_{t-1}(m_t(X_t)) \nabla^2 \log \tilde{q}_t(X_t) \right) \right]. \tag{16}$$

Also, for $\mathcal{W}_{\text{bias, rev-step}}$, since direct calculation yields

$$\mathbb{E}_{X_{t-1} \sim \tilde{P}_{t-1|t}} [\tilde{\zeta}_{\text{mis}}(x_t, X_{t-1})] = \mathbb{E}_{X_{t-1} \sim \tilde{Q}_{t-1|t}} [\tilde{\zeta}_{\text{mis}}(x_t, X_{t-1})] = 0,$$

$$\mathbb{E}_{X_{t-1} \sim P_{t-1|t}} [\tilde{\zeta}_{\text{mis}}(x_t, X_{t-1})] = -(1 - \alpha_t) \|\Delta_t(x_t)\|^2,$$

we have

$$\mathcal{W}_{\text{bias, rev-step}} = \mathbb{E}_{X_t \sim \tilde{Q}_t} \left( \mathbb{E}_{X_{t-1} \sim \tilde{P}_{-1|t}} - \mathbb{E}_{X_{t-1} \sim P_{t-1|t}} \right) [\tilde{\zeta}_{\text{mis}}(X_t, X_{t-1})]$$

$$= (1 - \alpha_t) \mathbb{E}_{X_t \sim \tilde{Q}_t} \|\Delta_t(X_t)\|^2. \tag{17}$$

### F.4 STEP 4: BOUNDING $\mathcal{W}_{\text{VANISH, REV-STEP}}$

Next, for $\mathcal{W}_{\text{vanish, rev-step}}$, we first note that $\tilde{P}_{t-1|t}$ is conditional Gaussian. Thus, under Assumption 3, we are able to exchange the limit (from Taylor series) and the expectation due to Gaussian-like moments (cf. (Liang et al., 2024, Lemma 11)), which gives us

$$\mathbb{E}_{X_{t-1} \sim \tilde{P}_{t-1|t}} [\tilde{\zeta}_{\text{van}}(x_t, X_{t-1})]$$

$$= \frac{1 - \alpha_t}{2\alpha_t} \text{Tr}(\nabla^2 \log \tilde{q}_{t-1}(m_t(x_t))) + \sum_{p=4}^{\infty} \mathbb{E}_{X_{t-1} \sim \tilde{P}_{t-1|t}} [T_p(\log \tilde{q}_{t-1}, X_{t-1}, m_t(x_t))].$$

Here the expected value at $p = 3$ is zero because all odd-order centralized moments of Gaussian vanish.

Now it remains to characterize the expectation of $\tilde{\zeta}_{\text{van}}(x_t, x_{t-1})$ under $P_{t-1|t}$. To this end, we introduce the following notation.

**Definition 2** (Big-O in $\mathcal{L}^p$ space). For a random variable $Z_T$, we say that $Z_T(x) = O_{\mathcal{L}^p(Q)}(1)$ if $(\mathbb{E}_{X \sim Q} |Z_T(X)|^p)^{1/p} = O(1)$ for all $p \geq 1$ as $T \to \infty$. Define $\tilde{O}_{\mathcal{L}^p(Q)}$ likewise.

One property is that if $Z_T(x) = O_{\mathcal{L}^p(Q)}(1)$ then $\mathbb{E}_{X \sim Q} |Z_T(X)| = O(1)$. Another property is that if $Z_1 = O_{\mathcal{L}^p(Q)}(a_T)$ and $Z_2 = O_{\mathcal{L}^p(Q)}(b_T)$ for all $p \geq 1$, applying Cauchy-Schwartz inequality we get, for all $p \geq 1$,

$$(\mathbb{E}_{X \sim Q} |Z_1 Z_2|^p)^{1/p} \leq \left( \mathbb{E}_{X \sim Q} Z_1^{2p} \mathbb{E}_{X \sim Q} Z_2^{2p} \right)^{1/(2p)} = O(a_T b_T),$$

which implies that $O_{\mathcal{L}^p(Q)}(a_T) O_{\mathcal{L}^p(Q)}(b_T) = O_{\mathcal{L}^p(Q)}(a_T b_T)$. Now, with this notation, the first lines of Assumption 3 can be equivalently written as

$$(1 - \alpha_t)^m \left| \partial_{\boldsymbol{a}}^k \log q_t(X_t) \right| = O_{\mathcal{L}^p(\tilde{Q}_t)} \left( (1 - \alpha_t)^m \right), \, \forall p \geq 1,$$

$$(1 - \alpha_t)^m \left| \partial_{\boldsymbol{a}}^k \log q_{t-1}(m_t(X_t)) \right| = O_{\mathcal{L}^p(\tilde{Q}_t)} \left( (1 - \alpha_t)^m \right), \, \forall p \geq 1.$$

Also, Assumption 4 can be equivalently written as

$$(1 - \alpha_t)^m \|\Delta_{t,y}(X_t)\| = O_{\mathcal{L}^p(\tilde{Q}_t)}(\bar{\alpha}_t (1 - \alpha_t)^m), \ \forall p \geq 1.$$

With these notations, the following lemma characterizes the expectation of $\tilde{\zeta}_{\text{van}}(x_t, x_{t-1})$ under $P_{t-1|t}$, which involves non-centralized Gaussian moments.

**Lemma 3.** *As long as $\tilde{q}_{t-1}$ is defined, with the definition of $\tilde{\zeta}_{van}$ in (14), under Assumptions 3 and 4, we have $\forall \ell \geq 1$,*

$$\mathbb{E}_{X_{t-1} \sim P_{t-1|t}}[\tilde{\zeta}_{van}(x_t, X_{t-1})]$$
$$= -\frac{1 - \alpha_t}{\sqrt{\alpha_t}}(\nabla \log \tilde{q}_{t-1}(m_t(x_t)) - \sqrt{\alpha_t}\nabla \log \tilde{q}_t(x_t))^\mathsf{T}\Delta_t(x_t)$$
$$+ \frac{1 - \alpha_t}{2\alpha_t}\text{Tr}(\nabla^2 \log \tilde{q}_{t-1}(m_t)) + \frac{(1 - \alpha_t)^2}{2\alpha_t}\Delta_t(x_t)^\mathsf{T}\nabla^2 \log \tilde{q}_{t-1}(m_t)\Delta_t(x_t)$$
$$- \frac{1}{3!}\left(\frac{(1 - \alpha_t)^2}{\alpha_t^{3/2}}\right)\left(3\sum_{i=1}^d \partial_{iii}^3 \log \tilde{q}_{t-1}(m_t)\Delta_t^i + \sum_{\substack{i,j=1 \\ i \neq j}}^d \partial_{iij}^3 \log \tilde{q}_{t-1}(m_t)\Delta_t^j\right)$$
$$+ \sum_{p=4}^\infty \mathbb{E}_{X_{t-1} \sim P_{t-1|t}}[T_p(\log \tilde{q}_{t-1}, X_{t-1}, m_t(x_t))] + O_{\mathcal{L}^\ell(\tilde{Q}_t)}\left((1 - \alpha_t)^3\right).$$

*Proof.* See Appendix H.3. $\qquad\square$

The following lemma provides the rate of decay of the difference in expectation of all Taylor polynomials with order $p \geq 4$.

**Lemma 4.** *As long as $\tilde{q}_{t-1}$ is defined, under Assumptions 3 and 4, we have, $\forall p \geq 4, \ell \geq 1$,*

$$\left(\mathbb{E}_{X_{t-1} \sim \tilde{P}_{t-1|t}} - \mathbb{E}_{X_{t-1} \sim P_{t-1|t}}\right)[T_p(\log \tilde{q}_{t-1}, X_{t-1}, m_t(X_t))] = O_{\mathcal{L}^\ell(\tilde{Q}_t)}\left((1 - \alpha_t)^3\right).$$

*Proof.* See Appendix H.4. $\qquad\square$

Thus, with the help of Lemmas 3 and 4, we can identify the dominating terms in $\mathcal{W}_{\text{vanish, rev-step}}$ when $1 - \alpha_t$ is small. The dominating term is

$$\mathcal{W}_{\text{vanish, rev-step}} = \mathbb{E}_{X_t \sim \tilde{Q}_t}\left(\mathbb{E}_{X_{t-1} \sim \tilde{P}_{t-1|t}} - \mathbb{E}_{X_{t-1} \sim P_{t-1|t}}\right)[\tilde{\zeta}_{van}(X_t, X_{t-1})]$$
$$= \frac{1 - \alpha_t}{\sqrt{\alpha_t}}\mathbb{E}_{X_t \sim \tilde{Q}_t}[(\nabla \log \tilde{q}_{t-1}(m_t(X_t)) - \sqrt{\alpha_t}\nabla \log \tilde{q}_t(X_t))^\mathsf{T}\Delta_t(X_t)]$$
$$- \frac{(1 - \alpha_t)^2}{2\alpha_t}\mathbb{E}_{X_t \sim \tilde{Q}_t}\left[\Delta_t(X_t)^\mathsf{T}\nabla^2 \log \tilde{q}_{t-1}(m_t(X_t))\Delta_t(X_t)\right]$$
$$+ \frac{1}{3!}\left(\frac{(1 - \alpha_t)^2}{\alpha_t^{3/2}}\right)\mathbb{E}_{X_t \sim \tilde{Q}_t}\left[3\sum_{i=1}^d \partial_{iii}^3 \log \tilde{q}_{t-1}(m_t(X_t))\Delta_t(X_t)^i\right.$$
$$\left. + \sum_{\substack{i,j=1 \\ i \neq j}}^d \partial_{iij}^3 \log \tilde{q}_{t-1}(m_t(X_t))\Delta_t(X_t)^j\right]$$
$$+ O((1 - \alpha_t)^3). \tag{18}$$

Therefore, with the decomposition in (15) in mind, an upper bound on the reverse-step error is achieved by summing up (16), (17) and (18). The proof of Theorem 1 is now complete.

### F.5 COROLLARY 1 AND ITS PROOF

Below we state and prove a corollary of Theorem 1 when $\tilde{q}_0$ does not exist. By (Liang et al., 2024, Lemma 6), $\tilde{q}_1$ always exists, which provides us with the following convergence result with early-stopping.

**Corollary 1.** *Suppose that Assumptions 1 to 4 are satisfied. Then, suppose that the $\alpha_t$ satisfies Definition 1 at $t \geq 2$, the distribution $\widehat{P}_1$ from the discrete-time DDPM under score bias satisfies*

$$\mathrm{KL}(\tilde{Q}_1 \| \widehat{P}_1)$$

$$\lesssim \sum_{t=2}^{T} (1 - \alpha_t) \mathbb{E}_{X_t \sim \tilde{Q}_t} \| \Delta_t(X_t) \|^2$$

$$+ \sum_{t=2}^{T} \frac{1 - \alpha_t}{\sqrt{\alpha_t}} \mathbb{E}_{X_t \sim \tilde{Q}_t} \left[ (\nabla \log \tilde{q}_{t-1}(m_t(X_t)) - \sqrt{\alpha_t} \nabla \log \tilde{q}_t(X_t))^\mathsf{T} \Delta_t(X_t) \right]$$

$$+ \sum_{t=2}^{T} \frac{(1 - \alpha_t)^2}{2\alpha_t} \mathbb{E}_{X_t \sim \tilde{Q}_t} \left[ \mathrm{Tr} \left( \nabla^2 \log \tilde{q}_{t-1}(m_t(X_t)) \left( \nabla^2 \log \tilde{q}_t(X_t) - \Delta_t(X_t) \Delta_t(X_t)^\mathsf{T} \right) \right) \right]$$

$$+ \sum_{t=2}^{T} \frac{(1 - \alpha_t)^2}{3! \alpha_t^{3/2}} \mathbb{E}_{X_t \sim \tilde{Q}_t} \left[ 3 \sum_{i=1}^{d} \partial_{iii}^3 \log \tilde{q}_{t-1}(m_t(X_t)) \Delta_t(X_t)^i + \sum_{\substack{i,j=1 \\ i \neq j}}^{d} \partial_{iij}^3 \log \tilde{q}_{t-1}(m_t(X_t)) \Delta_t(X_t)^j \right]$$

$$+ \max_{t \geq 2} \sqrt{\mathbb{E}_{X_t \sim \tilde{Q}_t} \| \Delta_t(X_t) \|^2} (\log T) \varepsilon + (\log T) \varepsilon^2,$$

*where $\mathrm{W}_2(\tilde{Q}_1, \tilde{Q}_0)^2 \lesssim (1 - \alpha_1) d$.*

*Proof.* The result directly follows with the same arguments as in the proof of Theorem 1. The only difference is the guarantee under the Wasserstein distance, which can be obtained using (Liang et al., 2024, Lemma 12). □

## G PROOF OF THEOREM 2

We first recall some of the properties of the noise schedule in (8). By Lemma 6, the noise schedule in (8) satisfies that $\frac{1 - \alpha_t}{(1 - \bar{\alpha}_{t-1})^p} \lesssim \frac{\log T \log(1/\delta)}{\delta^{p-1} T}$ for all $p \geq 1$ while $\bar{\alpha}_T = o(T^{-1})$, and thus such $\alpha_t$ satisfies Definition 1 when $t \geq 2$. Further, with the $\alpha_t$ in (8), (Liang et al., 2024, Lemmas 15 and 17) show that for any $Q_0$ with finite variance under early-stopping, $\forall p, \ell \geq 1$,

$$\mathbb{E}_{X_t \sim \tilde{Q}_t} |\partial_a^p \log \tilde{q}_t(X_t)|^\ell = O \left( \frac{1}{(1 - \bar{\alpha}_t)^{p\ell/2}} \right),$$

$$\mathbb{E}_{X_t \sim \tilde{Q}_t} |\partial_a^p \log \tilde{q}_{t-1}(m_t(X_t))|^\ell = O \left( \frac{1}{(1 - \bar{\alpha}_{t-1})^{p\ell/2}} \right).$$

Thus, using Lemma 6, Assumption 3 is satisfied (since $\delta$ is constant). In the following, we further verify the last relationship in Assumption 3 holds.

Therefore, since Assumptions 1, 2 and 4 have been satisfied, we can invoke Corollary 1 and get $\mathrm{KL}(\tilde{Q}_1 \| \widehat{P}_1) \lesssim \mathcal{W}_{\text{oracle}} + \mathcal{W}_{\text{bias}} + \mathcal{W}_{\text{vanish}}$. Now, we investigate the dimensional dependence for each term of the upper bound in Corollary 1.

To start, from (Liang et al., 2024, Theorem 3), for any $\tilde{Q}_0$ having finite variance, with the $\alpha_t$ in (8), we have

$$\mathcal{W}_{\text{oracle}} \lesssim \frac{d^2 \log^2(1/\delta)(\log T)^2}{T} + (\log T) \varepsilon^2.$$

Also, since by assumption $\mathbb{E}_{X_t \sim \tilde{Q}_t} \| \Delta_t(X_t) \|^2 \lesssim \frac{\bar{\alpha}_t}{(1 - \bar{\alpha}_t)^r} d^\gamma$, with the $\alpha_t$ in (8), we have from Lemma 7 that when $\delta \ll 1$,

$$\mathcal{W}_{\text{bias}} \lesssim \frac{d^\gamma}{\delta^r} \left( 1 - \frac{2 \log(1/\delta) \log T}{T} \right).$$

Now we investigate each term in $\mathcal{W}_{\text{vanish}}$. The following lemma is useful to determine the rate of difference of the first-order Taylor polynomials.

**Lemma 5.** *When $\mathbb{E}_{X_0 \sim \tilde{Q}_0} \|X_0\|^6 \lesssim d^3$, with the $\alpha_t$ in (8), we have*

$$(1 - \alpha_t)\sqrt{\mathbb{E}_{X_t \sim \tilde{Q}_t} \|\nabla \log \tilde{q}_{t-1}(m_t(X_t)) - \sqrt{\alpha_t}\nabla \log \tilde{q}_t(X_t)\|^2} \lesssim \frac{d^{3/2}(1 - \alpha_t)^2}{(1 - \bar{\alpha}_{t-1})^3}.$$

*As a result, Assumption 3 holds.*

*Proof.* See Appendix H.5. $\qquad\square$

In other words, combining Lemma 5 and Lemma 6, we have

$$(1 - \alpha_t)\sqrt{\mathbb{E}_{X_t \sim \tilde{Q}_t} \|\nabla \log \tilde{q}_{t-1}(m_t(X_t)) - \sqrt{\alpha_t}\nabla \log \tilde{q}_t(X_t)\|^2} = \tilde{O}\left(\frac{1}{T^2}\right). \qquad (19)$$

Now, by Cauchy-Schwartz inequality and Lemma 5,

$$\sum_{t=2}^{T} \frac{1 - \alpha_t}{\sqrt{\alpha_t}} \mathbb{E}_{X_t \sim \tilde{Q}_t} \left[ (\nabla \log \tilde{q}_{t-1}(m_t(X_t)) - \sqrt{\alpha_t}\nabla \log \tilde{q}_t(X_t))^\intercal \Delta_t(X_t) \right]$$

$$\leq \sum_{t=2}^{T} \frac{1 - \alpha_t}{\sqrt{\alpha_t}} \sqrt{\mathbb{E}_{X_t \sim \tilde{Q}_t} \|\Delta_t(X_t)\|^2} \times$$

$$\sqrt{\mathbb{E}_{X_t \sim \tilde{Q}_t} \|\nabla \log \tilde{q}_{t-1}(m_t(X_t)) - \sqrt{\alpha_t}\nabla \log \tilde{q}_t(X_t)\|^2}$$

$$\lesssim \frac{d^{\gamma/2}}{(1 - \bar{\alpha}_t)^{r/2}} \times \frac{d^{3/2}(1 - \alpha_t)^2}{(1 - \bar{\alpha}_{t-1})^3}$$

$$\lesssim \sum_{t=2}^{T} \frac{d^{\frac{3+\gamma}{2}} \log(1/\delta)^2 (\log T)^2}{\delta^{1+r/2} T^2}$$

$$\leq \frac{d^{\frac{3+\gamma}{2}} \log(1/\delta)^2 (\log T)^2}{\delta^{1+r/2} T}.$$

To proceed for higher orders of Taylor polynomials, we first note that from (Liang et al., 2024, Section G.2), the second and third derivatives of $\log \tilde{q}_t$ are

$$\nabla^2 \log \tilde{q}_t(x) = -\frac{1}{1 - \bar{\alpha}_t} I_d + \frac{1}{(1 - \bar{\alpha}_t)^2} \Bigg( \mathbb{E}_{X_0 \sim \tilde{Q}_{0|t}(\cdot|x)} \left[ (x - \sqrt{\bar{\alpha}_t}X_0)(x - \sqrt{\bar{\alpha}_t}X_0)^\intercal \right]$$

$$- \left( \mathbb{E}_{X_0 \sim \tilde{Q}_{0|t}(\cdot|x)} \left[ x - \sqrt{\bar{\alpha}_t}X_0 \right] \right) \left( \mathbb{E}_{X_0 \sim \tilde{Q}_{0|t}(\cdot|x)} \left[ x - \sqrt{\bar{\alpha}_t}X_0 \right] \right)^\intercal \Bigg)$$

$$\partial^3_{ijk} \log \tilde{q}_t(x) = -\int z^i z^j z^k \, d\tilde{Q}_{0|t}(x_0|x)$$

$$+ \sum_{\substack{a_1 = i,j,k \\ a_2 < a_3, \ a_2, a_3 \neq a_1}} \int z^{a_1} \, d\tilde{Q}_{0|t}(x_0|x) \int z^{a_2} z^{a_3} \, d\tilde{Q}_{0|t}(x_0|x)$$

$$- 2 \int z^i \, d\tilde{Q}_{0|t}(x_0|x) \int z^j \, d\tilde{Q}_{0|t}(x_0|x) \int z^k \, d\tilde{Q}_{0|t}(x_0|x)$$

where $z := \frac{x - \sqrt{\bar{\alpha}_t}x_0}{1 - \bar{\alpha}_t}$. Thus, in order to provide an upper bound on the expected norm of the second-order derivative of $\log \tilde{q}_{t-1}(m_t)$, we can calculate

$$\mathbb{E}_{X_t \sim \tilde{Q}_t} \left\| \mathbb{E}_{X_0 \sim \tilde{Q}_{0|t-1}(\cdot|m_t)} \left[ (m_t - \sqrt{\bar{\alpha}_{t-1}}X_0)(m_t - \sqrt{\bar{\alpha}_{t-1}}X_0)^\intercal \right] \right\|_F^2$$

$$\leq \mathbb{E}_{X_t \sim \tilde{Q}_t, X_0 \sim \tilde{Q}_{0|t-1}(\cdot|m_t)} \|(m_t - \sqrt{\bar{\alpha}_{t-1}}X_0)(m_t - \sqrt{\bar{\alpha}_{t-1}}X_0)^\intercal\|_F^2$$

$$= \mathbb{E}_{X_t \sim \tilde{Q}_t, X_0 \sim \tilde{Q}_{0|t-1}(\cdot|m_t)} \|m_t - \sqrt{\bar{\alpha}_{t-1}}X_0\|^4$$

$$\overset{(i)}{\lesssim} d^2(1-\bar{\alpha}_{t-1})^2,$$

and

$$\mathbb{E}_{X_t\sim\tilde{Q}_t}\left\|\left(\mathbb{E}_{X_0\sim\tilde{Q}_{0|t-1}}[m_t-\sqrt{\bar{\alpha}_{t-1}}X_0]\right)\left(\mathbb{E}_{X_0\sim\tilde{Q}_{0|t-1}}[m_t-\sqrt{\bar{\alpha}_{t-1}}X_0]\right)^{\mathsf{T}}\right\|_F^2$$

$$= \mathbb{E}_{X_t\sim\tilde{Q}_t}\left\|\mathbb{E}_{X_0\sim\tilde{Q}_{0|t-1}(\cdot|m_t)}[m_t-\sqrt{\bar{\alpha}_{t-1}}X_0]\right\|^4$$

$$\leq \mathbb{E}_{X_t\sim\tilde{Q}_t,X_0\sim\tilde{Q}_{0|t-1}(\cdot|m_t)}\|m_t-\sqrt{\bar{\alpha}_{t-1}}X_0\|^4$$

$$\overset{(ii)}{\lesssim} d^2(1-\bar{\alpha}_{t-1})^2,$$

where both $(i)$ and $(ii)$ follow from (Liang et al., 2024, Lemma 16). Thus,

$$\mathbb{E}_{X_t\sim\tilde{Q}_t}\left\|\nabla^2\log\tilde{q}_{t-1}(m_t(X_t))\right\|_F^2 \lesssim \frac{1}{(1-\bar{\alpha}_{t-1})^2}d^2.$$

For third-order derivatives, we can similarly use (Liang et al., 2024, Lemma 16) and get (cf. (Liang et al., 2024, Section G.2))

$$\mathbb{E}_{X_t\sim\tilde{Q}_t}\left[\sum_{i=1}^d(\partial_{iii}^3\log\tilde{q}_{t-1}(m_t(X_t)))^2\right] \lesssim \frac{1}{(1-\bar{\alpha}_{t-1})^3}\sum_{i=1}^d\mathbb{E}(Z^i)^6 \lesssim \frac{d}{(1-\bar{\alpha}_{t-1})^3},$$

$$\mathbb{E}_{X_t\sim\tilde{Q}_t}\left[\sum_{i,j=1}^d(\partial_{iij}^3\log\tilde{q}_{t-1}(m_t(X_t)))^2\right] \lesssim \frac{1}{(1-\bar{\alpha}_{t-1})^3}\sum_{i,j=1}^d\left(\mathbb{E}(Z^i)^6\right)^{2/3}\left(\mathbb{E}(Z^j)^6\right)^{1/3}$$

$$\lesssim \frac{d^2}{(1-\bar{\alpha}_{t-1})^3}.$$

Here we denote $Z\sim\mathcal{N}(0,I_d)$, and note that $\mathbb{E}(Z^i)^6,\mathbb{E}(Z^j)^6\lesssim 1$.

Therefore, by Cauchy-Schwartz inequality,

$$\sum_{t=2}^T\frac{(1-\alpha_t)^2}{2\alpha_t}\mathbb{E}_{X_t\sim\tilde{Q}_t}\left[\Delta_t(X_t)^{\mathsf{T}}\nabla^2\log\tilde{q}_{t-1}(m_t(X_t))\Delta_t(X_t)\right]$$

$$\leq \sum_{t=2}^T\frac{(1-\alpha_t)^2}{2\alpha_t}\sqrt{\mathbb{E}_{X_t\sim\tilde{Q}_t}\|\Delta_t(X_t)\|^4}\sqrt{\mathbb{E}_{X_t\sim\tilde{Q}_t}\|\nabla^2\log\tilde{q}_{t-1}(m_t(X_t))\|^2}$$

$$\lesssim \sum_{t=2}^T\frac{(1-\alpha_t)^2}{2\alpha_t}\frac{d^\gamma}{(1-\bar{\alpha}_t)^r}\sqrt{\mathbb{E}_{X_t\sim\tilde{Q}_t}\|\nabla^2\log\tilde{q}_{t-1}(m_t(X_t))\|_F^2}$$

$$\lesssim \sum_{t=2}^T\frac{(1-\alpha_t)^2}{2\alpha_t}\frac{d^\gamma}{(1-\bar{\alpha}_t)^r}\sqrt{\frac{d^2}{(1-\bar{\alpha}_{t-1})^2}}$$

$$\lesssim \frac{d^{1+\gamma}\log(1/\delta)^2(\log T)^2}{\delta^{r-1}T},$$

and

$$\sum_{t=2}^T\frac{(1-\alpha_t)^2}{3!\alpha_t^{3/2}}\mathbb{E}_{X_t\sim\tilde{Q}_t}\left[3\sum_{i=1}^d\partial_{iii}^3\log\tilde{q}_{t-1}(m_t(X_t))\Delta_t(X_t)^i\right]$$

$$\leq \sum_{t=2}^T\frac{3(1-\alpha_t)^2}{3!\alpha_t^{3/2}}\sqrt{\mathbb{E}_{X_t\sim\tilde{Q}_t}\|\Delta_t(X_t)\|^2}\sqrt{\mathbb{E}_{X_t\sim\tilde{Q}_t}\sum_{i=1}^d(\partial_{iii}^3\log\tilde{q}_{t-1}(m_t(X_t)))^2}$$

$$\lesssim \sum_{t=2}^T(1-\alpha_t)^2\frac{d^{\gamma/2}}{(1-\bar{\alpha}_t)^{r/2}}\sqrt{\frac{d}{(1-\bar{\alpha}_{t-1})^3}}$$

$$\lesssim \frac{d^{\frac{1+\gamma}{2}} \log(1/\delta)^2 (\log T)^2}{\delta^{\frac{r-1}{2}} T},$$

and, with $M$ being a matrix such that $M^{ij}(x) := \partial^3_{iij} \log \tilde{q}_{t-1}(m_t(x))$,

$$\sum_{t=2}^{T} \frac{(1-\alpha_t)^2}{3!\alpha_t^{3/2}} \mathbb{E}_{X_t \sim \tilde{Q}_t} \left[ \sum_{\substack{i,j=1 \\ i \neq j}}^{d} \partial^3_{iij} \log \tilde{q}_{t-1}(m_t(X_t)) \Delta_t(X_t)^j \right]$$

$$\leq \sum_{t=2}^{T} \frac{(1-\alpha_t)^2}{3!\alpha_t^{3/2}} \mathbb{E}_{X_t \sim \tilde{Q}_t} \|M(X_t)\Delta_t(X_t)\|_1$$

$$\leq \sum_{t=2}^{T} \frac{(1-\alpha_t)^2}{3!\alpha_t^{3/2}} \sqrt{d} \sqrt{\mathbb{E}_{X_t \sim \tilde{Q}_t} \|\Delta_t(X_t)\|^2} \sqrt{\mathbb{E}_{X_t \sim \tilde{Q}_t} \|M(X_t)\|^2}$$

$$\leq \sum_{t=2}^{T} \frac{(1-\alpha_t)^2}{3!\alpha_t^{3/2}} \sqrt{d} \sqrt{\mathbb{E}_{X_t \sim \tilde{Q}_t} \|\Delta_t(X_t)\|^2} \sqrt{\mathbb{E}_{X_t \sim \tilde{Q}_t} \left[ \sum_{i,j=1}^{d} (\partial^3_{iij} \log \tilde{q}_{t-1}(m_t(X_t)))^2 \right]}$$

$$\lesssim \sum_{t=2}^{T} (1-\alpha_t)^2 \frac{d^{(1+\gamma)/2}}{(1-\bar{\alpha}_t)^{r/2}} \sqrt{\frac{d^2}{(1-\bar{\alpha}_{t-1})^3}}$$

$$\lesssim \frac{d^{\frac{3+\gamma}{2}} \log(1/\delta)^2 (\log T)^2}{\delta^{\frac{r-1}{2}} T},$$

and

$$\max_{t \geq 2} \sqrt{\mathbb{E}_{X_t \sim \tilde{Q}_t} \|\Delta_t(X_t)\|^2} (\log T)\varepsilon \lesssim \frac{d^{\gamma/2}}{\delta^{r/2}} (\log T)\varepsilon.$$

Therefore, combining all the above, we get

$$\mathrm{KL}(\tilde{Q}_1 \| \widehat{P}_1) \lesssim d^\gamma \delta^{-r} \left( 1 - \frac{2\log(1/\delta)\log T}{T} \right)$$

$$+ \max\{d^{(3+\gamma)/2}\delta^{-\frac{r+2}{2}}, d^{1+\gamma}\delta^{-(r-1)}\} \frac{(\log T)^2}{T}$$

$$+ d^{\gamma/2}\delta^{-r/2}(\log T)\varepsilon.$$

# H   AUXILIARY LEMMAS AND PROOFS IN SECTION 3

## H.1   PROOF OF LEMMA 1

We remind readers that throughout this proof $x_t$ is fixed. For brevity write $m_t = m_t(x_t)$, $\mu_t = \mu_t(x_t)$, and $\Delta_t(x_t) = \Delta_t$. Recall that $m_t = x_t/\sqrt{\alpha_t} + (1-\alpha_t)/\sqrt{\alpha_t}\nabla \log \tilde{q}_t(x_t)$. By Bayes' rule, we have

$$\tilde{q}_{t-1|t}(x_{t-1}|x_t)$$

$$= \frac{\tilde{q}_{t|t-1}(x_t|x_{t-1})\tilde{q}_{t-1}(x_{t-1})}{\tilde{q}_t(x_t)}$$

$$\propto \tilde{q}_{t-1}(x_{t-1})\tilde{q}_{t|t-1}(x_t|x_{t-1})$$

$$\overset{(i)}{\propto} \tilde{q}_{t-1}(x_{t-1}) \exp\left( -\frac{\|x_t - \sqrt{\alpha_t}x_{t-1}\|^2}{2(1-\alpha_t)} \right)$$

$$\propto \tilde{q}_{t-1}(x_{t-1})p_{t-1|t}(x_{t-1}|x_t) \exp\left( \frac{\|x_{t-1} - \mu_t\|^2 - \|x_{t-1} - x_t/\sqrt{\alpha_t}\|^2}{2(1-\alpha_t)/\alpha_t} \right)$$

$$= \tilde{q}_{t-1}(x_{t-1})p_{t-1|t}(x_{t-1}|x_t) \exp\left( \frac{\|x_{t-1} - x_t/\sqrt{\alpha_t} + x_t/\sqrt{\alpha_t} - \mu_t\|^2 - \|x_{t-1} - x_t/\sqrt{\alpha_t}\|^2}{2(1-\alpha_t)/\alpha_t} \right)$$

$$\propto \tilde{q}_{t-1}(x_{t-1}) p_{t-1|t}(x_{t-1}|x_t) \exp\left( \frac{(x_{t-1} - x_t/\sqrt{\alpha_t})^\intercal (x_t/\sqrt{\alpha_t} - \mu_t)}{(1-\alpha_t)/\alpha_t} \right)$$

where $(i)$ follows because the forward process is Markov and $\tilde{q}_{t|t-1} = q_{t|t-1}$. Here, the exponent is equal to

$$
\frac{(x_{t-1} - x_t/\sqrt{\alpha_t})^\intercal (x_t/\sqrt{\alpha_t} - \mu_t)}{(1-\alpha_t)/\alpha_t}
$$
$$
= \frac{(x_{t-1} - x_t/\sqrt{\alpha_t})^\intercal (m_t - \mu_t)}{(1-\alpha_t)/\alpha_t} - \frac{(x_{t-1} - x_t/\sqrt{\alpha_t})^\intercal ((1-\alpha_t)/\sqrt{\alpha_t}) \nabla \log \tilde{q}_t(x_t)}{(1-\alpha_t)/\alpha_t}
$$
$$
= \sqrt{\alpha_t} \Delta_t^\intercal (x_{t-1} - x_t/\sqrt{\alpha_t}) - \sqrt{\alpha_t}(x_{t-1} - x_t/\sqrt{\alpha_t})^\intercal \nabla \log \tilde{q}_t(x_t).
$$

Thus,

$$\tilde{q}_{t-1|t}(x_{t-1}|x_t) \propto p_{t-1|t}(x_{t-1}|x_t) \exp\left( \tilde{\zeta}_{t,t-1}(x_t, x_{t-1}) \right)$$

where

$$\tilde{\zeta}_{t,t-1}(x_t, x_{t-1}) = \sqrt{\alpha_t} \Delta_t^\intercal (x_{t-1} - m_t) + \log \tilde{q}_{t-1}(x_{t-1}) - \sqrt{\alpha_t}(x_{t-1} - m_t)^\intercal \nabla \log \tilde{q}_t(x_t).$$

Finally, since all partial derivatives of $\tilde{q}_{t-1}$ exists for any $t \geq 2$ (See (Liang et al., 2024, Lemma 6)), the Taylor expansion of $\log \tilde{q}_{t-1}$ around $m_t$ gives the desirable result.

## H.2 PROOF OF LEMMA 2

For each $t = 1, \ldots, T$,

$$
\log \frac{p_{t-1|t}(x_{t-1}|x_t)}{\widehat{p}_{t-1|t}(x_{t-1}|x_t)} = \frac{\alpha_t}{2(1-\alpha_t)} \left( \|x_{t-1} - \widehat{\mu}_t(x_t)\|^2 - \|x_{t-1} - \mu_t(x_t)\|^2 \right)
$$
$$
= \frac{\alpha_t}{(1-\alpha_t)} (x_{t-1} - \mu_t(x_t))^\intercal (\mu_t(x_t) - \widehat{\mu}_t(x_t)) + \frac{\alpha_t}{2(1-\alpha_t)} \|\mu_t(x_t) - \widehat{\mu}_t(x_t)\|^2 .
$$

For the first term above,

$$
\mathbb{E}_{X_t, X_{t-1} \sim \tilde{Q}_{t,t-1}} \left[ (X_{t-1} - \mu_t(X_t))^\intercal (\mu_t(X_t) - \widehat{\mu}_t(X_t)) \right]
$$
$$
= \mathbb{E}_{X_t \sim \tilde{Q}_t} \left[ (m_t(X_t) - \mu_t(X_t))^\intercal (\mu_t(X_t) - \widehat{\mu}_t(X_t)) \right]
$$
$$
= \frac{1 - \alpha_t}{\sqrt{\alpha_t}} \mathbb{E}_{X_t \sim \tilde{Q}_t} \left[ \Delta_t(X_t)^\intercal (\mu_t(X_t) - \widehat{\mu}_t(X_t)) \right]
$$
$$
\leq \frac{1 - \alpha_t}{\sqrt{\alpha_t}} \sqrt{ \mathbb{E}_{X_t \sim \tilde{Q}_t} \|\Delta_t(X_t)\|^2 \, \mathbb{E}_{X_t \sim \tilde{Q}_t} \|\mu_t(X_t) - \widehat{\mu}_t(X_t)\|^2 }.
$$

Here we recall the definition of $\Delta_t$ from (2) where $m_t(x) - \mu_t(x) = \frac{1-\alpha_t}{\sqrt{\alpha_t}} \Delta_t(x)$. Thus,

$$
\sum_{t=1}^T \mathbb{E}_{X_t, X_{t-1} \sim \tilde{Q}_{t,t-1}} \left[ \log \frac{p_{t-1|t}(X_{t-1}|X_t)}{\widehat{p}_{t-1|t}(X_{t-1}|X_t)} \right]
$$
$$
\lesssim \sum_{t=1}^T \left( \sqrt{\alpha_t} \sqrt{ \mathbb{E}_{X_t \sim \tilde{Q}_t} \|\Delta_t(X_t)\|^2 \, \mathbb{E}_{X_t \sim \tilde{Q}_t} \|\mu_t(X_t) - \widehat{\mu}_t(X_t)\|^2 } + \frac{\alpha_t}{1 - \alpha_t} \mathbb{E}_{X_t \sim Q_t} \|\mu_t(X_t) - \widehat{\mu}_t(X_t)\|^2 \right)
$$
$$
= \sum_{t=1}^T (1-\alpha_t) \sqrt{ \mathbb{E}_{X_t \sim \tilde{Q}_t} \|\Delta_t(X_t)\|^2 } \times \sqrt{ \frac{\alpha_t}{(1-\alpha_t)^2} \mathbb{E}_{X_t \sim \tilde{Q}_t} \|\mu_t(X_t) - \widehat{\mu}_t(X_t)\|^2 }
$$
$$
+ \sum_{t=1}^T (1-\alpha_t) \frac{\alpha_t}{(1-\alpha_t)^2} \mathbb{E}_{X_t \sim Q_t} \|\mu_t(X_t) - \widehat{\mu}_t(X_t)\|^2
$$
$$
\overset{(i)}{\lesssim} \frac{\log T}{T} \sum_{t=1}^T \sqrt{ \mathbb{E}_{X_t \sim \tilde{Q}_t} \|\Delta_t(X_t)\|^2 } \times \sqrt{ \frac{\alpha_t}{(1-\alpha_t)^2} \mathbb{E}_{X_t \sim \tilde{Q}_t} \|\mu_t(X_t) - \widehat{\mu}_t(X_t)\|^2 }
$$
$$
+ \frac{\log T}{T} \sum_{t=1}^T \frac{\alpha_t}{(1-\alpha_t)^2} \mathbb{E}_{X_t \sim Q_t} \|\mu_t(X_t) - \widehat{\mu}_t(X_t)\|^2
$$

$$\leq \max_{t \geq 1} \sqrt{\mathbb{E}_{X_t \sim \tilde{Q}_t} \|\Delta_t(X_t)\|^2} \frac{\log T}{T} \sum_{t=1}^{T} \sqrt{\frac{\alpha_t}{(1-\alpha_t)^2} \mathbb{E}_{X_t \sim \tilde{Q}_t} \|\mu_t(X_t) - \hat{\mu}_t(X_t)\|^2}$$

$$+ \frac{\log T}{T} \sum_{t=1}^{T} \frac{\alpha_t}{(1-\alpha_t)^2} \mathbb{E}_{X_t \sim Q_t} \|\mu_t(X_t) - \hat{\mu}_t(X_t)\|^2$$

$$\overset{(ii)}{\lesssim} \max_{t \geq 1} \sqrt{\mathbb{E}_{X_t \sim \tilde{Q}_t} \|\Delta_t(X_t)\|^2} (\log T) \sqrt{\frac{1}{T} \sum_{t=1}^{T} \frac{\alpha_t}{(1-\alpha_t)^2} \mathbb{E}_{X_t \sim \tilde{Q}_t} \|\mu_t(X_t) - \hat{\mu}_t(X_t)\|^2}$$

$$+ \frac{\log T}{T} \sum_{t=1}^{T} \frac{\alpha_t}{(1-\alpha_t)^2} \mathbb{E}_{X_t \sim Q_t} \|\mu_t(X_t) - \hat{\mu}_t(X_t)\|^2$$

$$\overset{(iii)}{\lesssim} \max_{t \geq 1} \sqrt{\mathbb{E}_{X_t \sim \tilde{Q}_t} \|\Delta_t(X_t)\|^2} (\log T)\varepsilon + (\log T)\varepsilon^2$$

where $(i)$ follows from Definition 1, $(ii)$ follows from the fact that for any non-negative sequence $a_t$, $\frac{1}{T} \sum_{t=1}^{T} \sqrt{a_t} \leq \sqrt{\frac{1}{T} \sum_{t=1}^{T} a_t}$ by Jensen's inequality, and $(iii)$ follows from Assumption 2. The proof is complete.

### H.3 PROOF OF LEMMA 3

Recall that $P_{t-1|t} = \mathcal{N}(\mu_t, \frac{1-\alpha_t}{\alpha_t} I_d)$, and thus $\mathbb{E}_{X_{t-1} \sim P_{t-1|t}}[X_{t-1} - m_t(x_t)] = -\frac{1-\alpha_t}{\sqrt{\alpha_t}} \Delta_t(x_t)$. Note that we can change the limit and the expectation under Assumption 3. Now, we can calculate that

$$\mathbb{E}_{X_{t-1} \sim P_{t-1|t}}[\tilde{\zeta}_{\text{van}}(x_t, X_{t-1})]$$
$$= -\frac{1-\alpha_t}{\sqrt{\alpha_t}} (\nabla \log \tilde{q}_{t-1}(m_t(x_t)) - \sqrt{\alpha_t} \nabla \log \tilde{q}_t(x_t))^{\mathsf{T}} \Delta_t(x_t)$$
$$+ \sum_{p=2}^{\infty} \mathbb{E}_{X_{t-1} \sim P_{t-1|t}} [T_p(\log \tilde{q}_{t-1}, X_{t-1}, m_t(x_t))] .$$

Below we write $m_t = m_t(x_t)$. Since $T_2$ is in quadratic form, the expected value under $P_{t-1|t}$ for this term is

$$\mathbb{E}_{X_{t-1} \sim P_{t-1|t}} [T_2(\log \tilde{q}_{t-1}, X_{t-1}, m_t)]$$
$$= \frac{1-\alpha_t}{2\alpha_t} \text{Tr}(\nabla^2 \log \tilde{q}_{t-1}(m_t)) + \frac{(1-\alpha_t)^2}{2\alpha_t} \Delta_t(x_t)^{\mathsf{T}} \nabla^2 \log \tilde{q}_{t-1}(m_t) \Delta_t(x_t).$$

Recall the formula for Gaussian non-centralized third moment. If $Z \sim \mathcal{N}(\mu, \sigma^2)$, then $\mathbb{E}[Z^2] = \mu^2 + \sigma^2$ and $\mathbb{E}[Z^3] = \mu^3 + 3\mu\sigma^2$. Thus, the expected value under $P_{t-1|t}$ for $T_3$ is

$$\mathbb{E}_{X_{t-1} \sim P_{t-1|t}} [T_3(\log \tilde{q}_{t-1}, X_{t-1}, m_t)]$$

$$= \frac{1}{3!} \sum_{i=1}^{d} \partial_{iii}^3 \log \tilde{q}_{t-1}(m_t) \mathbb{E}_{X_{t-1} \sim P_{t-1|t}} (X_{t-1}^i - m_t^i)^3$$

$$+ \frac{1}{3!} \sum_{\substack{i,j=1 \\ i \neq j}}^{d} \partial_{iij}^3 \log \tilde{q}_{t-1}(m_t) \mathbb{E}_{X_{t-1} \sim P_{t-1|t}} (X_{t-1}^i - m_t^i)^2 (X_{t-1}^j - m_t^j)$$

$$+ \frac{1}{3!} \sum_{\substack{i,j,k=1 \\ i,j,k \text{ all differ}}}^{d} \partial_{ijk}^3 \log \tilde{q}_{t-1}(m_t) \mathbb{E}_{X_{t-1} \sim P_{t-1|t}} (X_{t-1}^i - m_t^i)(X_{t-1}^j - m_t^j)(X_{t-1}^k - m_t^k)$$

$$= \frac{1}{3!} \sum_{i=1}^{d} \partial_{iii}^3 \log \tilde{q}_{t-1}(m_t) \left( \left( -\frac{1-\alpha_t}{\sqrt{\alpha_t}} \Delta_t^i \right)^3 + 3 \left( -\frac{1-\alpha_t}{\sqrt{\alpha_t}} \Delta_t^i \right) \left( \frac{1-\alpha_t}{\alpha_t} \right) \right)$$

$$+ \frac{1}{3!} \sum_{\substack{i,j=1 \\ i \neq j}}^{d} \partial_{iij}^3 \log \tilde{q}_{t-1}(m_t) \left( \left( -\frac{1-\alpha_t}{\sqrt{\alpha_t}} \Delta_t^i \right)^2 + \left( \frac{1-\alpha_t}{\alpha_t} \right) \right) \left( -\frac{1-\alpha_t}{\sqrt{\alpha_t}} \Delta_t^j \right)$$

$$+ \frac{1}{3!} \sum_{\substack{i,j,k=1 \\ i,j,k \text{ all differ}}}^{d} \partial_{ijk}^3 \log \tilde{q}_{t-1}(m_t) \left( -\frac{1-\alpha_t}{\sqrt{\alpha_t}} \Delta_t^i \right) \left( -\frac{1-\alpha_t}{\sqrt{\alpha_t}} \Delta_t^j \right) \left( -\frac{1-\alpha_t}{\sqrt{\alpha_t}} \Delta_t^k \right)$$

$$= \frac{1}{3!} \left( -\frac{(1-\alpha_t)^2}{\alpha_t^{3/2}} \right) \left( 3 \sum_{i=1}^{d} \partial_{iii}^3 \log \tilde{q}_{t-1}(m_t) \Delta_t^i + \sum_{\substack{i,j=1 \\ i \neq j}}^{d} \partial_{iij}^3 \log \tilde{q}_{t-1}(m_t) \Delta_t^j \right)$$

$$+ O_{\mathcal{L}^\ell(\tilde{Q}_t)} \left( (1-\alpha_t)^3 \right), \ \forall \ell \geq 1.$$

Here the last line follows because $(1-\alpha_t)^3 \left| \partial_{ijk}^3 \log \tilde{q}_{t-1}(m_t) \right| = O_{\mathcal{L}^\ell(\tilde{Q}_t)} \left( (1-\alpha_t)^3 \right)$ under Assumption 3 and $(1-\alpha_t)^3 \|\Delta_t\| = O_{\mathcal{L}^\ell(\tilde{Q}_t)}((1-\alpha_t)^3)$ under Assumption 4, both for all $\ell \geq 1$. The proof is now complete.

### H.4 PROOF OF LEMMA 4

Fix $x_t \in \mathbb{R}^d$. For brevity write $m_t = m_t(x_t)$, $\mu_t = \mu_t(x_t)$, and $\Delta_t = \Delta_t(x_t)$. Recall that

$$T_p(\log \tilde{q}_{t-1}, x_{t-1}, m_t) = \frac{1}{p!} \sum_{\gamma \in \mathbb{N}^d : \sum_i \gamma^i = p} \partial_{\boldsymbol{a}}^p \log \tilde{q}_{t-1}(m_t) \prod_{i=1}^{d} (x_{t-1}^i - m_t^i)^{\gamma^i}$$

where $\boldsymbol{a} \in [d]^p$ are the indices of differentiation in which the multiplicity of $i$ is $\gamma^i$. First, for the expectation under $\tilde{P}_{t-1|t}$ (i.e., Gaussian centralized moments),

$$\mathbb{E}_{X_{t-1} \sim \tilde{P}_{t-1|t}} \left[ \prod_{i=1}^{d} (X_{t-1}^i - m_t^i)^{\gamma^i} \right] = \prod_{i=1}^{d} \mathbb{E}_{X_{t-1} \sim \tilde{P}_{t-1|t}} \left[ (X_{t-1}^i - m_t^i)^{\gamma^i} \right]$$

$$= \prod_{i=1}^{d} \left( \frac{1-\alpha_t}{\alpha_t} \right)^{\gamma^i/2} (\gamma^i - 1)!! \mathbb{1}\{\gamma^i \text{ is even}\}$$

$$= \left( \frac{1-\alpha_t}{\alpha_t} \right)^{p/2} \prod_{i=1}^{d} (\gamma^i - 1)!! \mathbb{1}\{\gamma^i \text{ is even}\},$$

where we use the convention that $(-1)!! = 1$. Next, for the expectation under $P_{t-1|t}$ (i.e., Gaussian non-centralized moments),

$$\mathbb{E}_{X_{t-1} \sim P_{t-1|t}} \left[ \prod_{i=1}^{d} (X_{t-1}^i - m_t^i)^{\gamma^i} \right]$$

$$= \prod_{i=1}^{d} \mathbb{E}_{X_{t-1} \sim P_{t-1|t}} \left[ (X_{t-1}^i - \mu_t^i - \frac{1-\alpha_t}{\sqrt{\alpha_t}} \Delta_t^i)^{\gamma^i} \right]$$

$$= \prod_{i=1}^{d} \sum_{\substack{\ell=0 \\ \ell \text{ even}}}^{\gamma^i} \binom{\gamma^i}{\ell} \left( -\frac{1-\alpha_t}{\sqrt{\alpha_t}} \Delta_t^i \right)^{\gamma^i - \ell} \left( \frac{1-\alpha_t}{\alpha_t} \right)^{\ell/2} (\ell-1)!!$$

To investigate their difference, we divide into the following few cases. Note that under Assumption 4, $(1-\alpha_t)^m \|\Delta_t(x_t)\| = O_{\mathcal{L}^\ell(\tilde{Q}_t)}((1-\alpha_t)^m)$ for any $m \geq 1/2$ and $\ell \geq 1$.

1. Case 1: $p$ is even and all elements of $\gamma^i$ are even. Then,

$$\mathbb{E}_{X_{t-1} \sim P_{t-1|t}} \left[ \prod_{i=1}^{d} (X_{t-1}^i - m_t^i)^{\gamma^i} \right]$$

$$= \prod_{i=1}^{d} \left( \left( \frac{1 - \alpha_t}{\alpha_t} \right)^{\gamma^i/2} (\gamma^i - 1)!! + O_{\mathcal{L}^\ell(\tilde{Q}_t)} \left( (1 - \alpha_t)^{\gamma^i/2+1} \right) \right)$$

$$= \left( \frac{1 - \alpha_t}{\alpha_t} \right)^{p/2} \prod_{i=1}^{d} (\gamma^i - 1)!! + O_{\mathcal{L}^\ell(\tilde{Q}_t)} \left( (1 - \alpha_t)^{p/2+1} \right)$$

2. Case 2: $p$ is even and $\exists i^*$ such that $\gamma^{i^*}$ is odd. Since $\sum_i \gamma^i = p$, there exists $j^*$ such that $\gamma^{j^*}$ is also odd. Then,

$$\mathbb{E}_{X_{t-1} \sim P_{t-1|t}} \left[ (X_{t-1}^{i^*} - m_t^{i^*})^{\gamma^{i^*}} \right] = O_{\mathcal{L}^\ell(\tilde{Q}_t)} \left( (1 - \alpha_t)^{(\gamma^{i^*}+1)/2} \right),$$

$$\mathbb{E}_{X_{t-1} \sim P_{t-1|t}} \left[ (X_{t-1}^{j^*} - m_t^{j^*})^{\gamma^{j^*}} \right] = O_{\mathcal{L}^\ell(\tilde{Q}_t)} \left( (1 - \alpha_t)^{(\gamma^{j^*}+1)/2} \right),$$

which implies that

$$\mathbb{E}_{X_{t-1} \sim P_{t-1|t}} \left[ \prod_{i=1}^{d} (X_{t-1}^i - m_t^i)^{\gamma^i} \right] = O_{\mathcal{L}^\ell(\tilde{Q}_t)} \left( (1 - \alpha_t)^{p/2+1} \right)$$

3. Case 3: $p$ is odd and $\exists i^*$ such that $\gamma^{i^*}$ is odd. Then,

$$\mathbb{E}_{X_{t-1} \sim P_{t-1|t}} \left[ (X_{t-1}^{i^*} - m_t^{i^*})^{\gamma^{i^*}} \right] = O_{\mathcal{L}^\ell(\tilde{Q}_t)} \left( (1 - \alpha_t)^{(\gamma^{i^*}+1)/2} \right),$$

which implies that

$$\mathbb{E}_{X_{t-1} \sim P_{t-1|t}} \left[ \prod_{i=1}^{d} (X_{t-1}^i - m_t^i)^{\gamma^i} \right] = O_{\mathcal{L}^\ell(\tilde{Q}_t)} \left( (1 - \alpha_t)^{(p+1)/2} \right)$$

Combining these cases, we get

$$\left( \mathbb{E}_{X_{t-1} \sim \tilde{P}_{t-1|t}} - \mathbb{E}_{X_{t-1} \sim P_{t-1|t}} \right) \left[ \prod_{i=1}^{d} (X_{t-1}^i - m_t^i)^{\gamma^i} \right]$$

$$= \begin{cases} O_{\mathcal{L}^\ell(\tilde{Q}_t)} \left( (1 - \alpha_t)^{\frac{p}{2}+1} \right), & \forall p \geq 4 \text{ even} \\ O_{\mathcal{L}^\ell(\tilde{Q}_t)} \left( (1 - \alpha_t)^{\frac{p+1}{2}} \right), & \forall p \geq 4 \text{ odd} \end{cases}$$

The proof is complete by noting that the rate does not change when we take the expectation over $\tilde{Q}_t$ under Assumptions 3 and 4.

## H.5 PROOF OF LEMMA 5

Note that $\tilde{q}_{t|0}(x|x_0) = q_{t|0}(x|x_0)$ is the p.d.f. of $\mathcal{N}(\sqrt{\bar{\alpha}_t} x_0, (1 - \bar{\alpha}_t) I_d)$. Thus, the gradient of $\log \tilde{q}_t(x)$ equals

$$\nabla \log \tilde{q}_t(x) = \frac{\int_{x_0 \in \mathbb{R}^d} \nabla \tilde{q}_{t|0}(x|x_0) d\tilde{Q}_0(x_0)}{\tilde{q}_t(x)} = -\frac{1}{1 - \bar{\alpha}_t} \int_{x_0 \in \mathbb{R}^d} (x - \sqrt{\bar{\alpha}_t} x_0) d\tilde{Q}_{0|t}(x_0|x). \quad (20)$$

Thus,

$$\nabla \log \tilde{q}_{t-1}(m_t) - \sqrt{\alpha_t} \nabla \log \tilde{q}_t(x_t)$$

$$= -\frac{1}{1 - \bar{\alpha}_{t-1}} \int_{x_0 \in \mathbb{R}^d} (m_t - \sqrt{\bar{\alpha}_{t-1}} x_0) d\tilde{Q}_{0|t-1}(x_0|m_t) + \frac{\sqrt{\alpha_t}}{1 - \bar{\alpha}_t} \int_{x_0 \in \mathbb{R}^d} (x_t - \sqrt{\bar{\alpha}_t} x_0) d\tilde{Q}_{0|t}(x_0|x_t)$$

$$= -\frac{1}{1 - \bar{\alpha}_t} \left( \left( \frac{1 - \bar{\alpha}_t}{1 - \bar{\alpha}_{t-1}} - 1 \right) \int_{x_0 \in \mathbb{R}^d} (m_t - \sqrt{\bar{\alpha}_{t-1}} x_0) d\tilde{Q}_{0|t-1}(x_0|m_t) \right.$$

$$\left. + \int_{x_0 \in \mathbb{R}^d} (m_t - \sqrt{\bar{\alpha}_{t-1}} x_0) d\tilde{Q}_{0|t-1}(x_0|m_t) - \sqrt{\alpha_t} \int_{x_0 \in \mathbb{R}^d} (x_t - \sqrt{\bar{\alpha}_t} x_0) d\tilde{Q}_{0|t}(x_0|x_t) \right)$$

$$\overset{(i)}{=} \frac{1}{1-\bar{\alpha}_t}\left( \left(1-\bar{\alpha}_t - (1-\bar{\alpha}_{t-1})\right)\nabla\log\tilde{q}_{t-1}(m_t)\right)$$

$$- \frac{1}{1-\bar{\alpha}_t}\left( \int_{x_0\in\mathbb{R}^d}(m_t - \sqrt{\bar{\alpha}_{t-1}}x_0)\mathrm{d}\tilde{Q}_{0|t-1}(x_0|m_t) - \sqrt{\alpha_t}\int_{x_0\in\mathbb{R}^d}(x_t - \sqrt{\bar{\alpha}_t}x_0)\mathrm{d}\tilde{Q}_{0|t}(x_0|x_t)\right)$$

$$= \underbrace{\frac{\bar{\alpha}_{t-1}(1-\alpha_t)}{1-\bar{\alpha}_t}\nabla\log\tilde{q}_{t-1}(m_t)}_{\text{term 1}} - \underbrace{\frac{1}{1-\bar{\alpha}_t}(m_t - \sqrt{\alpha_t}x_t)}_{\text{term 2}} + \underbrace{\frac{\sqrt{\bar{\alpha}_{t-1}}(1-\alpha_t)}{1-\bar{\alpha}_t}\int_{x_0\in\mathbb{R}^d}x_0\mathrm{d}\tilde{Q}_{0|t}(x_0|x_t)}_{\text{term 3}}$$

$$+ \underbrace{\frac{\sqrt{\bar{\alpha}_{t-1}}}{1-\bar{\alpha}_t}\left(\int_{x_0\in\mathbb{R}^d}x_0\mathrm{d}\tilde{Q}_{0|t-1}(x_0|m_t) - \int_{x_0\in\mathbb{R}^d}x_0\mathrm{d}\tilde{Q}_{0|t}(x_0|x_t)\right)}_{\text{term 4}} \tag{21}$$

where $(i)$ follows from Tweedie's formula. Among the four terms in (21), the first term satisfies that

$$\mathbb{E}_{X_t\sim\tilde{Q}_t}\left\|\frac{\bar{\alpha}_{t-1}(1-\alpha_t)}{1-\bar{\alpha}_t}\nabla\log\tilde{q}_{t-1}(m_t)\right\|^2 \lesssim \frac{d(1-\alpha_t)^2}{(1-\bar{\alpha}_t)^2(1-\bar{\alpha}_{t-1})}$$

by (Liang et al., 2024, Lemma 17). In the second term in (21), by Tweedie's formula,

$$m_t - \sqrt{\alpha_t}x_t = \frac{x_t}{\sqrt{\alpha_t}} + \frac{1-\alpha_t}{\sqrt{\alpha_t}}\nabla\log\tilde{q}_t(x_t) - \sqrt{\alpha_t}x_t$$

$$= \frac{1-\alpha_t}{\sqrt{\alpha_t}}(x_t + \nabla\log\tilde{q}_t(x_t)).$$

Thus, by (Liang et al., 2024, Lemma 15) and Assumption 1, the second term satisfies that

$$\mathbb{E}_{X_t\sim\tilde{Q}_t}\left\|\frac{1}{1-\bar{\alpha}_t}(m_t - \sqrt{\alpha_t}x_t)\right\|^2 \lesssim \frac{d(1-\alpha_t)^2}{(1-\bar{\alpha}_t)^3}.$$

The third term in (21) satisfies that

$$\mathbb{E}_{X_t\sim\tilde{Q}_t}\left\|\frac{\sqrt{\bar{\alpha}_{t-1}}(1-\alpha_t)}{1-\bar{\alpha}_t}\int_{x_0\in\mathbb{R}^d}x_0\mathrm{d}\tilde{Q}_{0|t}(x_0|x_t)\right\|^2 \lesssim \frac{d(1-\alpha_t)^2}{(1-\bar{\alpha}_t)^2}$$

by Jensen's inequality and Assumption 1.

To deal with the last term in (21), note that

$$\mathrm{d}\tilde{Q}_{0|t-1}(x_0|m_t) = \frac{\tilde{q}_{t-1|0}(m_t|x_0)}{\tilde{q}_{t-1}(m_t)}\mathrm{d}\tilde{Q}_0(x_0) = \frac{\tilde{q}_{t-1|0}(m_t|x_0)}{\int_{y\in\mathbb{R}^d}\tilde{q}_{t-1|0}(m_t|y)\mathrm{d}\tilde{Q}_0(y)}\mathrm{d}\tilde{Q}_0(x_0),$$

$$\mathrm{d}\tilde{Q}_{0|t}(x_0|x_t) = \frac{\tilde{q}_{t|0}(x_t|x_0)}{\tilde{q}_t(x_t)}\mathrm{d}\tilde{Q}_0(x_0) = \frac{\tilde{q}_{t|0}(x_t|x_0)}{\int_{y\in\mathbb{R}^d}\tilde{q}_{t|0}(x_t|y)\mathrm{d}\tilde{Q}_0(y)}\mathrm{d}\tilde{Q}_0(x_0).$$

Thus, the last term in (21) is equal to

$$\frac{\sqrt{\bar{\alpha}_{t-1}}}{1-\bar{\alpha}_t}\left(\int_{x_0\in\mathbb{R}^d}x_0\mathrm{d}\tilde{Q}_{0|t-1}(x_0|m_t) - \int_{x_0\in\mathbb{R}^d}x_0\mathrm{d}\tilde{Q}_{0|t}(x_0|x_t)\right)$$

$$= \frac{\sqrt{\bar{\alpha}_{t-1}}}{1-\bar{\alpha}_t}\cdot\frac{1}{\tilde{q}_{t-1}(m_t)\tilde{q}_t(x_t)}\left(\int_{x,y\in\mathbb{R}^d}x(\tilde{q}_{t-1|0}(m_t|x)\tilde{q}_{t|0}(x_t|y) - \tilde{q}_{t|0}(x_t|x)\tilde{q}_{t-1|0}(m_t|y))\mathrm{d}\tilde{Q}_0(x)\mathrm{d}\tilde{Q}_0(y)\right)$$

where

$$\tilde{q}_{t-1|0}(m_t|x)\tilde{q}_{t|0}(x_t|y) - \tilde{q}_{t|0}(x_t|x)\tilde{q}_{t-1|0}(m_t|y)$$

$$= \tilde{q}_{t|0}(x_t|x)\tilde{q}_{t-1|0}(m_t|y)\left(\frac{\tilde{q}_{t-1|0}(m_t|x)\tilde{q}_{t|0}(x_t|y)}{\tilde{q}_{t|0}(x_t|x)\tilde{q}_{t-1|0}(m_t|y)} - 1\right)$$

$$= \tilde{q}_{t|0}(x_t|x)\tilde{q}_{t-1|0}(m_t|y)\times$$

$$\left(\exp\left(-\frac{\|m_t - \sqrt{\bar{\alpha}_{t-1}}x\|^2}{2(1-\bar{\alpha}_{t-1})} - \frac{\|x_t - \sqrt{\bar{\alpha}_t}y\|^2}{2(1-\bar{\alpha}_t)} + \frac{\|m_t - \sqrt{\bar{\alpha}_{t-1}}y\|^2}{2(1-\bar{\alpha}_{t-1})} + \frac{\|x_t - \sqrt{\bar{\alpha}_t}x\|^2}{2(1-\bar{\alpha}_t)}\right) - 1\right)$$

$$=: \tilde{q}_{t|0}(x_t|x)\tilde{q}_{t-1|0}(m_t|y)(e^\Delta - 1)$$

in which we have defined the exponent as $\Delta$. Now,

$$
\Delta = -\frac{\|m_t - \sqrt{\bar{\alpha}_{t-1}}x\|^2}{2(1 - \bar{\alpha}_{t-1})} - \frac{\|x_t - \sqrt{\bar{\alpha}_t}y\|^2}{2(1 - \bar{\alpha}_t)} + \frac{\|m_t - \sqrt{\bar{\alpha}_{t-1}}y\|^2}{2(1 - \bar{\alpha}_{t-1})} + \frac{\|x_t - \sqrt{\bar{\alpha}_t}x\|^2}{2(1 - \bar{\alpha}_t)}
$$

$$
= \frac{\sqrt{\bar{\alpha}_{t-1}}(x - y)^\mathsf{T}m_t + \bar{\alpha}_{t-1}\|y\|^2 - \bar{\alpha}_{t-1}\|x\|^2}{2(1 - \bar{\alpha}_{t-1})} - \frac{\sqrt{\bar{\alpha}_t}(x - y)^\mathsf{T}x_t + \bar{\alpha}_t\|y\|^2 - \bar{\alpha}_t\|x\|^2}{2(1 - \bar{\alpha}_t)}
$$

$$
= \frac{1}{2}\left(\frac{\sqrt{\bar{\alpha}_t}(1 - \alpha_t)}{\alpha_t(1 - \bar{\alpha}_{t-1})(1 - \bar{\alpha}_t)}x_t + \frac{\sqrt{\bar{\alpha}_{t-1}}(1 - \alpha_t)}{\sqrt{\bar{\alpha}_t}(1 - \bar{\alpha}_{t-1})}\nabla\log\tilde{q}_t(x_t)\right)^\mathsf{T}(x - y)
$$

$$
+ \frac{\bar{\alpha}_{t-1}(1 - \alpha_t)}{(1 - \bar{\alpha}_{t-1})(1 - \bar{\alpha}_t)}(\|y\|^2 - \|x\|^2).
$$

Now, with the $\alpha_t$ defined in (8), following from Lemma 6,

$$
\frac{\sqrt{\bar{\alpha}_t}(1 - \alpha_t)}{\alpha_t(1 - \bar{\alpha}_{t-1})(1 - \bar{\alpha}_t)} = O\left(\frac{1 - \alpha_t}{(1 - \bar{\alpha}_{t-1})^2}\right) = O\left(\frac{\log T}{T}\right),
$$

$$
\frac{1 - \alpha_t}{1 - \bar{\alpha}_{t-1}} = O\left(\frac{\log T}{T}\right),
$$

$$
\frac{\bar{\alpha}_{t-1}(1 - \alpha_t)}{(1 - \bar{\alpha}_{t-1})(1 - \bar{\alpha}_t)} = O\left(\frac{1 - \alpha_t}{(1 - \bar{\alpha}_{t-1})^2}\right) = O\left(\frac{\log T}{T}\right).
$$

Thus, for fixed $x, y, x_t$, $\Delta \to 0$ as $T \to \infty$, and thus when $T$ becomes large,

$$
e^\Delta - 1 = \Delta + O(\Delta^2) \lesssim |\Delta|, \quad \forall x_t \in \mathbb{R}^d.
$$

Also, since $\tilde{q}_{t|0}(x_t|x)$ and $\tilde{q}_{t-1|0}(m_t|y)$ decay exponentially in terms of $x$ and $y$ (for any fixed $x_t$), we have

$$
\int \tilde{q}_{t|0}(x_t|x)\text{poly}(x)\mathrm{d}\tilde{Q}_0(x) < \infty,
$$

$$
\int \tilde{q}_{t-1|0}(m_t|y)\text{poly}(y)\mathrm{d}\tilde{Q}_0(y) < \infty.
$$

Thus, the limit and the integral can be exchanged due to Dominated Convergence Theorem. Thus, the fourth term in (21) gives us

$$
\frac{\sqrt{\bar{\alpha}_{t-1}}}{1 - \bar{\alpha}_t}\left(\int_{x_0 \in \mathbb{R}^d} x_0 \mathrm{d}\tilde{Q}_{0|t-1}(x_0|m_t) - \int_{x_0 \in \mathbb{R}^d} x_0 \mathrm{d}\tilde{Q}_{0|t}(x_0|x_t)\right)
$$

$$
\lesssim \frac{\sqrt{\bar{\alpha}_{t-1}}}{1 - \bar{\alpha}_t} \cdot \frac{1}{\tilde{q}_{t-1}(m_t)\tilde{q}_t(x_t)}\left(\int_{x,y \in \mathbb{R}^d} x\tilde{q}_{t|0}(x_t|x)\tilde{q}_{t-1|0}(m_t|y)|\Delta|\mathrm{d}\tilde{Q}_0(x)\mathrm{d}\tilde{Q}_0(y)\right)
$$

$$
= \frac{\sqrt{\bar{\alpha}_{t-1}}}{1 - \bar{\alpha}_t}\left(\int_{x,y \in \mathbb{R}^d}(x \cdot |\Delta|)\mathrm{d}\tilde{Q}_{0|t}(x|x_t)\mathrm{d}\tilde{Q}_{0|t-1}(y|m_t)\right)
$$

and, from definition of $\Delta$ and using Cauchy-Schwartz and Jensen's inequality, we have

$$
\mathbb{E}_{X_t \sim \tilde{Q}_t}\left\|\frac{\sqrt{\bar{\alpha}_{t-1}}}{1 - \bar{\alpha}_t}\int_{x,y \in \mathbb{R}^d}(x \cdot |\Delta|)\mathrm{d}\tilde{Q}_{0|t}(x|X_t)\mathrm{d}\tilde{Q}_{0|t-1}(y|m_t(X_t))\right\|^2
$$

$$
\lesssim \frac{(1 - \alpha_t)^2}{(1 - \bar{\alpha}_{t-1})^2(1 - \bar{\alpha}_t)^4} \cdot
$$

$$
\mathbb{E}_{\substack{X_t \sim \tilde{Q}_t \\ X \sim \tilde{Q}_{0|t}(\cdot|X_t) \\ Y \sim \tilde{Q}_{0|t-1}(\cdot|m_t(X_t))}}\left[\left\|\sqrt{\bar{\alpha}_t}X\right\|^2\left((\|X_t\|^2 + (1 - \bar{\alpha}_t)^2\|\nabla\log\tilde{q}_t(X_t)\|^2)\right.\right.
$$

$$
\left.\left.(\|\sqrt{\bar{\alpha}_t}X\|^2 + \|\sqrt{\bar{\alpha}_{t-1}}Y\|^2) + (\|\sqrt{\bar{\alpha}_t}X\|^4 + \|\sqrt{\bar{\alpha}_{t-1}}Y\|^4)\right)\right]
$$

$$
= \frac{(1 - \alpha_t)^2}{(1 - \bar{\alpha}_{t-1})^2(1 - \bar{\alpha}_t)^4} \cdot
$$

$$\mathbb{E}_{\substack{X_t \sim \tilde{Q}_t \\ X \sim \tilde{Q}_{0|t}(\cdot|X_t) \\ Y \sim \tilde{Q}_{0|t-1}(\cdot|m_t(X_t))}} \left[ \left\| \sqrt{\bar{\alpha}_t} X \right\|^4 \left( \|X_t\|^2 + (1 - \bar{\alpha}_t)^2 \left\| \nabla \log \tilde{q}_t(X_t) \right\|^2 \right) \right.$$

$$+ \left\| \sqrt{\bar{\alpha}_t} X \right\|^2 \left\| \sqrt{\bar{\alpha}_{t-1}} Y \right\|^2 \left( \|X_t\|^2 + (1 - \bar{\alpha}_t)^2 \left\| \nabla \log \tilde{q}_t(X_t) \right\|^2 \right)$$

$$\left. + \left\| \sqrt{\bar{\alpha}_t} X \right\|^6 + \left\| \sqrt{\bar{\alpha}_t} X \right\|^2 \left\| \sqrt{\bar{\alpha}_{t-1}} Y \right\|^4 \right]$$

$$\leq \frac{(1 - \alpha_t)^2}{(1 - \bar{\alpha}_{t-1})^2 (1 - \bar{\alpha}_t)^4} \cdot$$

$$\left( \mathbb{E}_{X \sim \tilde{Q}_0} \left\| \sqrt{\bar{\alpha}_t} X \right\|^6 \right)^{2/3} \left( \mathbb{E}_{X_t \sim \tilde{Q}_t} (\|X_t\|^6 + (1 - \bar{\alpha}_t)^6 \left\| \nabla \log \tilde{q}_t(X_t) \right\|^6) \right)^{1/3}$$

$$+ \left( \mathbb{E}_{X \sim \tilde{Q}_0} \left\| \sqrt{\bar{\alpha}_t} X \right\|^6 \right)^{1/3} \left( \mathbb{E}_{\substack{X_t \sim \tilde{Q}_t \\ Y \sim \tilde{Q}_{0|t-1}(\cdot|m_t(X_t))}} \left\| \sqrt{\bar{\alpha}_{t-1}} Y \right\|^6 \right)^{1/3}$$

$$\left( \mathbb{E}_{X_t \sim \tilde{Q}_t} (\|X_t\|^6 + (1 - \bar{\alpha}_t)^6 \left\| \nabla \log \tilde{q}_t(X_t) \right\|^6) \right)^{1/3}$$

$$+ \mathbb{E}_{X \sim \tilde{Q}_0} \left\| \sqrt{\bar{\alpha}_t} X \right\|^6 + \left( \mathbb{E}_{X \sim \tilde{Q}_0} \left\| \sqrt{\bar{\alpha}_t} X \right\|^6 \right)^{1/3} \left( \mathbb{E}_{\substack{X_t \sim \tilde{Q}_t \\ Y \sim \tilde{Q}_{0|t-1}(\cdot|m_t(X_t))}} \left\| \sqrt{\bar{\alpha}_{t-1}} Y \right\|^6 \right)^{2/3}$$

$$\overset{(ii)}{\lesssim} \frac{d^3 (1 - \alpha_t)^2}{(1 - \bar{\alpha}_{t-1})^2 (1 - \bar{\alpha}_t)^4},$$

where $(ii)$ follows because, following (Liang et al., 2024, Lemmas 15–17) and by the lemma assumption that $\mathbb{E}_{X_0 \sim \tilde{Q}_0} \|X_0\|^6 \lesssim d^3$, we have

$$\mathbb{E}_{X \sim \tilde{Q}_0} \left\| \sqrt{\bar{\alpha}_t} X \right\|^6 \lesssim d^3,$$

$$\mathbb{E}_{X_t \sim \tilde{Q}_t} \|X_t\|^6 \leq \mathbb{E}_{X_0 \sim \tilde{Q}_0} \left\| \sqrt{\bar{\alpha}_t} X_0 \right\|^6 + (1 - \bar{\alpha}_t)^3 \mathbb{E}_{\bar{W} \sim \mathcal{N}(0, I_d)} \left\| \bar{W} \right\|^6 \lesssim d^3,$$

$$\mathbb{E}_{X_t \sim \tilde{Q}_t} \left\| \nabla \log \tilde{q}_t(X_t) \right\|^6 \lesssim \frac{d^3}{(1 - \bar{\alpha}_t)^3},$$

$$\mathbb{E}_{\substack{X_t \sim \tilde{Q}_t \\ Y \sim \tilde{Q}_{0|t-1}(\cdot|m_t(X_t))}} \left\| \sqrt{\bar{\alpha}_{t-1}} Y \right\|^6$$

$$\leq \mathbb{E}_{\substack{X_t \sim \tilde{Q}_t \\ Y \sim \tilde{Q}_{0|t-1}(\cdot|m_t(X_t))}} \left\| m_t - \sqrt{\bar{\alpha}_{t-1}} Y \right\|^6 + \mathbb{E}_{X_t \sim \tilde{Q}_t} \|m_t\|^6 \lesssim d^3.$$

Hence, combining the rates of all parts, we obtain that

$$(1 - \alpha_t) \sqrt{\mathbb{E}_{X_t \sim \tilde{Q}_t} \left\| \nabla \log \tilde{q}_{t-1}(m_t(X_t)) - \sqrt{\alpha_t} \nabla \log \tilde{q}_t(X_t) \right\|^2} \lesssim \frac{d^{3/2} (1 - \alpha_t)^2}{(1 - \bar{\alpha}_{t-1})^3}.$$

### H.6 Lemma 6 and its proof

**Lemma 6.** *The $\alpha_t$ defined in (8) (with $c > 1$) satisfy*

$$\frac{1 - \alpha_t}{(1 - \bar{\alpha}_{t-1})^p} \lesssim \frac{\log T \log(1/\delta)}{\delta^{p-1} T} \quad \text{while } \bar{\alpha}_T = o(T^{-1}), \quad \forall 2 \leq t \leq T, \ p \geq 1.$$

*Proof.* The proof is similar to that of (Li et al., 2024c, Eq (39)). We first prove the second relationship. First, note that if $T$ is large,

$$\delta \left( 1 + \frac{c \log T}{T} \right)^{\frac{T}{\log T}} \asymp \delta e^c > 1.$$

Thus, with any fixed $r \in (0, 1)$ such that $t \geq rT \ (\geq \frac{T}{\log T})$, we have

$$1 - \alpha_t = \frac{c \log T}{T} \min \left\{ \delta \left( 1 + \frac{c \log T}{T} \right)^t, 1 \right\} = \frac{c \log T}{T}.$$

As a result,

$$\bar{\alpha}_T \leq \prod_{t=\lfloor rT \rfloor}^{T} \alpha_t = \left(1 - \frac{c \log T}{T}\right)^{\lceil (1-r)T \rceil} \asymp \exp\left(\lceil (1-r)T \rceil \left(-\frac{c \log T}{T}\right)\right) = O(T^{-(1-r)c}).$$

(22)

Given any $c > 1$, we can always find some $r$ such that $(1-r)c > 1$ (say, $r = (c-1)/2$ if $c \in (1,2)$ and $r = 1/4$ if $c \geq 2$). This shows that $\alpha_t$ satisfies $\bar{\alpha}_T = o\left(T^{-1}\right)$ if $c > 1$.

Now, for the first relationship, define $\tau$ such that

$$\delta \left(1 + \frac{c \log T}{T}\right)^{\tau} \leq 1 < \delta \left(1 + \frac{c \log T}{T}\right)^{\tau+1}.$$

(23)

Here $\tau$ is unique since $1 - \alpha_t$ is non-decreasing. In other words, $\tau$ is the last time that $1 - \alpha_t$ is exponentially growing. Assume that $T$ is large enough such that $\tau \geq 2$. Below, we show that

$$1 - \bar{\alpha}_{t-1} \geq \frac{1}{3} \delta \left(1 + \frac{c \log T}{T}\right)^t, \quad \forall 2 \leq t \leq \tau.$$

(24)

If $t = 2$,

$$1 - \bar{\alpha}_{t-1} = 1 - \bar{\alpha}_1 = 1 - \alpha_1 = \delta \geq \frac{1}{3} \delta \left(1 + \frac{c \log T}{T}\right).$$

Here the last inequality holds when $T$ is sufficiently large. For $t > 2$, suppose for purpose of contradiction that there exists $2 < t_0 \leq \tau$ such that

$$1 - \bar{\alpha}_{t_0-1} < \frac{1}{3} \delta \left(1 + \frac{c \log T}{T}\right)^{t_0} \text{ while } 1 - \bar{\alpha}_{t-1} \geq \frac{1}{3} \delta \left(1 + \frac{c \log T}{T}\right)^t, \forall 2 \leq t \leq t_0 - 1.$$

In words, $t_0$ is defined as the *first* time that (24) is violated. To arrive at a contradiction, we first write

$$1 - \bar{\alpha}_{t_0-1} = (1 - \bar{\alpha}_{t_0-2}) \left(1 + \frac{\bar{\alpha}_{t_0-2}(1 - \alpha_{t_0-1})}{1 - \bar{\alpha}_{t_0-2}}\right)$$

$$\geq \frac{1}{3} \delta \left(1 + \frac{c \log T}{T}\right)^{t_0-1} \left(1 + \frac{\bar{\alpha}_{t_0-2}(1 - \alpha_{t_0-1})}{1 - \bar{\alpha}_{t_0-2}}\right).$$

Here the inequality holds because $t_0$ is the first time that (24) is violated, and thus (24) stills holds for $t = t_0 - 1$. Also,

$$1 - \bar{\alpha}_{t_0-2} \leq 1 - \bar{\alpha}_{t_0-1} \overset{(i)}{<} \frac{1}{3} \delta \left(1 + \frac{c \log T}{T}\right)^{t_0} \overset{(ii)}{\leq} \frac{1}{2} \delta \left(1 + \frac{c \log T}{T}\right)^{t_0-1} \overset{(iii)}{\leq} \frac{1}{2}$$

where $(i)$ holds because (24) is violated at $t = t_0$, $(ii)$ holds when $T$ is sufficiently large, and $(iii)$ holds because $t_0 - 1 \leq \tau$ and by the definition of $\tau$ in (23). Thus,

$$\frac{\bar{\alpha}_{t_0-2}(1 - \alpha_{t_0-1})}{1 - \bar{\alpha}_{t_0-2}} \geq \frac{\frac{1}{2} \frac{c \log T}{T} \delta \left(1 + \frac{c \log T}{T}\right)^{t_0-1}}{\frac{1}{2} \delta \left(1 + \frac{c \log T}{T}\right)^{t_0-1}} = \frac{c \log T}{T},$$

and thus

$$1 - \bar{\alpha}_{t_0-1} \geq \frac{1}{3} \delta \left(1 + \frac{c \log T}{T}\right)^{t_0-1} \left(1 + \frac{\bar{\alpha}_{t_0-2}(1 - \alpha_{t_0-1})}{1 - \bar{\alpha}_{t_0-2}}\right) \geq \frac{1}{3} \delta \left(1 + \frac{c \log T}{T}\right)^{t_0}.$$

We have reached a contradiction. Therefore, we have shown that (24) holds.

Now, (24) implies that

$$1 - \bar{\alpha}_{t-1} \geq \frac{1}{3} \delta \left(1 + \frac{c \log T}{T}\right)^t \geq \frac{1}{3} \delta \left(1 + \frac{c \log T}{T}\right)^{t/p}, \quad \forall 2 \leq t \leq \tau.$$

There are two cases:

- If $2 \leq t \leq \tau$, then

$$
\frac{1 - \alpha_t}{(1 - \bar{\alpha}_{t-1})^p} \leq \frac{\frac{c \log T}{T} \delta \left(1 + \frac{c \log T}{T}\right)^t}{\frac{1}{3^p} \delta^p \left(1 + \frac{c \log T}{T}\right)^t} = \frac{3^p c \log T}{\delta^{p-1} T}.
$$

- If $t > \tau$, then

$$
\frac{1 - \alpha_t}{(1 - \bar{\alpha}_{t-1})^p} \leq \frac{1 - \alpha_t}{(1 - \bar{\alpha}_{\tau-1})^p} \leq \frac{\frac{c \log T}{T}}{\frac{1}{3^p} \delta^p \left(1 + \frac{c \log T}{T}\right)^{\tau}} = \frac{\frac{c \log T}{T} \left(1 + \frac{c \log T}{T}\right)}{3^{-p} \delta^{p-1} \left(1 + \frac{c \log T}{T}\right)^{\tau+1}}
$$

$$
< \frac{3^p c \log T}{\delta^{p-1} T} \left(1 + \frac{c \log T}{T}\right).
$$

In both cases, if $T$ is large enough, noting that $c \gtrsim \log(1/\delta)$, we have

$$
\frac{1 - \alpha_t}{(1 - \bar{\alpha}_{t-1})^p} \leq \frac{4^p c \log T}{\delta^{p-1} T} \lesssim \frac{\log T \log(1/\delta)}{\delta^{p-1} T}, \quad \forall 2 \leq t \leq T
$$

because $p$ and $c$ are constants (that do not depend on $T$, $d$, and $\delta$). The proof is now complete. $\qquad \square$

### H.7 LEMMA 7 AND ITS PROOF

**Lemma 7.** *With the $\alpha_t$ defined in* (8)*, given any $p > 0$, if $\delta p < 1$,*

$$
\sum_{t=2}^T (1 - \alpha_t) \bar{\alpha}_t^p \leq \left(\frac{1}{p}(1 - \delta)^p e^{-p\delta \log(1/\delta)} + (1 - \delta)^p \frac{e^{-p\delta \log(1/\delta)} - 1}{1 - \delta p}\right) \left(1 + O\left(\frac{\log T}{T}\right)\right).
$$

*Further, when $\delta \ll 1$,*

$$
\sum_{t=2}^T (1 - \alpha_t) \bar{\alpha}_t^p \leq \frac{1}{p} - \left(1 + \frac{p+1}{2p}\right) \frac{c \log T}{T} + \tilde{O}\left(\frac{1}{T^2}\right).
$$

*Proof.* Define the sum as $s_T$. Recall that

$$
1 - \alpha_1 = \delta, \quad 1 - \alpha_t = \frac{c \log T}{T} \min\left\{\delta \left(1 + \frac{c \log T}{T}\right)^t, 1\right\}, \quad \forall 2 \leq t \leq T.
$$

We first note a relationship that for fixed $\delta \neq 0$ and $p > 0$. As $z \to \infty$,

$$
(1 - \delta z^{-1})^{pz} = e^{pz \log(1 - \delta z^{-1})} = e^{pz(-\delta z^{-1} + \delta^2 z^{-2}/2 + O(z^{-3}))} = e^{-\delta p}(1 + \delta^2 p z^{-1}/2) + O(z^{-2}).
$$
(25)

We also use the fact from binomial series that

$$
(1 - z^{-1})^p = 1 - pz^{-1} + \frac{p(p-1)}{2} z^{-2} + O(z^{-3}).
$$
(26)

Define $t^* := \sup\left\{t \in [1, T] : \delta \left(1 + \frac{c \log T}{T}\right)^t \leq 1\right\}$. Thus, $\alpha_t \equiv 1 - \frac{c \log T}{T}$ for all $t > t^*$. Note that when $T$ becomes large, $t^* = \Theta\left(\frac{T}{\log T}\right)$. To further understand the big-$\Theta$ term, note that using (25),

$$
\delta \left(1 + \frac{c \log T}{T}\right)^{\frac{T \log(1/\delta)}{c \log T} - \log(1/\delta)}
$$

$$
= \left(1 + \frac{\log(1/\delta) c \log T}{2T} + \tilde{O}\left(\frac{1}{T^2}\right)\right) \left(1 - \frac{\log(1/\delta) c \log T}{T} + \tilde{O}\left(\frac{1}{T^2}\right)\right)
$$

$$
= 1 - \frac{\log(1/\delta) c \log T}{2T} + \tilde{O}\left(\frac{1}{T^2}\right)
$$

$$< 1 \text{ as } T \to \infty.$$

This implies that

$$t^* \geq \log(1/\delta) \left( \frac{T}{c \log T} - 1 \right). \tag{27}$$

To start, we suppose $T > t^*$ is large enough and decompose the sum as

$$
\begin{aligned}
s_T &= \sum_{t=2}^{t^*} (1 - \alpha_t) \bar{\alpha}_t^p + \sum_{t=t^*+1}^{T} (1 - \alpha_t) \bar{\alpha}_t^p \\
&= \frac{c \log T}{T} \delta (1 - \delta)^p \sum_{t=2}^{t^*} \left( 1 + \frac{c \log T}{T} \right)^t \prod_{i=2}^{t} \left( 1 - \delta \frac{c \log T}{T} \left( 1 + \frac{c \log T}{T} \right)^i \right)^p \\
&\quad + \bar{\alpha}_{t^*}^p \frac{c \log T}{T} \sum_{t=t^*+1}^{T} \left( 1 - \frac{c \log T}{T} \right)^{p(t-t^*)}. 
\end{aligned} \tag{28}
$$

Now we first focus on the second term in (28).

$$
\begin{aligned}
&\frac{c \log T}{T} \sum_{t=t^*+1}^{T} \left( 1 - \frac{c \log T}{T} \right)^{p(t-t^*)} \\
&= \left( 1 - \frac{c \log T}{T} \right)^p \frac{c \log T}{T} \cdot \frac{1 - \left( 1 - \frac{c \log T}{T} \right)^{p(T-t^*)}}{1 - \left( 1 - \frac{c \log T}{T} \right)^p} \\
&\overset{(i)}{=} \left( 1 - \frac{c \log T}{T} \right)^p \frac{c \log T}{T} \cdot \frac{1 - \left( 1 - \frac{c \log T}{T} \right)^{p(T-t^*)}}{1 - \left( 1 - \frac{pc \log T}{T} + p(p-1) \frac{c^2 (\log T)^2}{2T^2} + \tilde{O} \left( \frac{1}{T^3} \right) \right)} \\
&\overset{(ii)}{=} \frac{1}{p} \left( 1 - \frac{pc \log T}{T} + \tilde{O} \left( \frac{1}{T^2} \right) \right) \left( 1 + \frac{(p-1)c \log T}{2T} + \tilde{O} \left( \frac{1}{T^2} \right) - O \left( \frac{1}{T^{pc/2}} \right) \right) \\
&= \frac{1}{p} \left( 1 - \frac{(p+1)c \log T}{2T} \right) + \tilde{O} \left( \frac{1}{T^2} \right)
\end{aligned}
$$

where $(i)$ follows from (26), and $(ii)$ is because $t^* = \Theta(T/\log T)$ and thus $T - t^* > T/2$ for large $T$. Also, for all $t = 2, \ldots, t^*$,

$$
\begin{aligned}
\bar{\alpha}_t^p &= (1 - \delta)^p \prod_{i=2}^{t} \left( 1 - \delta \frac{c \log T}{T} \left( 1 + \frac{c \log T}{T} \right)^i \right)^p \\
&\leq (1 - \delta)^p \left( 1 - \delta \frac{c \log T}{T} \right)^{p(t-1)}
\end{aligned}
$$

Thus, we have

$$
\begin{aligned}
\bar{\alpha}_{t^*}^p &\leq (1 - \delta)^p \left( 1 - \delta \frac{c \log T}{T} \right)^{p(t^*-1)} \\
&= (1 - \delta)^p \left( 1 + \delta p \frac{c \log T}{T} + \tilde{O} \left( \frac{1}{T^2} \right) \right) \left( 1 - \delta \frac{c \log T}{T} \right)^{pt^*} \\
&\overset{(iii)}{\leq} (1 - \delta)^p \left( 1 + \delta p (1 + \log(1/\delta)) \frac{c \log T}{T} \right) \times \\
&\quad e^{-p\delta \log(1/\delta)} \left( 1 + \delta^2 p \log(1/\delta) \frac{c \log T}{2T} \right) + \tilde{O} \left( \frac{1}{T^2} \right) \\
&= (1 - \delta)^p e^{-p\delta \log(1/\delta)} \left( 1 + \delta p \left( 1 + \log(1/\delta) + (\delta/2) \log(1/\delta) \right) \frac{c \log T}{T} \right) + \tilde{O} \left( \frac{1}{T^2} \right).
\end{aligned}
$$

Here $(iii)$ follows because using (27) and (25), we have that

$$
\left(1 - \delta \frac{c \log T}{T}\right)^{pt^*} \leq \left(1 - \delta \frac{c \log T}{T}\right)^{p \log(1/\delta)\left(\frac{T}{c \log T} - 1\right)}
$$
$$
= \left(1 + \delta p \log(1/\delta) \frac{c \log T}{T}\right) e^{-p \delta \log(1/\delta)} \left(1 + \delta^2 p \log(1/\delta) \frac{c \log T}{2T}\right) + \tilde{O}\left(\frac{1}{T^2}\right). \quad (29)
$$

Thus, the second term in (28) satisfies that

$$
\sum_{t=t^*+1}^{T} (1 - \alpha_t) \bar{\alpha}_t^p
$$
$$
\leq \frac{1}{p} (1 - \delta)^p e^{-p \delta \log(1/\delta)} \left(1 - \left(\frac{p+1}{2} - \delta p \left(1 + \log(1/\delta) + (\delta/2) \log(1/\delta)\right)\right) \frac{c \log T}{T}\right)
$$
$$
+ \tilde{O}\left(\frac{1}{T^2}\right). \quad (30)
$$

Now we turn to the first term in (28), in which the summation can be upper-bounded as

$$
\sum_{t=2}^{t^*} \left(1 + \frac{c \log T}{T}\right)^t \prod_{i=2}^{t} \left(1 - \delta \frac{c \log T}{T} \left(1 + \frac{c \log T}{T}\right)^i\right)^p
$$
$$
\leq \sum_{t=2}^{t^*} \left(1 + \frac{c \log T}{T}\right)^t \left(1 - \delta \frac{c \log T}{T}\right)^{p(t-1)} =: \left(1 + \frac{c \log T}{T}\right) \sum_{t=1}^{t^*-1} q^t
$$

where

$$
q := \left(1 + \frac{c \log T}{T}\right) \left(1 - \delta \frac{c \log T}{T}\right)^p
$$
$$
= \left(1 + \frac{c \log T}{T}\right) \left(1 - \delta p \frac{c \log T}{T} + \delta^2 p(p-1) \frac{c^2 (\log T)^2}{2T^2} + \tilde{O}\left(\frac{1}{T^3}\right)\right)
$$
$$
= 1 + (1 - \delta p) \frac{c \log T}{T} + \left(\frac{\delta^2 p(p-1)}{2} - \delta p\right) \frac{c^2 (\log T)^2}{T^2} + \tilde{O}\left(\frac{1}{T^3}\right).
$$

Note that by assumption $\delta p < 1$. Also, by definition of $t^*$ and (29),

$$
\delta q^{t^*} \leq \left(1 - \delta \frac{c \log T}{T}\right)^{pt^*}
$$
$$
\leq e^{-p \delta \log(1/\delta)} \left(1 + \delta p \log(1/\delta)(1 + \delta/2) \frac{c \log T}{T}\right) + \tilde{O}\left(\frac{1}{T^2}\right).
$$

Thus, we have

$$
\delta \frac{c \log T}{T} \sum_{t=1}^{t^*-1} q^t = \frac{c \log T}{T} \times \frac{\delta q^{t^*} - \delta q}{q - 1}
$$
$$
\leq \frac{c \log T}{T} \times \frac{e^{-p \delta \log(1/\delta)} \left(1 + \delta p \log(1/\delta)(1 + \delta/2) \frac{c \log T}{T}\right) + \tilde{O}\left(\frac{1}{T^2}\right) - \delta q}{q - 1}
$$
$$
\overset{(iv)}{=} \frac{c \log T}{T} \times \frac{e^{-p \delta \log(1/\delta)} \left(1 + \delta p \log(1/\delta)(1 + \delta/2) \frac{c \log T}{T}\right) + \tilde{O}\left(\frac{1}{T^2}\right) - 1 - (1 - \delta p) \frac{c \log T}{T} + \tilde{O}\left(\frac{1}{T^2}\right)}{(1 - \delta p) \frac{c \log T}{T} + \left(\frac{\delta^2 p(p-1)}{2} - \delta p\right) \frac{c^2 (\log T)^2}{T^2} + \tilde{O}\left(\frac{1}{T^3}\right)}
$$
$$
= \left(\frac{e^{-p \delta \log(1/\delta)} - 1}{1 - \delta p} + \left(\frac{\delta p \log(1/\delta)(1 + \delta/2)}{1 - \delta p} - 1\right) \frac{c \log T}{T}\right) \times
$$
$$
\left(1 + \frac{\delta p - \frac{\delta^2 p(p-1)}{2}}{1 - \delta p} \cdot \frac{c \log T}{T}\right) + \tilde{O}\left(\frac{1}{T^2}\right)
$$

$$= \frac{e^{-p\delta \log(1/\delta)} - 1}{1 - \delta p} + \left( \frac{\delta p \log(1/\delta)(1 + \delta/2)}{1 - \delta p} - 1 + \frac{e^{-p\delta \log(1/\delta)} - 1}{1 - \delta p} \cdot \frac{\delta p(1 - \delta(p-1)/2)}{1 - \delta p} \right) \frac{c \log T}{T}$$
$$+ \tilde{O}\left( \frac{1}{T^2} \right).$$

where $(iv)$ follows from (26). Therefore,

$$\sum_{t=2}^{t^*}(1 - \alpha_t)\bar{\alpha}_t^p = (1 - \delta)^p \left( 1 + \frac{c \log T}{T} \right) \left( \delta \frac{c \log T}{T} \sum_{t=1}^{t^*-1} q^t \right)$$

$$\leq (1 - \delta)^p \frac{e^{-p\delta \log(1/\delta)} - 1}{1 - \delta p}$$

$$+ (1 - \delta)^p \left( \frac{\delta p \log(1/\delta)(1 + \delta/2)}{1 - \delta p} - 1 \right.$$

$$\left. + \frac{e^{-p\delta \log(1/\delta)} - 1}{1 - \delta p} \left( 1 + \frac{\delta p(1 - \delta(p-1)/2)}{1 - \delta p} \right) \right) \frac{c \log T}{T} + \tilde{O}\left( \frac{1}{T^2} \right). \quad (31)$$

Combining (30) and (31), we have that

$$s_T \leq \underbrace{\left( \frac{1}{p}(1 - \delta)^p e^{-p\delta \log(1/\delta)} + (1 - \delta)^p \frac{e^{-p\delta \log(1/\delta)} - 1}{1 - \delta p} \right)}_{=: s_\infty} \left( 1 + O\left( \frac{\log T}{T} \right) \right).$$

Also, for all large $T$'s, since $s_\infty \to \frac{1}{p}$ and $\delta \log(1/\delta) \to 0$ as $\delta \to 0$, when $\delta \ll 1$,

$$s_T \leq \frac{1}{p} - \left( 1 + \frac{p+1}{2p} \right) \frac{c \log T}{T} + \tilde{O}\left( \frac{1}{T^2} \right).$$

The proof is now complete. $\qquad \square$

# I  PROOFS IN SECTION 4

## I.1  PROOF OF THEOREM 3

Fix $t \geq 1$. Using the forward model in (5), we have that $q_{t|0,y}$ is the p.d.f. of $\mathcal{N}(\sqrt{\bar{\alpha}_t}(I_d - H^\dagger H)x_0 + \sqrt{\bar{\alpha}_t}H^\dagger y, \Sigma_{t|0,y})$. Thus,

$$\nabla \log q_{t|y}(x) = \frac{1}{q_{t|y}(x)} \int_{x_0 \in \mathbb{R}^d} \nabla q_{t|0,y}(x|x_0) \mathrm{d}Q_{0|y}(x_0)$$

$$= -\frac{1}{q_{t|y}(x)} \Sigma_{t|0,y}^{-1} \int_{x_0 \in \mathbb{R}^d} q_{t|0,y}(x|x_0)(x - \sqrt{\bar{\alpha}_t}(I_d - H^\dagger H)x_0 - \sqrt{\bar{\alpha}_t}H^\dagger y) \mathrm{d}Q_{0|y}(x_0)$$

$$= -\Sigma_{t|0,y}^{-1}(x - \sqrt{\bar{\alpha}_t}H^\dagger y)$$

$$+ \frac{\sqrt{\bar{\alpha}_t}}{q_{t|y}(x)} \Sigma_{t|0,y}^{-1}(I_d - H^\dagger H) \int_{x_0 \in \mathbb{R}^d} q_{t|0,y}(x|x_0)x_0 \mathrm{d}Q_{0|y}(x_0).$$

Thus, the equality for $\nabla \log q_{t|y}$ is established because by Lemma 9,

$$(\sigma_y^2 H^\dagger (H^\dagger)^\intercal + (1 - \bar{\alpha}_t)I_d)^{-1}(I_d - H^\dagger H) = (1 - \bar{\alpha}_t)^{-1}(I_d - H^\dagger H).$$

To see the optimality with $f_{t,y}^*$, fix $t \geq 1$ and $x \in \mathbb{R}^d$. First note that $(I_d - H^\dagger H)f_{t,y}^*(x) = (I_d - H^\dagger H)\Sigma_{t|0,y}^{-1}(\sqrt{\bar{\alpha}_t}H^\dagger y - H^\dagger Hx) = 0$ by Lemma 9. Now, suppose that $f_{t,y} = f_{t,y}^* + v$ such that $(I_d - H^\dagger H)f_{t,y} = 0 \implies (I_d - H^\dagger H)v = 0$. From the definition of $\Delta_{t,y}$ in (7),

$$\Delta_{t,y}(x) = (I_d - H^\dagger H)(\nabla \log q_{t|y}(x) - \nabla \log q_t(x)) + (H^\dagger H)\nabla \log q_{t|y}(x) - f_{t,y}(x)$$

where

$$(H^\dagger H)\nabla \log q_{t|y}(x) - f_{t,y}(x)$$

$$= (H^\dagger H)\Sigma_{t|0,y}^{-1}(\sqrt{\bar\alpha_t}H^\dagger y - x) - \Sigma_{t|0,y}^{-1}(\sqrt{\bar\alpha_t}H^\dagger y - H^\dagger H x) - v$$

$$= -(H^\dagger H)\Sigma_{t|0,y}^{-1}(I_d - H^\dagger H)x - (I_d - H^\dagger H)\Sigma_{t|0,y}^{-1}(\sqrt{\bar\alpha_t}H^\dagger y - H^\dagger H x) - v$$

$$= -v$$

where the last line follows from Lemma 9.

Thus, if $v = 0$, then $f_{t,y} = f_{t,y}^*$, and we have

$$\Delta_{t,y} = (I_d - H^\dagger H)(\nabla \log q_{t|y}(x) - \nabla \log q_t(x)). \tag{32}$$

Also, if $v \neq 0$, since $v$ is orthogonal to the space induced by $(I_d - H^\dagger H)$, we have

$$\|\Delta_{t,y}(x)\|^2 = \|(I_d - H^\dagger H)(\nabla \log q_{t|y}(x) - \nabla \log q_t(x))\|^2 + \|v\|^2 \tag{33}$$

which is minimized at $v = 0$. The proof is now complete.

### I.2 PROOF OF THEOREM 4

Fix $t \geq 2$. Recall that the unconditional score $\nabla \log q_t(x)$ is

$$\nabla \log q_t(x) = \frac{1}{q_t(x)} \int_{x_0 \in \mathbb{R}^d} \nabla q_{t|0}(x|x_0) \mathrm{d}Q_0(x_0)$$

$$= -\frac{1}{(1 - \bar\alpha_t)q_t(x)} \int_{x_0 \in \mathbb{R}^d} q_{t|0}(x|x_0)(x - \sqrt{\bar\alpha_t}x_0) \mathrm{d}Q_0(x_0)$$

$$= -\frac{1}{(1 - \bar\alpha_t)}x + \frac{\sqrt{\bar\alpha_t}}{(1 - \bar\alpha_t)q_t(x)} \int_{x_0 \in \mathbb{R}^d} q_{t|0}(x|x_0)x_0 \mathrm{d}Q_0(x_0)$$

since $q_{t|0}$ is the p.d.f. of $\mathcal{N}(\sqrt{\bar\alpha_t}x_0, (1 - \bar\alpha_t)I_d)$.

In the first half, we consider the case where $\sigma_y^2$ is known, and thus $f_{t,y} = f_{t,y}^*$ in (10). Note that from Theorem 3,

$$\nabla \log q_{t|y}(x) = \Sigma_{t|0,y}^{-1}(\sqrt{\bar\alpha_t}H^\dagger y - x)$$

$$+ \frac{\sqrt{\bar\alpha_t}}{q_{t|y}(x)}\Sigma_{t|0,y}^{-1}(I_d - H^\dagger H) \int_{x_0 \in \mathbb{R}^d} q_{t|0,y}(x|x_0)x_0 \mathrm{d}Q_{0|y}(x_0).$$

Here we also recall from Theorem 3 that

$$\Sigma_{t|0,y} := \bar\alpha_t \sigma_y^2 H^\dagger (H^\dagger)^\intercal + (1 - \bar\alpha_t)I_d.$$

Since $H^\dagger(H^\dagger)^\intercal$ is positive semi-definite, all its eigenvalues are non-negative. Write the eigendecomposition as $H^\dagger(H^\dagger)^\intercal = P\mathrm{diag}(D_1, \ldots, D_d)P^\intercal$ where $D_1 \geq \cdots \geq D_d \geq 0$, $\forall i \in [d]$. Then, $\lambda_{min}(\Sigma_{t|0,y}) \geq \bar\alpha_t \sigma_y^2 D_d + 1 - \bar\alpha_t \geq 1 - \bar\alpha_t$, and we get

$$\left\|\Sigma_{t|0,y}^{-1}\right\| \leq \frac{1}{1 - \bar\alpha_t}. \tag{34}$$

Also, from (6), with the $f_{t,y}^*$ in (10),

$$g_{t,y}(x) = f_{t,y}^*(x) + (I_d - H^\dagger H)\nabla \log q_t(x)$$

$$= \Sigma_{t|0,y}^{-1}\left(\sqrt{\bar\alpha_t}H^\dagger y - H^\dagger H x\right) - \frac{1}{(1 - \bar\alpha_t)}(I_d - H^\dagger H)x$$

$$+ \frac{\sqrt{\bar\alpha_t}}{(1 - \bar\alpha_t)q_t(x)}(I_d - H^\dagger H) \int_{x_0 \in \mathbb{R}^d} q_{t|0}(x|x_0)x_0 \mathrm{d}Q_0(x_0)$$

$$\overset{(i)}{=} \Sigma_{t|0,y}^{-1}\left(\sqrt{\bar\alpha_t}H^\dagger y - H^\dagger H x\right) - \Sigma_{t|0,y}^{-1}(I_d - H^\dagger H)x$$

$$+ \frac{\sqrt{\bar\alpha_t}}{q_t(x)}\Sigma_{t|0,y}^{-1}(I_d - H^\dagger H) \int_{x_0 \in \mathbb{R}^d} q_{t|0}(x|x_0)x_0 \mathrm{d}Q_0(x_0)$$

$$= \Sigma_{t|0,y}^{-1}\left(\sqrt{\bar\alpha_t}H^\dagger y - x\right) + \frac{\sqrt{\bar\alpha_t}}{q_t(x)}\Sigma_{t|0,y}^{-1}(I_d - H^\dagger H) \int_{x_0 \in \mathbb{R}^d} q_{t|0}(x|x_0)x_0 \mathrm{d}Q_0(x_0)$$

where $(i)$ follows from Lemma 9. Then, the norm-squared of the score mismatch at time $t \geq 2$ is

$$\|\Delta_{t,y}\|^2 = \left\|\nabla \log q_{t|y} - g_{t,y}\right\|^2 = \left\|(I_d - H^\dagger H)(\nabla \log q_{t|y} - \nabla \log q_t)\right\|^2$$

$$\leq \bar{\alpha}_t \left\|\Sigma_{t|0,y}^{-1}\right\|^2 \left\|\frac{\int_{x_0 \in \mathbb{R}^d} q_{t|0,y}(x|x_0) x_0 \mathrm{d}Q_{0|y}(x_0)}{q_{t|y}(x)} - \frac{\int_{x_0 \in \mathbb{R}^d} q_{t|0}(x|x_0) x_0 \mathrm{d}Q_0(x_0)}{q_t(x)}\right\|^2$$

$$\overset{(ii)}{\leq} \bar{\alpha}_t \left\|\Sigma_{t|0,y}^{-1}\right\|^2 \int_{x_a, x_b \in \mathbb{R}^d} \|x_a - x_b\|^2 \, \mathrm{d}Q_{0|t,y}(x_a) \mathrm{d}Q_{0|t}(x_b)$$

$$\leq \bar{\alpha}_t \left\|\Sigma_{t|0,y}^{-1}\right\|^2 \max_{\substack{x_a \in \mathrm{supp}(Q_{0|y}) \\ x_b \in \mathrm{supp}(Q_0)}} \|x_a - x_b\|^2$$

$$\overset{(iii)}{\lesssim} \frac{\bar{\alpha}_t}{(1 - \bar{\alpha}_t)^2} d. \tag{35}$$

Here $(ii)$ follows from Jensen's inequality, and $(iii)$ follows by (34) and from the assumption that $Q_0$ has bounded support (and thus also for both $Q_{0|t}$ and $Q_{0|t,y}$). Therefore, with the $\alpha_t$ in (8) (cf. Lemma 6), since $1 - \bar{\alpha}_t \geq 1 - \delta$ which is a constant, Assumption 4 is satisfied for all $\sigma_y^2 \geq 0$. Thus, Theorem 2 holds with $\gamma = 1$ and $r = 2$.

Now, we consider the case where $\sigma_y^2$ is unknown, and the conditional sampler of interest is $g_{t,y}^N(x) = f_{t,y}^N(x) + (I_d - H^\dagger H) \nabla \log q_t(x)$ where $f_{t,y}^N(x) = (1 - \bar{\alpha}_t)^{-1} \left(\sqrt{\bar{\alpha}_t} H^\dagger y - H^\dagger H x\right)$. With the same notation as in the proof of Theorem 3, we can write $v = f_{t,y}^N - f_{t,y}^* = \left((1 - \bar{\alpha}_t)^{-1} I_d - \Sigma_{t|0,y}^{-1}\right) \left(\sqrt{\bar{\alpha}_t} H^\dagger y - H^\dagger H x\right)$. Note that $v$ still satisfies that $(I_d - H^\dagger H)v = 0$. Using the result in (33), we have

$$\left\|\Delta_{t,y}^N\right\|^2 = \left\|(I_d - H^\dagger H)(\nabla \log q_{t|y}(x) - \nabla \log q_t(x))\right\|^2$$

$$+ \left\|(\Sigma_{t|0,y}^{-1} - (1 - \bar{\alpha}_t)^{-1} I_d)\left(\sqrt{\bar{\alpha}_t} H^\dagger y - H^\dagger H x\right)\right\|^2$$

where the first term is the same as in (35) which can be upper-bounded in a similar way. To upper-bound the second term, note that by Woodbury matrix identity,

$$\left\|\Sigma_{t|0,y}^{-1} - \frac{1}{1 - \bar{\alpha}_t} I_d\right\| = \frac{\bar{\alpha}_t \sigma_y^2}{(1 - \bar{\alpha}_t)^2} \left\|H^\dagger \left(I_p + \frac{\sigma_y^2}{1 - \bar{\alpha}_t}(H^\dagger)^\intercal H^\dagger\right)^{-1} (H^\dagger)^\intercal\right\|$$

$$\lesssim \frac{\bar{\alpha}_t \sigma_y^2}{(1 - \bar{\alpha}_t)^2} \tag{36}$$

where the inequality follows because $\|H^\dagger\| \lesssim 1$ is a constant and the minimum eigenvalue of $(I_p + \frac{\sigma_y^2}{1 - \bar{\alpha}_t}(H^\dagger)^\intercal H^\dagger)$ is at least 1. Thus,

$$\mathbb{E}_{Q_{t|y}} \left\|(\Sigma_{t|0,y}^{-1} - (1 - \bar{\alpha}_t)^{-1} I_d)\left(\sqrt{\bar{\alpha}_t} H^\dagger y - H^\dagger H X_t\right)\right\|^2$$

$$\overset{(iv)}{\leq} \frac{\bar{\alpha}_t^2 \sigma_y^4}{(1 - \bar{\alpha}_t)^4} \mathbb{E}_{Q_{t|y}} \left\|\sqrt{\bar{\alpha}_t} H^\dagger y - H^\dagger H X_t\right\|^2$$

$$= \frac{\bar{\alpha}_t^2 \sigma_y^4}{(1 - \bar{\alpha}_t)^4} \mathbb{E}_{Q_{0|y}} \mathbb{E}_{Q_{t|0,y}} \left\|\sqrt{\bar{\alpha}_t} H^\dagger y - H^\dagger H X_t\right\|^2$$

$$= \frac{\bar{\alpha}_t^2 \sigma_y^4}{(1 - \bar{\alpha}_t)^4} \mathbb{E}_{Q_{0|y}} \mathbb{E}_{Q_{t|0}} \left\|\sqrt{\bar{\alpha}_t} H^\dagger y - H^\dagger H X_t\right\|^2$$

$$\leq \frac{2\bar{\alpha}_t^2 \sigma_y^4}{(1 - \bar{\alpha}_t)^4} \mathbb{E}_{Q_{0|y}} \left[\bar{\alpha}_t \left\|H^\dagger y - H^\dagger H X_0\right\|^2 + \mathbb{E}_{Q_{t|0}} \left\|H^\dagger H(X_t - \sqrt{\bar{\alpha}_t} X_0)\right\|^2\right]$$

$$\leq \frac{2\bar{\alpha}_t^2 \sigma_y^4}{(1 - \bar{\alpha}_t)^4} \mathbb{E}_{Q_{0|y}} \left[\bar{\alpha}_t \left\|H^\dagger y\right\|^2 + \bar{\alpha}_t \left\|X_0\right\|^2 + d(1 - \bar{\alpha}_t)\right]$$

$$\overset{(v)}{\lesssim} \frac{\bar{\alpha}_t^2 \sigma_y^4}{(1 - \bar{\alpha}_t)^4} d$$

where $(iv)$ follows from (36), and $(v)$ follows from the fact that $Q_{0|y}$ has bounded support. Similarly, for general moments $\ell \geq 2$,

$$\mathbb{E}_{Q_{t|y}} \left\| \left(\Sigma_{t|0,y}^{-1} - (1 - \bar{\alpha}_t)^{-1} I_d\right) \left(\sqrt{\bar{\alpha}_t} H^\dagger y - H^\dagger H X_t\right) \right\|^\ell \lesssim \left( \frac{\bar{\alpha}_t \sigma_y^2}{(1 - \bar{\alpha}_t)^2} d \right)^{\ell/2}.$$

Therefore, with the $\alpha_t$ in (8), since $1 - \bar{\alpha}_t \geq 1 - \delta$ which is a constant, we still have that Assumption 4 is satisfied (see Lemma 6), and Theorem 2 still holds with $\gamma = 1$ and $r = 4$. The proof is now complete.

### I.3 THEOREM 6 AND ITS PROOF

Before we enter the proof of Proposition 1 and Theorem 5, we first state a similar set of results for Gaussian $Q_0$, which turns out to be useful for analyzing Gaussian mixture $Q_0$'s. To begin, the following lemma investigates $\mathbb{E}_{X_t \sim Q_{t|y}} \left\| \Delta_{t,y}(X_t) \right\|^2$ when $Q_0$ is Gaussian. This quantity is proportional to the asymptotic bias $\mathcal{W}_{\text{bias}}$.

**Proposition 2.** *For $Q_0 = \mathcal{N}(\mu_0, \Sigma_0)$, if $f_{t,y} = f_{t,y}^*$ in (10) and $H = (I_p \quad 0)$, with the $\alpha_t$'s according to Definition 1, Assumption 4 is satisfied, and*

$$\mathbb{E}_{X_t \sim Q_{t|y}} \left\| \Delta_{t,y}(X_t) \right\|^2 \leq \bar{\alpha}_t^2 \frac{\max\{\left\| H^\dagger y - H^\dagger H \mu_0 \right\|^2 + d(\lambda_1 + \sigma_y^2), d\}}{\min\{\lambda_d, 1\}^2 \min\{\tilde{\lambda}_{d-p}, 1\}^2} \left\| [\Sigma_0]_{y\bar{y}} [\Sigma_0]_{\bar{y}y} \right\|$$

$$\lesssim \bar{\alpha}_t^2 \cdot \left( \left\| H^\dagger y - H^\dagger H \mu_0 \right\|^2 + d \right)$$

*where $\lambda_1$ is the largest eigenvalue of $\Sigma_0$, and $\lambda_d$ and $\tilde{\lambda}_{d-p}$ are the smallest eigenvalues of $\Sigma_0$ and $[\Sigma_0]_{\bar{y}\bar{y}}$, respectively.*

*Proof.* See Appendix J.1. $\qquad \square$

With this lemma, the following theorem characterizes the conditional KL divergence when $Q_0$ is Gaussian.

**Theorem 6.** *Suppose that $\sigma_y^2 > 0$. Suppose that Assumptions 1 and 5 hold. Under the same conditions as in Proposition 2, if $\alpha_t$ further satisfies $\sum_{t=1}^T (1 - \alpha_t)\bar{\alpha}_t = 1 + o(1)$, we have*

$$\text{KL}(Q_{0|y} \| \widehat{P}_{0|y}) \lesssim \left( \left\| H^\dagger y - H^\dagger H \mu_0 \right\|^2 + d \right)$$

$$+ \left( \left\| H^\dagger y - H^\dagger H \mu_0 \right\|^2 + d \right) \frac{(\log T)^2}{T} + \sqrt{\left\| H^\dagger y - H^\dagger H \mu_0 \right\|^2 + d} \cdot (\log T)\varepsilon.$$

Note that a similar result can be obtained for $\text{KL}(Q_{1|y} \| \widehat{P}_{1|y})$ (where $\text{W}_2(Q_{1|y}, Q_{0|y})^2 \lesssim \delta d$) for any general $\sigma_y^2 \geq 0$ using the $\alpha_t$ in (8) (see Remark 1).

### I.3.1 PROOF OF THEOREM 6

Throughout the proof we use the same notations as in (55). Since Assumption 4 is satisfied from Proposition 2, in order to invoke Theorem 1, we still need to check Assumption 3. Since each $Q_{t|y}$ ($\forall t \geq 0$) is Gaussian, all partial derivatives of its log-p.d.f. higher than third-order equal zero. For the first and second-order, note that $\Sigma_{t|y} = \bar{\alpha}_t(I_d - H^\dagger H)\Sigma_0(I_d - H^\dagger H) + (1 - \bar{\alpha}_t)I_d + \bar{\alpha}_t \sigma_y^2 H^\dagger H$. Thus, when $\sigma_y^2 > 0$, $\lambda_{min}(\Sigma_{t-1|y}) \geq \min\{1 - \bar{\alpha}_t + \bar{\alpha}_t \sigma_y^2, 1 - \bar{\alpha}_t + \bar{\alpha}_t \tilde{\lambda}_d\} \geq \min\{1, \sigma_y^2, \tilde{\lambda}_d\} > 0$, which yields

$$\left\| \Sigma_{t-1|y}^{-1} \right\| \lesssim 1, \quad \forall t \geq 1. \tag{37}$$

Thus, we have, $\forall \ell \geq 1$,

$$\mathbb{E}_{Q_{t|y}} \left\| \nabla \log q_{t|y}(X_t) \right\|^\ell$$

$$= \mathbb{E}_{Q_{t|y}} \left\| \Sigma_{t|y}^{-1}(X_t - \mu_{t|y}) \right\|^{\ell} \leq \left\| \Sigma_{t|y}^{-\frac{1}{2}} \right\|^{\ell} \mathbb{E}_{Q_{t|y}} \left\| \Sigma_{t|y}^{-\frac{1}{2}}(X_t - \mu_{t|y}) \right\|^{\ell}$$

$$\lesssim d^{\ell/2} = O(1),$$

$$\mathbb{E}_{Q_{t|y}} \left\| \nabla \log q_{t-1|y}(m_{t,y}(X_t)) \right\|^{\ell}$$

$$= \mathbb{E}_{Q_{t|y}} \left\| \Sigma_{t|y}^{-1}(m_{t,y}(X_t) - \mu_{t|y}) \right\|^{\ell}$$

$$\lesssim \left\| \Sigma_{t|y}^{-\frac{1}{2}} \right\|^{\ell} \mathbb{E}_{Q_{t|y}} \left\| \Sigma_{t|y}^{-\frac{1}{2}}(X_t - \mu_{t|y}) \right\|^{\ell} + \left\| \Sigma_{t|y}^{-1} \right\|^{\ell} \mathbb{E}_{Q_{t|y}} \left\| \nabla \log q_{t|y}(X_t) \right\|^{\ell}$$

$$\lesssim d^{\ell/2} = O(1),$$

$$\mathbb{E}_{Q_{t|y}} \left\| \nabla^2 \log q_{t|y}(X_t) \right\|^{\ell} = \left\| \Sigma_{t|y}^{-1} \right\|^{\ell} = O(1),$$

$$\mathbb{E}_{Q_{t|y}} \left\| \nabla^2 \log q_{t-1|y}(m_{t,y}(X_t)) \right\|^{\ell} = \left\| \Sigma_{t-1|y}^{-1} \right\|^{\ell} = O(1).$$

Thus, Assumption 3 holds when $1 - \alpha_t$ satisfies Definition 1.

Now, we can invoke Theorem 1 and get $\mathrm{KL}(Q_{0|y} \| \widehat{P}_{0|y}) \lesssim \mathcal{W}_{\text{oracle}} + \mathcal{W}_{\text{bias}} + \mathcal{W}_{\text{vanish}}$, where

$$\mathcal{W}_{\text{oracle}} = \sum_{t=1}^{T} \frac{(1-\alpha_t)^2}{2\alpha_t} \mathbb{E}_{X_t \sim Q_{t|y}} \left[ \mathrm{Tr}\left( \nabla^2 \log q_{t-1|y}(m_{t,y}(X_t)) \nabla^2 \log q_{t|y}(X_t) \right) \right] + (\log T)\varepsilon^2$$

$$\mathcal{W}_{\text{bias}} = \sum_{t=1}^{T} (1-\alpha_t) \mathbb{E}_{X_t \sim Q_{t|y}} \left\| \Delta_{t,y}(X_t) \right\|^2$$

$$\mathcal{W}_{\text{vanish}} = \sum_{t=1}^{T} \frac{1-\alpha_t}{\sqrt{\alpha_t}} \mathbb{E}_{X_t \sim Q_{t|y}} \left[ (\nabla \log q_{t-1|y}(m_{t,y}(X_t)) - \sqrt{\alpha_t} \nabla \log q_{t|y}(X_t))^{\mathsf{T}} \Delta_{t,y}(X_t) \right]$$

$$- \sum_{t=1}^{T} \frac{(1-\alpha_t)^2}{2\alpha_t} \mathbb{E}_{X_t \sim Q_{t|y}} \left[ \Delta_{t,y}(X_t)^{\mathsf{T}} \nabla^2 \log q_{t-1|y}(m_{t,y}(X_t)) \Delta_{t,y}(X_t) \right]$$

$$+ \sum_{t=1}^{T} \frac{(1-\alpha_t)^2}{3! \alpha_t^{3/2}} \mathbb{E}_{X_t \sim Q_{t|y}} \left[ 3 \sum_{i=1}^{d} \partial_{iii}^3 \log q_{t-1|y}(m_{t,y}(X_t)) \Delta_{t,y}(X_t)^i \right.$$

$$\left. + \sum_{\substack{i,j=1 \\ i \neq j}}^{d} \partial_{iij}^3 \log q_{t-1|y}(m_{t,y}(X_t)) \Delta_{t,y}(X_t)^j \right]$$

$$+ \max_{t \geq 1} \sqrt{\mathbb{E}_{X_t \sim Q_{t|y}} \left\| \Delta_{t,y}(X_t) \right\|^2} (\log T)\varepsilon.$$

We first consider the estimation error (in both $\mathcal{W}_{\text{oracle}}$ and $\mathcal{W}_{\text{vanish}}$), which can be upper-bounded as

$$\max_{t \geq 1} \sqrt{\mathbb{E}_{X_t \sim Q_{t|y}} \left\| \Delta_{t,y}(X_t) \right\|^2} (\log T)\varepsilon + (\log T)\varepsilon^2 \lesssim \left( \left\| H^\dagger y - H^\dagger H \mu_0 \right\|^2 + d \right)^{\frac{1}{2}} (\log T)\varepsilon$$

from Proposition 2. Also, when $Q_{t|y}$ is Gaussian, we can calculate, for any $x_t \in \mathbb{R}^d$,

$$\mathrm{Tr}\left( \nabla^2 \log q_{t-1|y}(m_{t,y}(x_t)) \nabla^2 \log q_{t|y}(x_t) \right) = \mathrm{Tr}(\Sigma_{t-1|y}^{-1} \Sigma_{t|y}^{-1})$$

$$= \mathrm{Tr}(\Sigma_{t-1|y}^{-1}(\alpha_t \Sigma_{t-1|y} + (1-\alpha_t)I_d)^{-1})$$

$$\stackrel{(i)}{=} \mathrm{Tr}(\Sigma_{t-1|y}^{-1}(\alpha_t^{-1}\Sigma_{t-1|y}^{-1} - \frac{1-\alpha_t}{\alpha_t^2}\Sigma_{t-1|y}^{-2} + O((1-\alpha_t)^2)))$$

$$\lesssim \frac{1}{\alpha_t}\mathrm{Tr}(\Sigma_{t-1|y}^{-2})$$

where $(i)$ follows from Taylor expansion when $1 - \alpha_t$ is small. Using (37), this implies that

$$\mathcal{W}_{\text{oracle}} \lesssim \frac{d(\log T)^2}{T} + (\log T)\varepsilon^2.$$

Also, from the condition on $\alpha_t$,

$$\sum_{t=1}^{T}(1-\alpha_t)\mathbb{E}_{X_t \sim Q_{t|y}}\|\Delta_{t,y}(X_t)\|^2 \lesssim (\|H^\dagger y - H^\dagger H\mu_0\|^2 + d).$$

Now we focus on $\mathcal{W}_{\text{vanish}}$ (except the estimation error). Since $Q_{t|y}$ is Gaussian, all third-order partial derivatives are zero, and only the first two terms in $\mathcal{W}_{\text{vanish}}$ remain. In the following we fix $t \geq 1$. Also recall from (56) that when $H = (I_p \quad 0)$,

$$\begin{aligned}
\Delta_{t,y} &= -\bar{\alpha}_t(I_d - H^\dagger H)\Sigma_{t,sig}^{-1}(I_d - H^\dagger H)\Sigma_0(H^\dagger H)\Sigma_t^{-1}(x_t - \sqrt{\bar{\alpha}_t}\mu_0) \\
&= -\bar{\alpha}_t(\bar{\alpha}_t[\Sigma_0]_{\bar{y}\bar{y}} + (1-\bar{\alpha}_t)I_{d-p})^{-1}[\Sigma_0]_{\bar{y}y}[\Sigma_t^{-1}]_{y:}(x_t - \sqrt{\bar{\alpha}_t}\mu_0).
\end{aligned}$$

For the first term of $\mathcal{W}_{\text{vanish}}$, we first calculate for each $x_t$ that

$$\begin{aligned}
&\nabla \log q_{t-1|y}(m_{t,y}) - \sqrt{\alpha_t}\nabla \log q_{t|y}(x_t) \\
&= \sqrt{\alpha_t}\Sigma_{t|y}^{-1}(x_t - \sqrt{\bar{\alpha}_t}\mu_{0|y}) - \Sigma_{t-1|y}^{-1}(m_{t,y} - \sqrt{\bar{\alpha}_{t-1}}\mu_{0|y})
\end{aligned}$$

Recall that

$$m_{t,y} = \frac{1}{\sqrt{\alpha_t}}x_t + \frac{1-\alpha_t}{\sqrt{\alpha_t}}\nabla \log q_{t|y}(x_t) = \frac{1}{\sqrt{\alpha_t}}x_t - \frac{1-\alpha_t}{\sqrt{\alpha_t}}\Sigma_{t|y}^{-1}(x_t - \sqrt{\bar{\alpha}_t}\mu_{0|y}),$$

$$\Sigma_{t|y}^{-1} = (\alpha_t\Sigma_{t-1|y} + (1-\alpha_t)I_d)^{-1} = \frac{1}{\alpha_t}\Sigma_{t-1|y}^{-1} - \frac{1-\alpha_t}{\alpha_t^2}\Sigma_{t-1|y}^{-2} + O((1-\alpha_t)^2).$$

Thus,

$$\begin{aligned}
&\nabla \log q_{t-1|y}(m_{t,y}) - \sqrt{\alpha_t}\nabla \log q_{t|y}(x_t) \\
&= \sqrt{\alpha_t}\left(\frac{1}{\alpha_t}\Sigma_{t-1|y}^{-1} - \frac{1-\alpha_t}{\alpha_t^2}\Sigma_{t-1|y}^{-2}\right)(x_t - \sqrt{\bar{\alpha}_t}\mu_{0|y}) \\
&\quad - \Sigma_{t-1|y}^{-1}\left(\frac{1}{\sqrt{\alpha_t}}x_t + \frac{1-\alpha_t}{\sqrt{\alpha_t}}\nabla \log q_{t|y}(x_t) - \sqrt{\bar{\alpha}_{t-1}}\mu_{0|y}\right) + O((1-\alpha_t)^2) \\
&= -\frac{1-\alpha_t}{\alpha_t^{3/2}}\Sigma_{t-1|y}^{-2}(x_t - \sqrt{\bar{\alpha}_t}\mu_{0|y}) + \frac{1-\alpha_t}{\sqrt{\alpha_t}}\Sigma_{t-1|y}^{-1}\Sigma_{t|y}^{-1}(x_t - \sqrt{\bar{\alpha}_t}\mu_{0|y}) + O((1-\alpha_t)^2).
\end{aligned}$$

Combining with the definition for $\Delta_{t,y}$ in (56) and using Lemma 9, we have

$$\mathbb{E}_{X_t \sim Q_{t|y}}\left[\Delta_{t,y}(X_t)^\intercal(\nabla \log q_{t-1|y}(m_{t,y}(X_t)) - \sqrt{\alpha_t}\nabla \log q_{t|y}(X_t))\right]$$

$$= \bar{\alpha}_t\mathbb{E}_{X_t \sim Q_{t|y}}\left[(X_t - \sqrt{\bar{\alpha}_t}\mu_0)^\intercal\Sigma_t^{-1}(H^\dagger H)\Sigma_0(I_d - H^\dagger H)\Sigma_{t,sig}^{-1}(I_d - H^\dagger H)\right.$$

$$\left.\left(\frac{1-\alpha_t}{\alpha_t^{3/2}}\Sigma_{t-1|y}^{-2} - \frac{1-\alpha_t}{\sqrt{\alpha_t}}\Sigma_{t-1|y}^{-1}\Sigma_{t|y}^{-1}\right)(X_t - \sqrt{\bar{\alpha}_t}\mu_{0|y})\right] + O((1-\alpha_t)^2)$$

$$\overset{(ii)}{=} \bar{\alpha}_t\mathbb{E}_{X_t \sim Q_{t|y}}\left[(X_t - \sqrt{\bar{\alpha}_t}\mu_0)^\intercal\Sigma_t^{-1}(H^\dagger H)\Sigma_0(I_d - H^\dagger H)\Sigma_{t,sig}^{-1}(I_d - H^\dagger H)\right.$$

$$\left(\frac{1-\alpha_t}{\alpha_t^{3/2}}\Sigma_{t-1|y}^{-1}(I_d - H^\dagger H)\Sigma_{t-1|y}^{-1} - \frac{1-\alpha_t}{\sqrt{\alpha_t}}\Sigma_{t-1|y}^{-1}(I_d - H^\dagger H)\Sigma_{t|y}^{-1}\right)$$

$$\left.(I_d - H^\dagger H)(X_t - \sqrt{\bar{\alpha}_t}\mu_0)\right] + O((1-\alpha_t)^2)$$

$$= \bar{\alpha}_t\mathbb{E}_{X_t \sim Q_{t|y}}\left[(X_t - \sqrt{\bar{\alpha}_t}\mu_0)^\intercal[\Sigma_t^{-1}]_{:y}[\Sigma_0]_{y\bar{y}}(\bar{\alpha}_t[\Sigma_0]_{\bar{y}\bar{y}} + (1-\bar{\alpha}_t)I_{d-p})^{-1}\right.$$

$$\left(\frac{1-\alpha_t}{\alpha_t^{3/2}}[\Sigma_{t-1|y}^{-1}]_{\bar{y}\bar{y}}^2 - \frac{1-\alpha_t}{\sqrt{\alpha_t}}[\Sigma_{t-1|y}^{-1}]_{\bar{y}\bar{y}}[\Sigma_{t|y}^{-1}]_{\bar{y}\bar{y}}\right)(0 \quad I_{d-p})(X_t - \sqrt{\bar{\alpha}_t}\mu_0)\right]$$

$$+ O((1-\alpha_t)^2)$$

$$= \bar{\alpha}_t \mathrm{Tr}\Bigg( [\Sigma_t^{-1}]_{:y}[\Sigma_0]_{y\bar{y}}(\bar{\alpha}_t[\Sigma_0]_{\bar{y}\bar{y}} + (1-\bar{\alpha}_t)I_{d-p})^{-1}\left( \frac{1-\alpha_t}{\alpha_t^{3/2}}[\Sigma_{t-1|y}^{-1}]_{\bar{y}\bar{y}}^2 - \frac{1-\alpha_t}{\sqrt{\alpha_t}}[\Sigma_{t-1|y}^{-1}]_{\bar{y}\bar{y}}[\Sigma_{t|y}^{-1}]_{\bar{y}\bar{y}} \right)$$

$$(0 \quad I_{d-p})\,\mathbb{E}_{X_t \sim Q_{t|y}}\left[ (X_t - \sqrt{\bar{\alpha}_t}\mu_0)(X_t - \sqrt{\bar{\alpha}_t}\mu_0)^\intercal \right] \Bigg)$$

$$+ O((1-\alpha_t)^2)$$

$$\overset{(iii)}{\leq} \left\| [\Sigma_t^{-1}]_{:y} \right\| \left\| [\Sigma_0]_{y\bar{y}} \right\| \left\| (\bar{\alpha}_t[\Sigma_0]_{\bar{y}\bar{y}} + (1-\bar{\alpha}_t)I_{d-p})^{-1} \right\| \times$$

$$\left( \frac{1-\alpha_t}{\alpha_t^{3/2}} \left\| [\Sigma_{t-1|y}^{-1}]_{\bar{y}\bar{y}} \right\|^2 + \frac{1-\alpha_t}{\sqrt{\alpha_t}} \left\| [\Sigma_{t-1|y}^{-1}]_{\bar{y}\bar{y}} \right\| \left\| [\Sigma_{t|y}^{-1}]_{\bar{y}\bar{y}} \right\| \right) \times$$

$$\mathrm{Tr}\left( \mathbb{E}_{X_t \sim Q_{t|y}}\left[ (X_t - \sqrt{\bar{\alpha}_t}\mu_0)(X_t - \sqrt{\bar{\alpha}_t}\mu_0)^\intercal \right] \right)$$

$$\overset{(iv)}{\lesssim} \left( \frac{1-\alpha_t}{\alpha_t^{3/2}} + \frac{1-\alpha_t}{\sqrt{\alpha_t}} \right) \max\left\{ \left\| H^\dagger y - H^\dagger H \mu_0 \right\|^2 + d(\lambda_1 + \sigma_y^2), d \right\}$$

$$\lesssim (1-\alpha_t)\left( \left\| H^\dagger y - H^\dagger H \mu_0 \right\|^2 + d \right)$$

where $(ii)$ follows by Lemma 9 and from definition that $(I_d - H^\dagger H)\mu_{0|y} = (I_d - H^\dagger H)\mu_0$, $(iii)$ follows because $|\mathrm{Tr}(UV)| \leq \|U\| \mathrm{Tr}(V)$ if $V$ is positive semi-definite, and $(iv)$ follows from (58) and the same reasons for (57). In particular, we note that $[\Sigma_{t|y}^{-1}]_{\bar{y}\bar{y}} = [\Sigma_{t,sig}^{-1}]_{\bar{y}\bar{y}} = (\bar{\alpha}_t[\Sigma_0]_{\bar{y}\bar{y}} + (1-\bar{\alpha}_t)I_{d-p})^{-1}$ and $\left\| (\bar{\alpha}_t[\Sigma_0]_{\bar{y}\bar{y}} + (1-\bar{\alpha}_t)I_{d-p})^{-1} \right\| \leq \frac{1}{\min\{\tilde{\lambda}_{d-p}, 1\}} < \infty$.

For the second term of $\mathcal{W}_{\text{vanish}}$, we use the fact that $[\Sigma_{t-1|y}^{-1}]_{\bar{y}\bar{y}} = (\bar{\alpha}_{t-1}[\Sigma_0]_{\bar{y}\bar{y}} + (1-\bar{\alpha}_{t-1})I_{d-p})^{-1}$ and have

$$- \mathbb{E}_{X_t \sim Q_{t|y}} \Delta_{t,y}(X_t)^\intercal \nabla^2 \log q_{t-1|y}(m_{t,y}(X_t))\Delta_{t,y}(X_t)$$

$$= \bar{\alpha}_t^2 \mathbb{E}_{X_t \sim Q_{t|y}} \Big[ (X_t - \sqrt{\bar{\alpha}_t}\mu_0)^\intercal \Sigma_t^{-1}(H^\dagger H)\Sigma_0(I_d - H^\dagger H)\Sigma_{t,sig}^{-1}(I_d - H^\dagger H)\Sigma_{t-1|y}^{-1}$$

$$(I_d - H^\dagger H)\Sigma_{t,sig}^{-1}(I_d - H^\dagger H)\Sigma_0(H^\dagger H)\Sigma_t^{-1}(X_t - \sqrt{\bar{\alpha}_t}\mu_0) \Big]$$

$$= \bar{\alpha}_t^2 \mathrm{Tr}\Bigg( [\Sigma_t^{-1}]_{:y}[\Sigma_0]_{y\bar{y}}(\bar{\alpha}_t[\Sigma_0]_{\bar{y}\bar{y}} + (1-\bar{\alpha}_t)I_{d-p})^{-1}(\bar{\alpha}_{t-1}[\Sigma_0]_{\bar{y}\bar{y}} + (1-\bar{\alpha}_{t-1})I_{d-p})^{-1}$$

$$(\bar{\alpha}_t[\Sigma_0]_{\bar{y}\bar{y}} + (1-\bar{\alpha}_t)I_{d-p})^{-1}[\Sigma_0]_{\bar{y}y}[\Sigma_t^{-1}]_{y:}\,\mathbb{E}_{X_t \sim Q_{t|y}}(X_t - \sqrt{\bar{\alpha}_t}\mu_0)(X_t - \sqrt{\bar{\alpha}_t}\mu_0)^\intercal \Bigg)$$

$$\overset{(v)}{\leq} \max\left\{ \left\| H^\dagger y - H^\dagger H \mu_0 \right\|^2 + d(\lambda_1 + \sigma_y^2), d \right\} \left\| (\bar{\alpha}_t[\Sigma_0]_{\bar{y}\bar{y}} + (1-\bar{\alpha}_t)I_{d-p})^{-1} \right\|^2$$

$$\left\| (\bar{\alpha}_{t-1}[\Sigma_0]_{\bar{y}\bar{y}} + (1-\bar{\alpha}_{t-1})I_{d-p})^{-1} \right\| \left\| \Sigma_t^{-1} \right\|^2 \left\| [\Sigma_0]_{y\bar{y}}[\Sigma_0]_{\bar{y}y} \right\|$$

$$\lesssim \max\left\{ \left\| H^\dagger y - H^\dagger H \mu_0 \right\|^2 + d(\lambda_1 + \sigma_y^2), d \right\}$$

$$\lesssim \left\| H^\dagger y - H^\dagger H \mu_0 \right\|^2 + d.$$

Here $(v)$ follows from the fact that $|\mathrm{Tr}(UV)| \leq \|U\| \mathrm{Tr}(V)$ if $V$ is positive semi-definite and (58). The proof is complete by plugging all the results above into Theorem 1.

*Remark* 1. Before we end the proof, we leave a note for the case of $\sigma_y^2 = 0$ (indeed, for any general $\sigma_y^2 \geq 0$). The only difference is how to upper-bound $\mathcal{W}_{\text{oracle}}$. In particular, if $\sigma_y^2 = 0$, (37) no longer holds (i.e., we can no longer upper-bound $\left\| \Sigma_{t-1|y}^{-1} \right\|$ as a constant). Instead, we can obtain an upper bound as $\left\| \Sigma_{t-1|y}^{-1} \right\| \lesssim (1-\bar{\alpha}_{t-1})^{-1}$. Then, with the $\alpha_t$ in (8), we have

$$\mathcal{W}_{\text{oracle}} \lesssim \frac{d(\log T)^2 \log(1/\delta)^2}{T} + (\log T)\varepsilon^2.$$

The rest of the proof still follows because the $\alpha_t$ satisfies Definition 1 when $t \geq 2$. Combining with Lemma 7, we would finally obtain

$$
\mathrm{KL}(Q_{1|y}\|\widehat{P}_{1|y}) \lesssim (\left\|H^\dagger y - H^\dagger H \mu_0\right\|^2 + d)\left(1 - \frac{3\log(1/\delta)\log T}{T}\right)
$$

$$
+ (\left\|H^\dagger y - H^\dagger H \mu_0\right\|^2 + d)\frac{(\log T)^2 \log(1/\delta)^2}{T} + \sqrt{\left\|H^\dagger y - H^\dagger H \mu_0\right\|^2 + d} \cdot (\log T)\varepsilon.
$$

Here $\mathrm{W}_2(Q_{1|y}, Q_{0|y})^2 \lesssim \delta d$.

## I.4 PROOF OF PROPOSITION 1

We first introduce some useful notations for this subsection. Recall that $Q_0$ has mixture p.d.f. in which the mixture prior $\pi_n$ is independent of $y$ ($= Hx_0 + n$). Thus, using the fact that $x_0 = (I_d - H^\dagger H)x_0 + H^\dagger y - H^\dagger n$, we can define $Q_{0,n|y}$ as (cf. Flåm (2013))

$$
Q_{0|y} = \sum_{n=1}^N \pi_n Q_{0,n|y}
$$

$$
:= \sum_{n=1}^N \pi_n \mathcal{N}((I_d - H^\dagger H)\mu_{0,n} + H^\dagger y, (I_d - H^\dagger H)\Sigma_0(I_d - H^\dagger H) + \sigma_y^2 H^\dagger (H^\dagger)^\intercal).
$$

Note that when $H = (I_p \quad 0)$ and $\sigma_y^2 > 0$, $q_{0|y}$ exists. From the conditional forward model in (5), we further define

$$
Q_t = \sum_{n=1}^N \pi_n Q_{t,n}, \quad Q_{t|y} = \sum_{n=1}^N \pi_n Q_{t,n|y}, \quad Q_{t,n} := \mathcal{N}(\mu_{t,n}, \Sigma_t), \quad Q_{t,n|y} := \mathcal{N}(\mu_{t,n|y}, \Sigma_{t|y})
$$

$$
\mu_{t,n} := \sqrt{\bar{\alpha}_t}\mu_{0,n}, \quad \Sigma_t := \bar{\alpha}_t \Sigma_0 + (1 - \bar{\alpha}_t)I_d, \quad \mu_{t,n|y} := \sqrt{\bar{\alpha}_t}(I_d - H^\dagger H)\mu_{0,n} + \sqrt{\bar{\alpha}_t}H^\dagger y
$$

$$
\Sigma_{t|y} := \Sigma_{t,sig} + \bar{\alpha}_t\sigma_y^2 H^\dagger (H^\dagger)^\intercal, \quad \Sigma_{t,sig} := \bar{\alpha}_t(I_d - H^\dagger H)\Sigma_0(I_d - H^\dagger H) + (1 - \bar{\alpha}_t)I_d. \quad (38)
$$

Similar to (37), we still have

$$
\left\|\Sigma_{t-1|y}^{-1}\right\| \lesssim 1, \quad \forall t \geq 1.
$$

We can also calculate the scores of $Q_t$ and $Q_{t|y}$ in as follows.

$$
\nabla \log q_t(x_t) = -\frac{1}{q_t(x_t)}\sum_{n=1}^N \pi_n q_{t,n}(x_t)\Sigma_t^{-1}(x_t - \mu_{t,n}),
$$

$$
\nabla \log q_{t|y}(x_t) = -\frac{1}{q_{t|y}(x_t)}\sum_{n=1}^N \pi_n q_{t,n|y}(x_t)\Sigma_{t|y}^{-1}(x_t - \mu_{t,n|y}). \quad (39)
$$

Now, with $f_{t,y} = f_{t,y}^*$ (in (10)), from the expression of $\Delta_{t,y}$ in (32), under the assumption $H = (I_p \quad 0)$, the score mismatch at each diffusion step is equal to

$$
\Delta_{t,y} = (I_d - H^\dagger H)(\nabla \log q_{t|y} - \nabla \log q_t)
$$

$$
= \frac{1}{q_t(x_t)}\sum_{n=1}^N \pi_n q_{t,n}(x_t)(I_d - H^\dagger H)\Sigma_t^{-1}(x_t - \sqrt{\bar{\alpha}_t}\mu_{0,n})
$$

$$
- \frac{1}{q_{t|y}(x_t)}\sum_{n=1}^N \pi_n q_{t,n|y}(x_t)(I_d - H^\dagger H)\Sigma_{t|y}^{-1}(x_t - \sqrt{\bar{\alpha}_t}\mu_{0,n|y})
$$

$$
= \sum_{n=1}^N \pi_n \left(\frac{q_{t,n}(x_t)}{q_t(x_t)} - \frac{q_{t,n|y}(x_t)}{q_{t|y}(x_t)}\right)(I_d - H^\dagger H)\Sigma_t^{-1}(x_t - \sqrt{\bar{\alpha}_t}\mu_{0,n})
$$

$$
+ \sum_{n=1}^N \pi_n \frac{q_{t,n|y}(x_t)}{q_{t|y}(x_t)}(I_d - H^\dagger H)\left(\Sigma_t^{-1}(x_t - \sqrt{\bar{\alpha}_t}\mu_{0,n}) - \Sigma_{t|y}^{-1}(x_t - \sqrt{\bar{\alpha}_t}\mu_{0,n|y})\right)
$$

$$= -\sqrt{\bar{\alpha}_t} \sum_{n=1}^{N} \pi_n \left( \frac{q_{t,n}(x_t)}{q_t(x_t)} - \frac{q_{t,n|y}(x_t)}{q_{t|y}(x_t)} \right) (I_d - H^\dagger H) \Sigma_t^{-1} \mu_{0,n}$$

$$+ \sum_{n=1}^{N} \pi_n \frac{q_{t,n|y}(x_t)}{q_{t|y}(x_t)} (I_d - H^\dagger H) \left( \Sigma_t^{-1}(x_t - \sqrt{\bar{\alpha}_t} \mu_{0,n}) - \Sigma_{t|y}^{-1}(x_t - \sqrt{\bar{\alpha}_t} \mu_{0,n|y}) \right)$$

$$\overset{(i)}{=} -\sqrt{\bar{\alpha}_t} \sum_{n=1}^{N} \pi_n \left( \frac{q_{t,n}(x_t)}{q_t(x_t)} - \frac{q_{t,n|y}(x_t)}{q_{t|y}(x_t)} \right) (I_d - H^\dagger H) \Sigma_t^{-1} \mu_{0,n}$$

$$- \bar{\alpha}_t \sum_{n=1}^{N} \pi_n \frac{q_{t,n|y}(x_t)}{q_{t|y}(x_t)} A_t \Sigma_0 (H^\dagger H) \Sigma_t^{-1}(x_t - \sqrt{\bar{\alpha}_t} \mu_{0,n}) \tag{40}$$

where $A_t := (I_d - H^\dagger H) \Sigma_{t,sig}^{-1} (I_d - H^\dagger H)$. Here $(i)$ follows from similar arguments as in (56). Note that since $H = (I_p \quad 0)$, we have equivalently $A_t = (I_d - H^\dagger H) \Sigma_{t|y}^{-1} (I_d - H^\dagger H)$. Since $H^\dagger H = \begin{pmatrix} I_p & 0 \\ 0 & 0 \end{pmatrix}$, we can also re-express the second term in $\Delta_{t,y}$ such that $[\Delta_{t,y}]_y = 0$ and

$$[\Delta_{t,y}]_{\bar{y}} = -\sqrt{\bar{\alpha}_t} \sum_{n=1}^{N} \pi_n \left( \frac{q_{t,n}(x_t)}{q_t(x_t)} - \frac{q_{t,n|y}(x_t)}{q_{t|y}(x_t)} \right) [\Sigma_t^{-1}]_{\bar{y}:} \mu_{0,n}$$

$$- \bar{\alpha}_t \sum_{n=1}^{N} \pi_n \frac{q_{t,n|y}(x_t)}{q_{t|y}(x_t)} (\bar{\alpha}_t [\Sigma_0]_{\bar{y}\bar{y}} + (1 - \bar{\alpha}_t) I_{d-p})^{-1} [\Sigma_0]_{\bar{y}y} [\Sigma_t^{-1}]_{y:} (x_t - \sqrt{\bar{\alpha}_t} \mu_{0,n})$$

$$\tag{41}$$

since when $H^\dagger H = \begin{pmatrix} I_p & 0 \\ 0 & 0 \end{pmatrix}$, $A_t = \begin{pmatrix} 0 & 0 \\ 0 & (\bar{\alpha}_t [\Sigma_0]_{\bar{y}\bar{y}} + (1 - \bar{\alpha}_t) I_{d-p})^{-1} \end{pmatrix}$.

Now, for the second moment, we follow similar analyses in (57) and get

$$\mathbb{E}_{X_t \sim Q_{t|y}} \|\Delta_{t,y}\|^2$$

$$\leq 4\bar{\alpha}_t \mathbb{E}_{X_t \sim Q_{t|y}} \max_{n \in [N]} \left\| \Sigma_t^{-1} \mu_{0,n} \right\|^2$$

$$+ 2\bar{\alpha}_t^2 \left\| (\bar{\alpha}_t [\Sigma_0]_{\bar{y}\bar{y}} + (1 - \bar{\alpha}_t) I_{d-p})^{-1} \right\|^2 \left\| [\Sigma_t^{-1}]_{y:} \right\|^2 \left\| [\Sigma_0]_{y\bar{y}} \right\|^2 \times$$

$$\mathbb{E}_{X_t \sim Q_{t|y}} \left[ \sum_{n=1}^{N} \pi_n \frac{q_{t,n|y}(X_t)}{q_{t|y}(X_t)} \left\| X_t - \sqrt{\bar{\alpha}_t} \mu_{0,n} \right\|^2 \right]$$

where

$$\mathbb{E}_{X_t \sim Q_{t|y}} \left[ \sum_{n=1}^{N} \pi_n \frac{q_{t,n|y}(X_t)}{q_{t|y}(X_t)} \left\| X_t - \sqrt{\bar{\alpha}_t} \mu_{0,n} \right\|^2 \right]$$

$$= \mathbb{E}_{X_t \sim Q_{t|y}} \mathbb{E}_{N \sim \Pi_{\cdot|t,y}} \left\| X_t - \sqrt{\bar{\alpha}_t} \mu_{0,N} \right\|^2$$

$$= \mathbb{E}_{N \sim \Pi_{\cdot|y}} \mathbb{E}_{X_t \sim Q_{t,N|y}} \left\| X_t - \sqrt{\bar{\alpha}_t} \mu_{0,N} \right\|^2$$

$$\overset{(ii)}{=} \mathbb{E}_{N \sim \Pi_{\cdot|y}} \left[ \text{Tr}(\Sigma_{t|y}) + \bar{\alpha}_t \left\| H^\dagger y - H^\dagger H \mu_{0,N} \right\|^2 \right]$$

$$= \text{Tr}(\Sigma_{t|y}) + \bar{\alpha}_t \sum_{n=1}^{N} \pi_n \left\| H^\dagger y - H^\dagger H \mu_{0,n} \right\|^2$$

where $(ii)$ follows from (58) and note that $Q_{t,N|y}$ is Gaussian for each $N = n$. Denote $\lambda_1 \geq \cdots \geq \lambda_d > 0$ and $\tilde{\lambda}_1 \geq \cdots \geq \tilde{\lambda}_{d-p} > 0$ to be the eigenvalues of $\Sigma_0$ and $[\Sigma_0]_{\bar{y}\bar{y}}$, respectively. Similarly as the proof of Proposition 2, we have $\|[\Sigma_0]_{y\bar{y}}\| \leq \|\Sigma_0\| = \lambda_1$, $\left\| [\Sigma_t^{-1}]_{y:} \right\| \leq \left\| \Sigma_t^{-1} \right\| \leq (\bar{\alpha}_t \lambda_d + (1 - \bar{\alpha}_t))^{-1} \leq \frac{1}{\min\{\lambda_d, 1\}}$, $\left\| (\bar{\alpha}_t [\Sigma_0]_{\bar{y}\bar{y}} + (1 - \bar{\alpha}_t) I_{d-p})^{-1} \right\| \leq \frac{1}{\bar{\alpha}_t \tilde{\lambda}_{d-p} + (1 - \bar{\alpha}_t)} \leq \frac{1}{\min\{\tilde{\lambda}_{d-p}, 1\}}$, and $\left\| \Sigma_{t|y} \right\| \leq \bar{\alpha}_t (\lambda_1 + \sigma_y^2) + (1 - \bar{\alpha}_t)$. Therefore,

$$\mathbb{E}_{X_t \sim Q_{t|y}} \|\Delta_{t,y}\|^2$$

$$\lesssim \bar{\alpha}_t d + \bar{\alpha}_t^2 \frac{\|[\Sigma_0]_{y\bar{y}}\|^2}{\min\{\tilde{\lambda}_{d-p}, 1\}^2 \min\{\lambda_d, 1\}^2} \times$$

$$\left( d(1 - \bar{\alpha}_t) + \bar{\alpha}_t d(\lambda_1 + \sigma_y^2) + \bar{\alpha}_t \sum_{n=1}^{N} \pi_n \left\| H^\dagger y - H^\dagger H \mu_{0,n} \right\|^2 \right)$$

$$\lesssim \bar{\alpha}_t d + \bar{\alpha}_t^2 \frac{\|[\Sigma_0]_{y\bar{y}}\|^2}{\min\{\tilde{\lambda}_{d-p}, 1\}^2 \min\{\lambda_d, 1\}^2} \max \left\{ d(\lambda_1 + \sigma_y^2) + \sum_{n=1}^{N} \pi_n \left\| H^\dagger y - H^\dagger H \mu_{0,n} \right\|^2, d \right\}.$$

The proof is complete.

## I.5 PROOF OF THEOREM 5

We first recall all the notations in (38) under Gaussian mixture. We also recall the scores from (39):

$$\nabla \log q_t(x_t) = -\frac{1}{q_t(x_t)} \sum_{n=1}^{N} \pi_n q_{t,n}(x_t) \Sigma_t^{-1}(x_t - \mu_{t,n})$$

$$= -\Sigma_t^{-1} x_t + \frac{1}{q_t(x_t)} \sum_{n=1}^{N} \pi_n q_{t,n}(x_t) \Sigma_t^{-1} \mu_{t,n}$$

$$\nabla \log q_{t|y}(x_t) = -\frac{1}{q_{t|y}(x_t)} \sum_{n=1}^{N} \pi_n q_{t,n|y}(x_t) \Sigma_{t|y}^{-1}(x_t - \mu_{t,n|y})$$

$$= -\Sigma_{t|y}^{-1} x_t + \frac{1}{q_{t|y}(x_t)} \sum_{n=1}^{N} \pi_n q_{t,n|y}(x_t) \Sigma_{t|y}^{-1} \mu_{t,n|y}$$

Also, we recall the explicit expression of $\Delta_{t,y}$ from (41), such that $[\Delta_{t,y}]_y = 0$ and

$$[\Delta_{t,y}]_{\bar{y}} = -\sqrt{\bar{\alpha}_t} \sum_{n=1}^{N} \pi_n \left( \frac{q_{t,n}(x_t)}{q_t(x_t)} - \frac{q_{t,n|y}(x_t)}{q_{t|y}(x_t)} \right) [\Sigma_t^{-1}]_{\bar{y}:} \mu_{0,n}$$

$$- \bar{\alpha}_t \sum_{n=1}^{N} \pi_n \frac{q_{t,n|y}(x_t)}{q_{t|y}(x_t)} (\bar{\alpha}_t[\Sigma_0]_{\bar{y}\bar{y}} + (1 - \bar{\alpha}_t)I_{d-p})^{-1} [\Sigma_0]_{\bar{y}y} [\Sigma_t^{-1}]_{y:} (x_t - \sqrt{\bar{\alpha}_t} \mu_{0,n}).$$

In order to invoke Theorem 1, we need to check Assumptions 3 and 4. From (Liang et al., 2024, Lemmas 13 and 14), since $\left\| \Sigma_{t|y}^{-1} \right\| \lesssim 1$ for all $t \geq 0$, the absolute values of any-order partial derivative are bounded by $O(1)$ in expectation, and thus Assumption 3 is satisfied. The following lemma verifies Assumption 4 using the $\alpha_t$ in Definition 1.

**Lemma 8.** *Under the same condition of Theorem 5, Assumption 4 holds if the $\alpha_t$ satisfies Definition 1.*

*Proof.* See Appendix J.2. □

Now we start to upper-bound the conditional KL-divergence of interest. Recall that from Theorem 1, $\mathrm{KL}(Q_{0|y} \| \widehat{P}_{0|y}) \lesssim \mathcal{W}_{\text{oracle}} + \mathcal{W}_{\text{bias}} + \mathcal{W}_{\text{vanish}}$, where

$$\mathcal{W}_{\text{oracle}} = \sum_{t=1}^{T} \frac{(1 - \alpha_t)^2}{2\alpha_t} \mathbb{E}_{X_t \sim Q_{t|y}} \left[ \mathrm{Tr} \left( \nabla^2 \log q_{t-1|y}(m_{t,y}(X_t)) \nabla^2 \log q_{t|y}(X_t) \right) \right] + (\log T)\varepsilon^2$$

$$\mathcal{W}_{\text{bias}} = \sum_{t=1}^{T} (1 - \alpha_t) \mathbb{E}_{X_t \sim Q_{t|y}} \left\| \Delta_{t,y}(X_t) \right\|^2$$

$$\mathcal{W}_{\text{vanish}} = \sum_{t=1}^{T} \frac{1 - \alpha_t}{\sqrt{\alpha_t}} \mathbb{E}_{X_t \sim Q_{t|y}} \left[ (\nabla \log q_{t-1|y}(m_{t,y}(X_t)) - \sqrt{\alpha_t} \nabla \log q_{t|y}(X_t))^\mathsf{T} \Delta_{t,y}(X_t) \right]$$

$$- \sum_{t=1}^{T} \frac{(1 - \alpha_t)^2}{2\alpha_t} \mathbb{E}_{X_t \sim Q_{t|y}} \left[ \Delta_{t,y}(X_t)^\mathsf{T} \nabla^2 \log q_{t-1|y}(m_{t,y}(X_t)) \Delta_{t,y}(X_t) \right]$$

$$+ \sum_{t=1}^{T} \frac{(1-\alpha_t)^2}{3!\alpha_t^{3/2}} \mathbb{E}_{X_t \sim Q_{t|y}} \left[ 3 \sum_{i=1}^{d} \partial_{iii}^3 \log q_{t-1|y}(m_{t,y}(X_t)) \Delta_{t,y}(X_t)^i \right.$$

$$\left. + \sum_{\substack{i,j=1 \\ i \neq j}}^{d} \partial_{iij}^3 \log q_{t-1|y}(m_{t,y}(X_t)) \Delta_{t,y}(X_t)^j \right]$$

$$+ \max_{t \geq 1} \sqrt{\mathbb{E}_{X_t \sim Q_{t|y}} \|\Delta_{t,y}(X_t)\|^2} (\log T)\varepsilon.$$

From (Liang et al., 2024, Theorem 2) (and by assumption $N \leq d$), if the $\alpha_t$ satisfies Definition 1,

$$\mathcal{W}_{\text{oracle}} \lesssim \frac{d^2 (\log T)^2}{T} + (\log T)\varepsilon^2.$$

Also, from Proposition 1, under the assumption on $\alpha_t$, $\mathcal{W}_{\text{bias}}$ can be upper-bounded as

$$\mathcal{W}_{\text{bias}} \lesssim d + \sum_{n=1}^{N} \pi_n \left\| H^\dagger y - H^\dagger H \mu_{0,n} \right\|^2.$$

Among the terms in $\mathcal{W}_{\text{vanish}}$, the last estimation error term can be upper-bounded using Proposition 1 as

$$\max_{t \geq 1} \sqrt{\mathbb{E}_{X_t \sim Q_{t|y}} \|\Delta_{t,y}(X_t)\|^2} (\log T)\varepsilon \lesssim \sqrt{d + \sum_{n=1}^{N} \pi_n \left\| H^\dagger y - H^\dagger H \mu_{0,n} \right\|^2} (\log T)\varepsilon.$$

It remains to analyze the rest of the terms in $\mathcal{W}_{\text{vanish}}$. In the following we fix $t \geq 1$. We remind readers of the notations in (38). For the first term in $\mathcal{W}_{\text{vanish}}$, we first provide the following useful calculations. Note that by exchanging the order of expectation, for any function fn we have

$$\mathbb{E}_{X_t \sim Q_{t|y}} \left[ \sum_{n=1}^{N} \pi_n \frac{q_{t,n|y}(X_t)}{q_{t|y}(X_t)} \text{fn}(X_t, n) \right] = \sum_{n=1}^{N} \pi_n \mathbb{E}_{X_t \sim Q_{t,n|y}} \text{fn}(X_t, n).$$

Thus,

$$\mathbb{E}_{X_t \sim Q_{t|y}} \sum_{n=1}^{N} \pi_n \frac{q_{t,n|y}(x_t)}{q_{t|y}(x_t)} \left| (X_t - \sqrt{\bar{\alpha}_t} \mu_{0,n})^\intercal X_t \right|$$

$$= \sum_{n=1}^{N} \pi_n \mathbb{E}_{X_t \sim Q_{t,n|y}} \left| (X_t - \sqrt{\bar{\alpha}_t} \mu_{0,n})^\intercal X_t \right|$$

$$\leq \sum_{n=1}^{N} \pi_n \mathbb{E}_{X_t \sim Q_{t,n|y}} \left\| X_t - \sqrt{\bar{\alpha}_t} \mu_{0,n} \right\|^2 + \sqrt{\bar{\alpha}_t} \sum_{n=1}^{N} \pi_n \left\| \mu_{0,n} \right\| \sqrt{\mathbb{E}_{X_t \sim Q_{t,n|y}} \left\| X_t - \sqrt{\bar{\alpha}_t} \mu_{0,n} \right\|^2}$$

$$\overset{(i)}{=} \text{Tr}(\Sigma_{t|y}) + \bar{\alpha}_t \sum_{n=1}^{N} \pi_n \left\| H^\dagger y - H^\dagger H \mu_{0,n} \right\|^2$$

$$+ \sqrt{\bar{\alpha}_t} \sum_{n=1}^{N} \pi_n \left\| \mu_{0,n} \right\| \sqrt{\text{Tr}(\Sigma_{t|y}) + \bar{\alpha}_t \sum_{n=1}^{N} \pi_n \left\| H^\dagger y - H^\dagger H \mu_{0,n} \right\|^2}$$

$$\lesssim d + \sum_{n=1}^{N} \pi_n \left\| H^\dagger y - H^\dagger H \mu_{0,n} \right\|^2 \tag{42}$$

where $(i)$ follows from (58).

Also, note that $(I_d - H^\dagger H)\Sigma_{t-1|y}^{-r}(H^\dagger H) = 0$ using the following simple induction argument. For the base case, we have $(I_d - H^\dagger H)\Sigma_{t-1|y}^{-1}(H^\dagger H) = 0$ from Lemma 9. Then, suppose

$(I_d - H^\dagger H)\Sigma_{t-1|y}^{-(r-1)}(H^\dagger H) = 0$, we have $(I_d - H^\dagger H)\Sigma_{t-1|y}^{-r}(H^\dagger H) = (I_d - H^\dagger H)\Sigma_{t-1|y}^{-(r-1)}(I_d - H^\dagger H + H^\dagger H)\Sigma_{t-1|y}^{-1}(H^\dagger H) = (I_d - H^\dagger H)\Sigma_{t-1|y}^{-(r-1)}(I_d - H^\dagger H)\Sigma_{t-1|y}^{-1}(H^\dagger H) + (I_d - H^\dagger H)\Sigma_{t-1|y}^{-(r-1)}(H^\dagger H)\Sigma_{t-1|y}^{-1}(H^\dagger H) = 0$. Thus, for all $r \geq 1$ and any fixed vector $v$, with the definition of $\Delta_{t,y}$ in (40) and (41),

$$\mathbb{E}_{X_t \sim Q_{t|y}} \left| (\Sigma_{t-1|y}^{-r} v)^\intercal \Delta_{t,y} \right|$$

$$\leq \|v\| \, \mathbb{E}_{X_t \sim Q_{t|y}} \left\| \Sigma_{t-1|y}^{-r} \Delta_{t,y} \right\| = \|v\| \, \mathbb{E}_{X_t \sim Q_{t|y}} \left\| A_{t-1}^r \Delta_{t,y} \right\|$$

$$\leq 4 \|v\| \left\| (\bar{\alpha}_{t-1}[\Sigma_0]_{\bar{y}\bar{y}} + (1 - \bar{\alpha}_{t-1})I_{d-p})^{-r} \right\| \max_{n \in [N]} \left\| [\Sigma_t^{-1}]_{\bar{y}:} \mu_{0,n} \right\|$$

$$+ 2 \|v\| \left\| (\bar{\alpha}_{t-1}[\Sigma_0]_{\bar{y}\bar{y}} + (1 - \bar{\alpha}_{t-1})I_{d-p})^{-r} \right\| \left\| (\bar{\alpha}_t[\Sigma_0]_{\bar{y}\bar{y}} + (1 - \bar{\alpha}_t)I_{d-p})^{-1}[\Sigma_0]_{\bar{y}y}[\Sigma_t^{-1}]_{y:} \right\| \times$$

$$\sqrt{\mathbb{E}_{X_t \sim Q_{t|y}} \sum_{n=1}^{N} \pi_n \frac{q_{t,n|y}(X_t)}{q_{t|y}(X_t)} \left\| X_t - \sqrt{\bar{\alpha}_t}\mu_{0,n} \right\|^2}$$

$$\lesssim d + \sum_{n=1}^{N} \pi_n \left\| H^\dagger y - H^\dagger H \mu_{0,n} \right\|^2 \tag{43}$$

where the last line follows from (58). Similarly,

$$\mathbb{E}_{X_t \sim Q_{t|y}} \left| (\Sigma_{t|y}^{-r} v)^\intercal \Delta_{t,y} \right| \lesssim d + \sum_{n=1}^{N} \pi_n \left\| H^\dagger y - H^\dagger H \mu_{0,n} \right\|^2.$$

Also, for all $r \geq 1$,

$$\mathbb{E}_{X_t \sim Q_{t|y}} \left| (\Sigma_{t-1|y}^{-r} X_t)^\intercal \Delta_{t,y} \right| = \mathbb{E}_{X_t \sim Q_{t|y}} \left| X_t^\intercal \Sigma_{t-1|y}^{-r}(I_d - H^\dagger H)\Delta_{t,y} \right|$$

$$= \mathbb{E}_{X_t \sim Q_{t|y}} \left| X_t^\intercal A_{t-1}^r \Delta_{t,y} \right|$$

$$\leq 4 \left\| (\bar{\alpha}_{t-1}[\Sigma_0]_{\bar{y}\bar{y}} + (1 - \bar{\alpha}_{t-1})I_{d-p})^{-r} \right\| \max_{n \in [N]} \left\| [\Sigma_t^{-1}]_{\bar{y}:} \mu_{0,n} \right\| \mathbb{E}_{X_t \sim Q_{t|y}} \|X_t\|$$

$$+ 2 \left\| (\bar{\alpha}_{t-1}[\Sigma_0]_{\bar{y}\bar{y}} + (1 - \bar{\alpha}_{t-1})I_{d-p})^{-r} \right\| \left\| (\bar{\alpha}_t[\Sigma_0]_{\bar{y}\bar{y}} + (1 - \bar{\alpha}_t)I_{d-p})^{-1}[\Sigma_0]_{\bar{y}y}[\Sigma_t^{-1}]_{y:} \right\| \times$$

$$\mathbb{E}_{X_t \sim Q_{t|y}} \sum_{n=1}^{N} \pi_n \frac{q_{t,n|y}(x_t)}{q_{t|y}(x_t)} \left| (X_t - \sqrt{\bar{\alpha}_t}\mu_{0,n})^\intercal X_t \right|$$

$$\lesssim d + \sum_{n=1}^{N} \pi_n \left\| H^\dagger y - H^\dagger H \mu_{0,n} \right\|^2 \tag{44}$$

where the last line follows from (42) and the fact that, from (58),

$$\sqrt{d} \cdot \mathbb{E}_{X_t \sim Q_{t|y}} \|X_t\| \leq \sqrt{d} \sum_{n=1}^{N} \pi_n \sqrt{2\mathbb{E}_{X_t \sim Q_{t,n|y}} \|X_t - \mu_{t,n}\|^2 + 2 \|\mu_{t,n}\|^2}$$

$$\lesssim \sqrt{d} \sum_{n=1}^{N} \pi_n \sqrt{\text{Tr}(\Sigma_{t|y}) + \bar{\alpha}_t \|H^\dagger y - H^\dagger H \mu_{0,n}\|^2 + d}$$

$$\lesssim d + \sqrt{d} \sum_{n=1}^{N} \pi_n \left\| H^\dagger y - H^\dagger H \mu_{0,n} \right\|$$

$$\lesssim d + \sum_{n=1}^{N} \pi_n \left\| H^\dagger y - H^\dagger H \mu_{0,n} \right\|^2.$$

Similarly, we also have $\mathbb{E}_{X_t \sim Q_{t|y}} \left| (\Sigma_{t|y}^{-r} X_t)^\intercal \Delta_{t,y} \right| \lesssim d + \sum_{n=1}^{N} \pi_n \left\| H^\dagger y - H^\dagger H \mu_{0,n} \right\|^2$.

Also, for all $r \geq 1$, using the expression of $\nabla \log q_{t|y}$ in (39), and noting that by definition $(I_d - H^\dagger H)(x_t - \mu_{t,n|y}) = (I_d - H^\dagger H)(x_t - \mu_{t,n})$, we have

$$\mathbb{E}_{X_t \sim Q_{t|y}} \left| (\Sigma_{t-1|y}^{-r} \nabla \log q_{t|y}(X_t))^\intercal \Delta_{t,y} \right|$$

$$= \mathbb{E}_{X_t \sim Q_{t|y}} \left| \left( A_{t-1}^r \sum_{n=1}^N \pi_n \frac{q_{t,n|y}(X_t)}{q_{t|y}(X_t)} A_t (I_d - H^\dagger H)(X_t - \mu_{t,n|y}) \right)^\intercal \Delta_{t,y} \right|$$

$$\leq 4 \left\| (\bar{\alpha}_{t-1}[\Sigma_0]_{\bar{y}\bar{y}} + (1-\bar{\alpha}_{t-1})I_{d-p})^{-r} \right\| \left\| (\bar{\alpha}_t [\Sigma_0]_{\bar{y}\bar{y}} + (1-\bar{\alpha}_t)I_{d-p})^{-1} \right\| \times$$

$$\mathbb{E}_{X_t \sim Q_{t|y}} \left\| \sum_{\ell=1}^N \pi_\ell \frac{q_{t,\ell|y}(X_t)}{q_{t|y}(X_t)} [X_t - \sqrt{\bar{\alpha}_t}\mu_{0,\ell}]_{\bar{y}} \right\| \times \max_{n \in [N]} \left\| [\Sigma_t^{-1}]_{\bar{y}:}\mu_{0,n} \right\|$$

$$+ 2 \left\| (\bar{\alpha}_{t-1}[\Sigma_0]_{\bar{y}\bar{y}} + (1-\bar{\alpha}_{t-1})I_{d-p})^{-r} \right\| \left\| (\bar{\alpha}_t [\Sigma_0]_{\bar{y}\bar{y}} + (1-\bar{\alpha}_t)I_{d-p})^{-1} \right\|^2 \times$$

$$\mathbb{E}_{\substack{X_t \sim Q_{t|y} \\ N,L \sim \Pi_{\cdot|t,y}(\cdot|X_t)}} \left[ [X_t - \sqrt{\bar{\alpha}_t}\mu_{0,L}]_{\bar{y}}^\intercal [\Sigma_0]_{\bar{y}y}[\Sigma_t^{-1}]_{y:}(X_t - \sqrt{\bar{\alpha}_t}\mu_{0,N}) \right]$$

$$\lesssim \sqrt{\sum_{n=1}^N \pi_n \mathbb{E}_{X_t \sim Q_{t,n|y}} \left\| X_t - \sqrt{\bar{\alpha}_t}\mu_{0,n} \right\|^2} \times \sqrt{d}$$

$$+ \mathbb{E}_{\substack{X_t \sim Q_{t|y} \\ N,L \sim \Pi_{\cdot|t,y}(\cdot|X_t)}} \left\| X_t - \sqrt{\bar{\alpha}_t}\mu_{0,L} \right\| \left\| X_t - \sqrt{\bar{\alpha}_t}\mu_{0,N} \right\|$$

$$\lesssim d + \sum_{n=1}^N \pi_n \left\| H^\dagger y - H^\dagger H \mu_{0,n} \right\|^2 \tag{45}$$

where the last line follows because, from (58),

$$\mathbb{E}_{\substack{X_t \sim Q_{t|y} \\ N,L \sim \Pi_{\cdot|t,y}(\cdot|X_t)}} \left[ \left\| X_t - \sqrt{\bar{\alpha}_t}\mu_{0,L} \right\| \left\| X_t - \sqrt{\bar{\alpha}_t}\mu_{0,N} \right\| \right]$$

$$\leq \sqrt{\mathbb{E}_{\substack{X_t \sim Q_{t|y} \\ L \sim \Pi_{\cdot|t,y}(\cdot|X_t)}} \left\| X_t - \sqrt{\bar{\alpha}_t}\mu_{0,L} \right\|^2} \times \sqrt{\mathbb{E}_{\substack{X_t \sim Q_{t|y} \\ N \sim \Pi_{\cdot|t,y}(\cdot|X_t)}} \left\| X_t - \sqrt{\bar{\alpha}_t}\mu_{0,N} \right\|^2}$$

$$= \mathbb{E}_{\substack{X_t \sim Q_{t|y} \\ N \sim \Pi_{\cdot|t,y}(\cdot|X_t)}} \left\| X_t - \sqrt{\bar{\alpha}_t}\mu_{0,N} \right\|^2$$

$$\lesssim d + \sum_{n=1}^N \pi_n \left\| H^\dagger y - H^\dagger H \mu_{0,n} \right\|^2.$$

Now, we start to analyze the first term of $\mathcal{W}_{\text{vanish}}$. Recall that $m_{t,y}(x_t) = \mathbb{E}_{X_{t-1} \sim Q_{t-1|t,y}}[X_{t-1}] = \frac{1}{\sqrt{\alpha_t}}x_t + \frac{1-\alpha_t}{\sqrt{\alpha_t}}\nabla \log q_{t|y}(x_t)$. Using the score expressions in (39), we can calculate that given $x_t$ (and thus $m_{t,y} = m_{t,y}(x_t)$),

$$\nabla \log q_{t-1|y}(m_{t,y}) - \sqrt{\alpha_t}\nabla \log q_{t|y}(x_t)$$

$$= \sqrt{\alpha_t}\Sigma_{t|y}^{-1}x_t - \Sigma_{t-1|y}^{-1}m_{t,y}$$

$$- \sqrt{\alpha_t}\Sigma_{t|y}^{-1}\sum_{n=1}^N \pi_n \frac{q_{t,n|y}(x_t)}{q_{t|y}(x_t)}\mu_{t,n|y} + \Sigma_{t-1|y}^{-1}\sum_{n=1}^N \pi_n \frac{q_{t-1,n|y}(m_{t,y})}{q_{t-1|y}(m_{t,y})}\mu_{t-1,n|y}$$

$$= \left(\Sigma_{t|y}^{-1} - \Sigma_{t-1|y}^{-1}\right)\sqrt{\alpha_t}x_t - \frac{1-\alpha_t}{\sqrt{\alpha_t}}\Sigma_{t-1|y}^{-1}(x_t + \nabla \log q_{t|y}(x_t))$$

$$+ (1-\alpha_t)\Sigma_{t-1|y}^{-1}\sum_{n=1}^N \pi_n \frac{q_{t-1,n|y}(m_{t,y})}{q_{t-1|y}(m_{t,y})}\mu_{t-1,n|y}$$

$$+ \alpha_t \frac{\sum_{n,\ell=1}^N \pi_n \pi_\ell \left(q_{t-1,n|y}(m_{t,y})q_{t,\ell|y}(x_t)\Sigma_{t-1|y}^{-1} - q_{t-1,\ell|y}(m_{t,y})q_{t,n|y}(x_t)\Sigma_{t|y}^{-1}\right)\mu_{t-1,n|y}}{q_{t-1|y}(m_{t,y})q_{t|y}(x_t)}.$$

Here, using similar analyses as in the proof of Lemma 5, we get

$$(m_{t,y} - \mu_{t-1,n|y}) - (x_t - \mu_{t,n|y}) = \frac{1-\sqrt{\alpha_t}}{\sqrt{\alpha_t}}x_t + \frac{1-\alpha_t}{\sqrt{\alpha_t}}\nabla \log q_{t|y}(x_t) - (1-\sqrt{\alpha_t})\mu_{t-1,n|y}$$

$$\Sigma_{t|y}^{-1} - \Sigma_{t-1|y}^{-1} = \frac{1-\alpha_t}{\alpha_t}\Sigma_{t-1,n}^{-1} + \frac{1-\alpha_t}{\alpha_t^2}\Sigma_{t-1,n}^{-2} + O((1-\alpha_t)^2)$$

$$q_{t-1,n|y}(m_{t,y})q_{t,\ell|y}(x_t)\Sigma^{-1}_{t-1|y} - q_{t-1,\ell|y}(m_{t,y})q_{t,n|y}(x_t)\Sigma^{-1}_{t|y}$$

$$= q_{t-1,n|y}(m_{t,y})q_{t,\ell|y}(x_t)(\Sigma^{-1}_{t-1|y} - \Sigma^{-1}_{t|y})$$
$$+ (q_{t-1,n|y}(m_{t,y})q_{t,\ell|y}(x_t) - q_{t-1,\ell|y}(m_{t,y})q_{t,n|y}(x_t))\Sigma^{-1}_{t|y}$$

$$= q_{t-1,n|y}(m_{t,y})q_{t,\ell|y}(x_t)(\Sigma^{-1}_{t-1|y} - \Sigma^{-1}_{t|y})$$
$$+ \Bigg(\frac{1}{2}((m_{t,y} - \mu_{t-1,\ell|y}) - (x_t - \mu_{t,\ell|y}))^\mathsf{T}\Sigma^{-1}_{t-1|y}(m_{t,y} - \mu_{t-1,\ell|y})$$
$$+ \frac{1}{2}(x_t - \mu_{t,\ell|y})^\mathsf{T}(\Sigma^{-1}_{t-1|y} - \Sigma^{-1}_{t|y})(m_{t,y} - \mu_{t-1,\ell|y})$$
$$+ \frac{1}{2}(x_t - \mu_{t,\ell|y})^\mathsf{T}\Sigma^{-1}_{t|y}((m_{t,y} - \mu_{t-1,\ell|y}) - (x_t - \mu_{t,\ell|y}))$$
$$- \frac{1}{2}((m_{t,y} - \mu_{t-1,n|y}) - (x_t - \mu_{t,n|y}))^\mathsf{T}\Sigma^{-1}_{t-1|y}(m_{t,y} - \mu_{t-1,n|y})$$
$$- \frac{1}{2}(x_t - \mu_{t,n|y})^\mathsf{T}(\Sigma^{-1}_{t-1|y} - \Sigma^{-1}_{t|y})(m_{t,y} - \mu_{t-1,n|y})$$
$$- \frac{1}{2}(x_t - \mu_{t,n|y})^\mathsf{T}\Sigma^{-1}_{t|y}((m_{t,y} - \mu_{t-1,n|y}) - (x_t - \mu_{t,n|y}))\Bigg)\Sigma^{-1}_{t|y}$$
$$+ O((1 - \alpha_t)^2)$$

Thus,

$$\left|\mathbb{E}_{X_t \sim Q_{t|y}}\Big[(\nabla \log q_{t-1|y}(m_{t,y}) - \sqrt{\alpha_t}\nabla \log q_{t|y})^\mathsf{T}\Delta_{t,y}\Big]\right|$$

$$\leq \mathbb{E}_{X_t \sim Q_{t|y}}\left[\left|X_t^\mathsf{T}(\Sigma^{-1}_{t|y} - \Sigma^{-1}_{t-1|y})\Delta_{t,y}\right|\right]$$
$$+ \frac{1 - \alpha_t}{\sqrt{\alpha_t}}\mathbb{E}_{X_t \sim Q_{t|y}}\left[\left|(X_t + \nabla \log q_{t|y})^\mathsf{T}(\Sigma^{-1}_{t-1|y})\Delta_{t,y}\right|\right]$$
$$+ (1 - \alpha_t)\mathbb{E}_{X_t \sim Q_{t|y}}\left[\left|\left(\sum_{n=1}^N \pi_n \frac{q_{t-1,n|y}(m_{t,y})}{q_{t-1|y}(m_{t,y})}\mu_{t-1,n|y}\right)^\mathsf{T}\Sigma^{-1}_{t-1|y}\Delta_{t,y}\right|\right]$$
$$+ \mathbb{E}_{X_t \sim Q_{t|y}}\left|\Delta_{t,y}^\mathsf{T}\sum_{n,\ell=1}^N \pi_n\pi_\ell\left(\frac{q_{t-1,n|y}(m_{t,y})q_{t,\ell|y}(X_t)}{q_{t-1|y}(m_{t,y})q_{t|y}(X_t)}\Sigma^{-1}_{t-1|y} - \frac{q_{t-1,\ell|y}(m_{t,y})q_{t,n|y}(X_t)}{q_{t-1|y}(m_{t,y})q_{t|y}(X_t)}\Sigma^{-1}_{t|y}\right)\mu_{t-1,n|y}\right|.$$

Among the four terms above, the first term $\lesssim d + \sum_{n=1}^N \pi_n \left\|H^\dagger y - H^\dagger H\mu_{0,n}\right\|^2$ from (44) (along with the similar result for $\Sigma^{-1}_{t|y}$), the second term $\lesssim d + \sum_{n=1}^N \pi_n \left\|H^\dagger y - H^\dagger H\mu_{0,n}\right\|^2$ from (44) and (45), and both the third and the fourth term $\lesssim d + \sum_{n=1}^N \pi_n \left\|H^\dagger y - H^\dagger H\mu_{0,n}\right\|^2$ from (43) (along with the similar result for $\Sigma^{-1}_{t|y}$). Thus,

$$\left|\mathbb{E}_{X_t \sim Q_{t|y}}\Big[(\nabla \log q_{t-1|y}(m_{t,y}) - \sqrt{\alpha_t}\nabla \log q_{t|y}(X_t))^\mathsf{T}\Delta_{t,y}\Big]\right| \lesssim d + \sum_{n=1}^N \pi_n \left\|H^\dagger y - H^\dagger H\mu_{0,n}\right\|^2,$$

and the first term in $\mathcal{W}_{\text{vanish}}$ satisfies that

$$\sum_{t=1}^T \frac{1 - \alpha_t}{\sqrt{\alpha_t}}\mathbb{E}_{X_t \sim Q_{t|y}}\Big[(\nabla \log q_{t-1|y}(m_{t,y}(X_t)) - \sqrt{\alpha_t}\nabla \log q_{t|y}(X_t))^\mathsf{T}\Delta_{t,y}(X_t)\Big]$$
$$\lesssim \left(d + \sum_{n=1}^N \pi_n \left\|H^\dagger y - H^\dagger H\mu_{0,n}\right\|^2\right)\frac{\log(1/\delta)^2(\log T)^2}{T}.$$

For the second term in $\mathcal{W}_{\text{vanish}}$, we first provide the following useful calculation. Similar to (43), for all $r \geq 1$ and any fixed vector $v$,

$$\mathbb{E}_{X_t \sim Q_{t|y}}\left|(\Sigma^{-r}_{t|y}v)^\mathsf{T}\Delta_{t,y}\right|^2$$

$$\leq \|v\|^2 \, \mathbb{E}_{X_t \sim Q_{t|y}} \left\|\Sigma_{t|y}^{-r}\Delta_{t,y}\right\|^2 = \|v\|^2 \, \mathbb{E}_{X_t \sim Q_{t|y}} \|A_t^r \Delta_{t,y}\|^2$$

$$\lesssim \|v\|^2 \left\|(\bar\alpha_t[\Sigma_0]_{\bar y \bar y} + (1-\bar\alpha_t)I_{d-p})^{-r}\right\|^2 \max_{n\in[N]} \left\|[\Sigma_t^{-1}]_{\bar y:}\mu_{0,n}\right\|^2$$

$$+ \|v\|^2 \left\|(\bar\alpha_t[\Sigma_0]_{\bar y \bar y} + (1-\bar\alpha_t)I_{d-p})^{-r}\right\|^2 \left\|(\bar\alpha_t[\Sigma_0]_{\bar y \bar y} + (1-\bar\alpha_t)I_{d-p})^{-1}[\Sigma_0]_{\bar y y}[\Sigma_t^{-1}]_{y:}\right\|^2 \times$$

$$\mathbb{E}_{X_t \sim Q_{t|y}} \sum_{n=1}^{N} \pi_n \frac{q_{t,n|y}(X_t)}{q_{t|y}(X_t)} \left\|X_t - \sqrt{\bar\alpha_t}\mu_{0,n}\right\|^2$$

$$\lesssim d^2 + d \sum_{n=1}^{N} \pi_n \left\|H^\dagger y - H^\dagger H \mu_{0,n}\right\|^2 \tag{46}$$

where the last line follows from (58).

Also, similar to (44), for all $r \geq 1$,

$$\mathbb{E}_{X_t \sim Q_{t|y}} \left|(\Sigma_{t|y}^{-r}m_{t,y})^{\mathsf{T}}\Delta_{t,y}\right|^2$$

$$\lesssim \mathbb{E}_{X_t \sim Q_{t|y}} \left|X_t^{\mathsf{T}}\Sigma_{t|y}^{-r}\Delta_{t,y}\right|^2 + (1-\alpha_t)\mathbb{E}_{X_t \sim Q_{t|y}} \left|(\nabla \log q_{t|y}(X_t))^{\mathsf{T}}\Sigma_{t|y}^{-r}\Delta_{t,y}\right|^2$$

$$\overset{(ii)}{\lesssim} \mathbb{E}_{X_t \sim Q_{t|y}} \left|X_t^{\mathsf{T}}\Sigma_{t|y}^{-r}(I_d - H^\dagger H)\Delta_{t,y}\right|^2 = \mathbb{E}_{X_t \sim Q_{t|y}} |X_t^{\mathsf{T}}A_t^r\Delta_{t,y}|^2$$

$$\lesssim \left\|(\bar\alpha_t[\Sigma_0]_{\bar y \bar y} + (1-\bar\alpha_t)I_{d-p})^{-r}\right\|^2 \max_{n\in[N]} \left\|[\Sigma_t^{-1}]_{\bar y:}\mu_{0,n}\right\|^2 \mathbb{E}_{X_t \sim Q_{t|y}} \|X_t\|^2$$

$$+ \left\|(\bar\alpha_t[\Sigma_0]_{\bar y \bar y} + (1-\bar\alpha_t)I_{d-p})^{-r}\right\|^2 \left\|(\bar\alpha_t[\Sigma_0]_{\bar y \bar y} + (1-\bar\alpha_t)I_{d-p})^{-1}[\Sigma_0]_{\bar y y}[\Sigma_t^{-1}]_{y:}\right\|^2 \times$$

$$\mathbb{E}_{X_t \sim Q_{t|y}} \sum_{n=1}^{N} \pi_n \frac{q_{t,n|y}(X_t)}{q_{t|y}(X_t)} \left|(X_t - \sqrt{\bar\alpha_t}\mu_{0,n})^{\mathsf{T}}X_t\right|^2$$

$$\lesssim d^2 + \sum_{n=1}^{N} \pi_n \left\|H^\dagger y - H^\dagger H \mu_{0,n}\right\|^4 \tag{47}$$

where $(ii)$ follows from the fact that $\mathbb{E}_{X_t \sim Q_{t|y}} \left|(\nabla \log q_{t|y}(X_t))^{\mathsf{T}}\Sigma_{t|y}^{-r}\Delta_{t,y}\right|^2 \lesssim d^2$ (using a similar argument for deriving (45)), and the last line follows because

$$d \cdot \mathbb{E}_{X_t \sim Q_{t|y}} \|X_t\|^2 \leq d \sum_{n=1}^{N} \pi_n \left(2\mathbb{E}_{X_t \sim Q_{t,n|y}} \|X_t - \mu_{t,n}\|^2 + 2\|\mu_{t,n}\|^2\right)$$

$$\overset{(iii)}{\lesssim} d \sum_{n=1}^{N} \pi_n \left(\mathrm{Tr}(\Sigma_{t|y}) + \bar\alpha_t \left\|H^\dagger y - H^\dagger H \mu_{0,n}\right\|^2 + d\right)$$

$$\lesssim d^2 + d \sum_{n=1}^{N} \pi_n \left\|H^\dagger y - H^\dagger H \mu_{0,n}\right\|^2$$

where $(iii)$ follows from (58), and also

$$\mathbb{E}_{X_t \sim Q_{t|y}} \sum_{n=1}^{N} \pi_n \frac{q_{t,n|y}(X_t)}{q_{t|y}(X_t)} \left|(X_t - \sqrt{\bar\alpha_t}\mu_{0,n})^{\mathsf{T}}X_t\right|^2$$

$$= \mathbb{E}_{\substack{X_t \sim Q_{t|y} \\ N \sim \Pi_{\cdot|t,y}(\cdot|X_t)}} \left((X_t - \sqrt{\bar\alpha_t}\mu_{0,N})^{\mathsf{T}}X_t\right)^2$$

$$\leq 2\mathbb{E}_{\substack{X_t \sim Q_{t|y} \\ N \sim \Pi_{\cdot|t,y}(\cdot|X_t)}} \left\|X_t - \sqrt{\bar\alpha_t}\mu_{0,N}\right\|^4 + 2\sqrt{\mathbb{E}_{\substack{X_t \sim Q_{t|y} \\ N \sim \Pi_{\cdot|t,y}(\cdot|X_t)}} \left\|X_t - \sqrt{\bar\alpha_t}\mu_{0,N}\right\|^4} \sqrt{\mathbb{E}_{N\sim\Pi} \left\|\mu_{0,N}\right\|^4}$$

$$\lesssim d^2 + \sum_{n=1}^{N} \pi_n \left\|H^\dagger y - H^\dagger H \mu_{0,n}\right\|^4$$

where the last line above follows because for all $r \geq 1$ and each $n \in [N]$,

$$
\begin{aligned}
&\mathbb{E}_{X_t \sim Q_{t,n|y}} \left\| X_t - \sqrt{\bar{\alpha}_t} \mu_{0,n} \right\|^r \\
&\lesssim \mathbb{E}_{X_t \sim Q_{t,n|y}} \left\| X_t - \sqrt{\bar{\alpha}_t} \mu_{0,n|y} \right\|^r + \left\| \sqrt{\bar{\alpha}_t} \mu_{0,n|y} - \sqrt{\bar{\alpha}_t} \mu_{0,n} \right\|^r \\
&\leq \left\| \Sigma_{t|y}^{\frac{1}{2}} \right\|^r \mathbb{E}_{X_t \sim Q_{t,n|y}} \left\| \Sigma_{t|y}^{-\frac{1}{2}} (X_t - \sqrt{\bar{\alpha}_t} \mu_{0,n|y}) \right\|^r + (\bar{\alpha}_t)^{r/2} \left\| H^\dagger y - H^\dagger H \mu_{0,n} \right\|^r \\
&\lesssim d^{r/2} + \left\| H^\dagger y - H^\dagger H \mu_{0,n} \right\|^r .
\end{aligned} \tag{48}
$$

Now we are ready to analyze the second term of $\mathcal{W}_{\text{vanish}}$. Note that

$$
\begin{aligned}
&\nabla^2 \log q_{t-1|y}(m_{t,y}) \\
&= \sum_{n=1}^N \pi_n \frac{q_{t-1,n|y}(m_{t,y})}{q_{t-1|y}(m_{t,y})} \left( \Sigma_{t|y}^{-1} (m_{t,y} - \mu_{t,n|y})(m_{t,y} - \mu_{t,n|y})^\intercal \Sigma_{t|y}^{-1} \right) - \Sigma_{t|y}^{-1} \\
&\quad - \left( \sum_{n=1}^N \pi_n \frac{q_{t-1,n|y}(m_{t,y})}{q_{t-1|y}(m_{t,y})} \Sigma_{t|y}^{-1} (m_{t,y} - \mu_{t,n|y}) \right) \left( \sum_{n=1}^N \pi_n \frac{q_{t-1,n|y}(m_{t,y})}{q_{t-1|y}(m_{t,y})} \Sigma_{t|y}^{-1} (m_{t,y} - \mu_{t,n|y}) \right)^\intercal .
\end{aligned}
$$

Thus,

$$
\begin{aligned}
&\mathbb{E}_{X_t \sim Q_{t|y}} \left| \Delta_{t,y}^\intercal \nabla^2 \log q_{t-1|y}(m_{t,y}) \Delta_{t,y} \right| \\
&\leq 3 \mathbb{E}_{X_t \sim Q_{t|y}} \left[ \sum_{n=1}^N \pi_n \frac{q_{t-1,n|y}(m_{t,y})}{q_{t-1|y}(m_{t,y})} \left( \Delta_{t,y}^\intercal \Sigma_{t|y}^{-1} (m_{t,y} - \mu_{t,n|y}) \right)^2 \right] \\
&\quad + 3 \mathbb{E}_{X_t \sim Q_{t|y}} \left| \Delta_{t,y}^\intercal \Sigma_{t|y}^{-1} \Delta_{t,y} \right| \\
&\quad + 3 \mathbb{E}_{X_t \sim Q_{t|y}} \left( \sum_{n=1}^N \pi_n \frac{q_{t-1,n|y}(m_{t,y})}{q_{t-1|y}(m_{t,y})} \Delta_{t,y}^\intercal \Sigma_{t|y}^{-1} (m_{t,y} - \mu_{t,n|y}) \right)^2 \\
&\leq 3 \mathbb{E}_{X_t \sim Q_{t|y}} \left| \Delta_{t,y}^\intercal \Sigma_{t|y}^{-1} \Delta_{t,y} \right| \\
&\quad + 6 \mathbb{E}_{X_t \sim Q_{t|y}} \left[ \sum_{n=1}^N \pi_n \frac{q_{t-1,n|y}(m_{t,y})}{q_{t-1|y}(m_{t,y})} \left( \Delta_{t,y}^\intercal \Sigma_{t|y}^{-1} (m_{t,y} - \mu_{t,n|y}) \right)^2 \right] .
\end{aligned}
$$

To determine the rate of these two terms, we get

$$
\mathbb{E}_{X_t \sim Q_{t|y}} \left| \Delta_{t,y}^\intercal \Sigma_{t|y}^{-1} \Delta_{t,y} \right| = \mathbb{E}_{X_t \sim Q_{t|y}} \left| \Delta_{t,y}^\intercal A_t \Delta_{t,y} \right| \lesssim d + \sum_{n=1}^N \pi_n \left\| H^\dagger y - H^\dagger H \mu_{0,n} \right\|^2 ,
$$

and

$$
\begin{aligned}
&\mathbb{E}_{X_t \sim Q_{t|y}} \sum_{n=1}^N \pi_n \frac{q_{t-1,n|y}(m_{t,y})}{q_{t-1|y}(m_{t,y})} \left| (m_{t,y} - \mu_{t,n|y})^\intercal \Sigma_{t|y}^{-1} \Delta_{t,y} \right|^2 \\
&\leq \mathbb{E}_{X_t \sim Q_{t|y}} \max_{n \in [N]} \left| (m_{t,y} - \mu_{t,n|y})^\intercal \Sigma_{t|y}^{-1} \Delta_{t,y} \right|^2 \\
&\lesssim \mathbb{E}_{X_t \sim Q_{t|y}} \left| m_{t,y}^\intercal \Sigma_{t|y}^{-1} \Delta_{t,y} \right|^2 + \mathbb{E}_{X_t \sim Q_{t|y}} \max_{n \in [N]} \left| \mu_{t,n|y}^\intercal \Sigma_{t|y}^{-1} \Delta_{t,y} \right|^2 \\
&\lesssim N \left( d^2 + \sum_{n=1}^N \pi_n \left\| H^\dagger y - H^\dagger H \mu_{0,n} \right\|^4 \right)
\end{aligned}
$$

where the last line follows from (46) and (47). Thus, the second term of $\mathcal{W}_{\text{vanish}}$ satisfies that

$$
\sum_{t=1}^T \frac{(1-\alpha_t)^2}{2\alpha_t} \mathbb{E}_{X_t \sim Q_{t|y}} \left| \Delta_{t,y}(X_t)^\intercal \nabla^2 \log q_{t-1|y}(m_{t,y}(X_t)) \Delta_{t,y}(X_t) \right|
$$

$$\lesssim N \left( d^2 + \sum_{n=1}^{N} \pi_n \left\| H^\dagger y - H^\dagger H \mu_{0,n} \right\|^4 \right) \frac{c^2 (\log T)^2}{T}.$$

For the third term of $\mathcal{W}_{\mathrm{vanish}}$, we provide the following useful calculations. Denote $v^{\circ 3}$ as the element-wise (Hadamard) third power of a vector $v$. For each $n \in [N]$, we have

$$\mathbb{E}_{X_t \sim Q_{t|y}} \left| (m_{t,y} - \mu_{t,n|y})^{\circ 3} (I_d - H^\dagger H) \Delta_{t,y} \right| = \mathbb{E}_{X_t \sim Q_{t|y}} \left| [\Delta_{t,y}]_{\bar{y}}^{\mathsf{T}} [m_{t,y} - \mu_{t,n}]_{\bar{y}}^{\circ 3} \right|$$

$$\lesssim \mathbb{E}_{X_t \sim Q_{t|y}} \left| [\Delta_{t,y}]_{\bar{y}}^{\mathsf{T}} [X_t - \mu_{t,n}]_{\bar{y}}^{\circ 3} \right| + (1 - \alpha_t) \sqrt{\mathbb{E}_{X_t \sim Q_{t|y}} \|\Delta_{t,y}\|^2} \sqrt{\mathbb{E}_{X_t \sim Q_{t|y}} \|\nabla \log q_{t|y}(X_t)\|_6^6}$$

$$\overset{(iv)}{\lesssim} \mathbb{E}_{X_t \sim Q_{t|y}} \left| [\Delta_{t,y}]_{\bar{y}}^{\mathsf{T}} [X_t - \mu_{t,n}]_{\bar{y}}^{\circ 3} \right|$$

$$\lesssim \max_\ell \left\| [\Sigma_t^{-1}]_{\bar{y}:\mu_{0,\ell}} \right\| \sqrt{\mathbb{E}_{X_t \sim Q_{t|y}} \|X_t - \mu_{t,n}\|_6^6} + \mathbb{E}_{\substack{X_t \sim Q_{t|y} \\ L \sim \Pi_{\cdot|t,y}(\cdot|X_t)}} \|X_t - \mu_{t,L}\|_4^4$$

$$\lesssim \sqrt{\mathbb{E}_{\substack{X_t \sim Q_{t|y} \\ L \sim \Pi_{\cdot|t,y}(\cdot|X_t)}} \|X_t - \mu_{t,L}\|^6} + \sqrt{\mathbb{E}_{L \sim \Pi} \|\mu_{t,n} - \mu_{t,L}\|^6} + \mathbb{E}_{\substack{X_t \sim Q_{t|y} \\ L \sim \Pi_{\cdot|t,y}(\cdot|X_t)}} \|X_t - \mu_{t,L}\|^4$$

$$\lesssim d^2 + \sum_{n=1}^{N} \pi_n \left\| H^\dagger y - H^\dagger H \mu_{0,n} \right\|^4 \tag{49}$$

where $(iv)$ follows from Lemma 6 (using the $\alpha_t$ in (8)) and (Liang et al., 2024, Lemma 15), and the last line follows from (48). With a similar argument,

$$\mathbb{E}_{X_t \sim Q_{t|y}} \left| (m_{t,y} - \mu_{t,n|y})(I_d - H^\dagger H) \Delta_{t,y} \right| \lesssim \mathbb{E}_{\substack{X_t \sim Q_{t|y} \\ L \sim \Pi_{\cdot|t,y}(\cdot|X_t)}} \|X_t - \mu_{t,L}\|^2$$

$$\lesssim d + \sum_{n=1}^{N} \pi_n \left\| H^\dagger y - H^\dagger H \mu_{0,n} \right\|^2. \tag{50}$$

Now, employing the notations from (Liang et al., 2024, Section G.1), we define

$$z_{t,n}(x) := \Sigma_{t|y}^{-1}(x - \mu_{t,n|y}), \quad \xi_t(x,i) := \max_n \left| z_{t,n}^i(x) \right|, \quad \bar{\Sigma}_t^{ij} := \max_n \left| [\Sigma_{t|y}^{-1}]^{ij} \right|.$$

When $H = (I_p \quad 0)$, we note that

$$\Sigma_{t|y}^{-1} = \begin{pmatrix} (1 - \bar{\alpha}_t + \bar{\alpha}_t \sigma_y^2)^{-1} I_p & 0 \\ 0 & (\bar{\alpha}_t [\Sigma_0]_{\bar{y}\bar{y}} + (1 - \bar{\alpha}_t) I_{d-p})^{-1} \end{pmatrix}.$$

Thus, we have $\bar{\Sigma}_t^{ij} \equiv 0$ whenever $(i,j) \in [1,p] \times [p+1,d]$ or $(i,j) \in [p+1,d] \times [1,p]$ and $\max_{i,j \in [p+1,d]} \bar{\Sigma}_t^{ij} = O(1)$. Since $\sigma_y^2 > 0$, we also have $\left\| \Sigma_{t|y}^{-1} \right\| \lesssim 1$.

From (Liang et al., 2024, Section G.1.2), an upper bound for third-order partial derivatives is

$$\left| \partial_{ijk}^3 \log q_{t|y}(x) \right| \leq 6 \xi_t(x,i) \xi_t(x,j) \xi_t(x,k) + 2 \bar{\Sigma}_t^{ij} \xi_t(x,k) + 2 \bar{\Sigma}_t^{ik} \xi_t(x,j) + 2 \bar{\Sigma}_t^{jk} \xi_t(x,i).$$

We also remind readers that $\Delta_{t,y}$ is supported on $\mathrm{range}(I_d - H^\dagger H)$, namely that $[\Delta_{t,y}]_y \equiv 0$.

Now, the third term of $\mathcal{W}_{\mathrm{vanish}}$ can be upper-bounded as

$$\mathbb{E}_{X_t \sim Q_{t|y}} \left| \sum_{i=1}^{d} \partial_{iii}^3 \log q_{t-1|y}(m_{t,y}(X_t)) \Delta_{t,y}(X_t)^i \right|$$

$$= \mathbb{E}_{X_t \sim Q_{t|y}} \left| \sum_{i=p+1}^{d} \partial_{iii}^3 \log q_{t-1|y}(m_{t,y}(X_t)) \Delta_{t,y}(X_t)^i \right|$$

$$\lesssim \mathbb{E}_{X_t \sim Q_{t|y}} \left| \sum_{i=p+1}^{d} \xi_t(m_{t,y}(X_t), i)^3 \Delta_{t,y}(X_t)^i \right| + \mathbb{E}_{X_t \sim Q_{t|y}} \left| \sum_{i=p+1}^{d} \bar{\Sigma}_t^{ii} \xi_t(m_{t,y}(X_t), i) \Delta_{t,y}(X_t)^i \right|$$

$$\lesssim \sum_{n=1}^{N} \left\| (\bar{\alpha}_t [\Sigma_0]_{\bar{y}\bar{y}} + (1-\bar{\alpha}_t) I_{d-p})^{-1} \right\|^3 \mathbb{E}_{X_t \sim Q_{t|y}} \left| \sum_{i=p+1}^{d} (m_{t,y}(X_t)^i - \mu_{t,n|y}^i)^3 \Delta_{t,y}(X_t)^i \right|$$

$$+ \sum_{n=1}^{N} \left\| (\bar{\alpha}_t [\Sigma_0]_{\bar{y}\bar{y}} + (1-\bar{\alpha}_t) I_{d-p})^{-1} \right\| \mathbb{E}_{X_t \sim Q_{t|y}} \left| \sum_{i=p+1}^{d} (m_{t,y}(X_t)^i - \mu_{t,n|y}^i) \Delta_{t,y}(X_t)^i \right|$$

$$\lesssim N \left( d^2 + \sum_{n=1}^{N} \pi_n \left\| H^\dagger y - H^\dagger H \mu_{0,n} \right\|^4 \right).$$

Here the last line follows from (49) and (50).

We provide the following useful calculations to upper-bound the fourth term of $\mathcal{W}_{\text{vanish}}$. First, for all $r \geq 1$ and any fixed vector $v$,

$$\mathbb{E}_{X_t \sim Q_{t|y}} \|m_{t,y} - v\|^r = \mathbb{E}_{X_t \sim Q_{t|y}} \left\| \frac{1}{\sqrt{\alpha_t}} X_t + \frac{1-\alpha_t}{\sqrt{\alpha_t}} \nabla \log q_{t|y}(X_t) - v \right\|^r$$

$$\lesssim \mathbb{E}_{X_t \sim Q_{t|y}} \|X_t - v\|^r + (1-\alpha_t) \mathbb{E}_{X_t \sim Q_{t|y}} \left\| \nabla \log q_{t|y}(X_t) \right\|^r$$

$$\overset{(v)}{\lesssim} \mathbb{E}_{\substack{X_t \sim Q_{t|y} \\ L \sim \Pi_{\cdot|t,y}(\cdot|X_t)}} \|X_t - \mu_{t,L}\|^r + \mathbb{E}_{L \sim \Pi} \|\mu_{t,L} - v\|^r$$

$$\lesssim d^{r/2} + \sum_{n=1}^{N} \pi_n \left\| H^\dagger y - H^\dagger H \mu_{0,n} \right\|^r \tag{51}$$

where $(v)$ follows from Lemma 6 (using the $\alpha_t$ in (8)) and (Liang et al., 2024, Lemma 15), and the last line follows from (48). Now,

$$\mathbb{E}_{X_t \sim Q_{t|y}} \sum_{i=1}^{d} \xi_t(m_{t,y}, i)^2 \left| \sum_{j=p+1}^{d} \xi_t(m_{t,y}, j) \Delta_{t,y}(X_t)^j \right|$$

$$\lesssim \sum_{n,\ell=1}^{N} \mathbb{E}_{X_t \sim Q_{t|y}} \left\| m_{t,y} - \mu_{t,n|y} \right\|^2 \left| (m_{t,y} - \mu_{t,\ell|y})^\mathsf{T} \Delta_{t,y}(X_t) \right|$$

$$\leq \sum_{n,\ell=1}^{N} \sqrt{\mathbb{E}_{X_t \sim Q_{t|y}} \left\| m_{t,y} - \mu_{t,n|y} \right\|^4} \left( \mathbb{E}_{X_t \sim Q_{t|y}} \left\| m_{t,y} - \mu_{t,\ell|y} \right\|^4 \mathbb{E}_{X_t \sim Q_{t|y}} \left\| \Delta_{t,y}(X_t) \right\|^4 \right)^{1/4}$$

$$\lesssim N^2 \left( d^2 + \sum_{n=1}^{N} \pi_n \left\| H^\dagger y - H^\dagger H \mu_{0,n} \right\|^4 \right) \tag{52}$$

where the first inequality follows from $\left\| \Sigma_{t|y}^{-1} \right\| \lesssim 1$ and the last line follows from (51) and (59). Also,

$$\mathbb{E}_{X_t \sim Q_{t|y}} \left| \sum_{i,j=p+1}^{d} \bar{\Sigma}_t^{ii} \xi_t(m_{t,y}, j) \Delta_{t,y}(X_t)^j \right|$$

$$= \left| \sum_{i=p+1}^{d} \bar{\Sigma}_t^{ii} \right| \cdot \mathbb{E}_{X_t \sim Q_{t|y}} \left| \sum_{j=p+1}^{d} \xi_t(m_{t,y}, j) \Delta_{t,y}(X_t)^j \right|$$

$$\lesssim \left| \sum_{i=p+1}^{d} \bar{\Sigma}_t^{ii} \right| \cdot \sum_{n=1}^{N} \mathbb{E}_{X_t \sim Q_{t|y}} |(m_{t,y} - \mu_{t,n})^\mathsf{T} \Delta_{t,y}(X_t)|$$

$$\lesssim \left| \sum_{i=p+1}^{d} \bar{\Sigma}_t^{ii} \right| \cdot \sum_{n=1}^{N} \sqrt{\mathbb{E}_{X_t \sim Q_{t|y}} \|m_{t,y} - \mu_{t,n}\|^2} \sqrt{\mathbb{E}_{X_t \sim Q_{t|y}} \|\Delta_{t,y}(X_t)\|^2}$$

$$\lesssim Nd \left( d + \sum_{n=1}^{N} \pi_n \left\| H^\dagger y - H^\dagger H \mu_{0,n} \right\|^2 \right) \tag{53}$$

where the last line follows from Proposition 1 and (51). Also,

$$\mathbb{E}_{X_t \sim Q_{t|y}} \left| \sum_{i,j=p+1}^{d} \bar{\Sigma}_t^{ij} \xi_t(m_{t,y}, i) \Delta_{t,y}(X_t)^j \right|$$

$$\lesssim \mathbb{E}_{X_t \sim Q_{t|y}} \left( \sum_{i=p+1}^{d} \xi_t(m_{t,y}, i) \right) \left| \sum_{j=p+1}^{d} \Delta_{t,y}(X_t)^j \right|$$

$$\leq \sqrt{d \cdot \mathbb{E}_{X_t \sim Q_{t|y}} \left( \sum_{i=p+1}^{d} \xi_t(m_{t,y}, i)^2 \right)} \sqrt{d \cdot \mathbb{E}_{X_t \sim Q_{t|y}} \left\| \Delta_{t,y}(X_t) \right\|^2}$$

$$\lesssim \sqrt{d \cdot \sum_{n=1}^{N} \mathbb{E}_{X_t \sim Q_{t|y}} \left\| m_{t,y} - \mu_{t,n} \right\|^2} \sqrt{d \cdot \mathbb{E}_{X_t \sim Q_{t|y}} \left\| \Delta_{t,y}(X_t) \right\|^2}$$

$$\lesssim \sqrt{N} d \left( d + \sum_{n=1}^{N} \pi_n \left\| H^\dagger y - H^\dagger H \mu_{0,n} \right\|^2 \right) \tag{54}$$

where the last line follows from Proposition 1 and (51).

Now, the fourth term of $\mathcal{W}_{\text{vanish}}$ can be upper-bounded as

$$\mathbb{E}_{X_t \sim Q_{t|y}} \left| \sum_{i,j=1}^{d} \partial_{iij}^3 \log q_{t-1|y}(m_{t,y}(X_t)) \Delta_{t,y}(X_t)^j \right|$$

$$= \mathbb{E}_{X_t \sim Q_{t|y}} \left| \sum_{i=1}^{d} \sum_{j=p+1}^{d} \partial_{iij}^3 \log q_{t-1|y}(m_{t,y}(X_t)) \Delta_{t,y}(X_t)^j \right|$$

$$\lesssim \mathbb{E}_{X_t \sim Q_{t|y}} \sum_{i=1}^{d} \xi_t(m_{t,y}(X_t), i)^2 \left| \sum_{j=p+1}^{d} \xi_t(m_{t,y}(X_t), j) \Delta_{t,y}(X_t)^j \right|$$

$$+ \mathbb{E}_{X_t \sim Q_{t|y}} \left| \sum_{i,j=p+1}^{d} \bar{\Sigma}_t^{ii} \xi_t(m_{t,y}(X_t), j) \Delta_{t,y}(X_t)^j \right|$$

$$+ \mathbb{E}_{X_t \sim Q_{t|y}} \left| \sum_{i,j=p+1}^{d} \bar{\Sigma}_t^{ij} \xi_t(m_{t,y}(X_t), i) \Delta_{t,y}(X_t)^j \right|.$$

For the three terms above, the first term $\lesssim N^2 \left( d^2 + \sum_{n=1}^{N} \pi_n \left\| H^\dagger y - H^\dagger H \mu_{0,n} \right\|^4 \right)$ from (52), the second term $\lesssim Nd \left( d + \sum_{n=1}^{N} \pi_n \left\| H^\dagger y - H^\dagger H \mu_{0,n} \right\|^2 \right)$ from (53), and the last term $\lesssim \sqrt{N} d \left( d + \sum_{n=1}^{N} \pi_n \left\| H^\dagger y - H^\dagger H \mu_{0,n} \right\|^2 \right)$ from (54).

Thus, overall, with the $\alpha_t$ in (8) (cf. Lemma 6), the third and fourth terms give us

$$\sum_{t=1}^{T} \frac{(1-\alpha_t)^2}{3! \alpha_t^{3/2}} \mathbb{E}_{X_t \sim Q_{t|y}} \left[ 3 \sum_{i=1}^{d} \partial_{iii}^3 \log q_{t-1|y}(m_{t,y}(X_t)) \Delta_{t,y}(X_t)^i \right.$$

$$\left. + \sum_{\substack{i,j=1 \\ i \neq j}}^{d} \partial_{iij}^3 \log q_{t-1|y}(m_{t,y}(X_t)) \Delta_{t,y}(X_t)^j \right]$$

$$\lesssim N^2 \left( d^2 + \sum_{n=1}^{N} \pi_n \left\| H^{\dagger} y - H^{\dagger} H \mu_{0,n} \right\|^4 \right) \frac{(\log T)^2}{T}.$$

Therefore, combining all the above, since $N$ is constant,

$$\mathrm{KL}(Q_{0|y} \| \widehat{P}_{0|y}) \lesssim \left( d + \sum_{n=1}^{N} \pi_n \left\| H^{\dagger} y - H^{\dagger} H \mu_{0,n} \right\|^2 \right)$$
$$+ \left( d^2 + \sum_{n=1}^{N} \pi_n \left\| H^{\dagger} y - H^{\dagger} H \mu_{0,n} \right\|^4 \right) \frac{(\log T)^2}{T}$$
$$+ \sqrt{d + \sum_{n=1}^{N} \pi_n \left\| H^{\dagger} y - H^{\dagger} H \mu_{0,n} \right\|^2} (\log T) \varepsilon.$$

*Remark* 2. In case of general $\sigma_y \geq 0$, the same upper bound can be applied to $\left\| \Sigma_{t-1|y}^{-1} \right\|$ as detailed in Remark 1. Thus, with the $\alpha_t$ in (8), we similarly have

$$\mathcal{W}_{\text{oracle}} \lesssim \frac{d^2 (\log T)^2 \log(1/\delta)^2}{T} + (\log T) \varepsilon^2.$$

The rest of the proof is similar. Combining with Lemma 7, we would finally obtain

$$\mathrm{KL}(Q_{1|y} \| \widehat{P}_{1|y}) \lesssim \left( d + \sum_{n=1}^{N} \pi_n \left\| H^{\dagger} y - H^{\dagger} H \mu_{0,n} \right\|^2 \right) \left( 1 - \frac{2 \log(1/\delta) \log T}{T} \right)$$
$$+ \left( d^2 + \sum_{n=1}^{N} \pi_n \left\| H^{\dagger} y - H^{\dagger} H \mu_{0,n} \right\|^4 \right) \frac{(\log T)^2 \log(1/\delta)^2}{T} + \sqrt{d + \sum_{n=1}^{N} \pi_n \left\| H^{\dagger} y - H^{\dagger} H \mu_{0,n} \right\|^2} (\log T) \varepsilon.$$

Here $\mathrm{W}_2(Q_{1|y}, Q_{0|y})^2 \lesssim \delta d$.

## J  AUXILIARY LEMMAS AND PROOFS IN SECTION 4

### J.1  PROOF OF PROPOSITION 2

Given $Q_0 = \mathcal{N}(\mu_0, \Sigma_0)$, from the conditional forward model in (5), we can calculate

$$Q_t = \mathcal{N}(\sqrt{\bar{\alpha}_t} \mu_0, \bar{\alpha}_t \Sigma_0 + (1 - \bar{\alpha}_t) I_d) =: \mathcal{N}(\mu_t, \Sigma_t)$$
$$Q_{t|y} = \mathcal{N}(\sqrt{\bar{\alpha}_t}(I_d - H^{\dagger} H) \mu_0 + \sqrt{\bar{\alpha}_t} H^{\dagger} y,$$
$$\bar{\alpha}_t (I_d - H^{\dagger} H) \Sigma_0 (I_d - H^{\dagger} H) + \bar{\alpha}_t \sigma_y^2 H^{\dagger} (H^{\dagger})^{\mathsf{T}} + (1 - \bar{\alpha}_t) I_d).$$

Note that when $H = (I_p \quad 0)$ and $\sigma_y^2 > 0$, $q_{0|y}$ exists. Define

$$\mu_{t|y} := \sqrt{\bar{\alpha}_t}(I_d - H^{\dagger} H) \mu_0 + \sqrt{\bar{\alpha}_t} H^{\dagger} y$$
$$\Sigma_{t,sig} := \bar{\alpha}_t (I_d - H^{\dagger} H) \Sigma_0 (I_d - H^{\dagger} H) + (1 - \bar{\alpha}_t) I_d$$
$$\Sigma_{t|y} := \Sigma_{t,sig} + \bar{\alpha}_t \sigma_y^2 H^{\dagger} (H^{\dagger})^{\mathsf{T}}. \tag{55}$$

Here $\Sigma_{t,sig}$ is the signal variance at time $t$, and $\Sigma_{t|y}$ is the total variance of the signal and the measurement noise. Note that when $H = (I_p \quad 0)$, $[\Sigma_{t|y}^{-1}]_{\bar{y}\bar{y}} = [\Sigma_{t,sig}^{-1}]_{\bar{y}\bar{y}}$. We also calculate the respective scores of $Q_t$ and $Q_{t|y}$:

$$\nabla \log q_t(x_t) = -\Sigma_t^{-1}(x_t - \mu_t), \quad \nabla \log q_{t|y}(x_t) = -\Sigma_{t|y}^{-1}(x_t - \mu_{t|y}).$$

Since $f_{t,y} = f_{t,y}^*$ (defined in (10)), from (32), the bias at each time is equal to

$$\Delta_{t,y} = (I_d - H^{\dagger} H)(\nabla \log q_{t|y}(x_t) - \nabla \log q_t(x_t))$$
$$= (I_d - H^{\dagger} H) \left( \Sigma_t^{-1}(x_t - \sqrt{\bar{\alpha}_t} \mu_0) - \Sigma_{t|y}^{-1}(x_t - \sqrt{\bar{\alpha}_t} \mu_{0|y}) \right)$$

$$= (I_d - H^\dagger H)\Sigma_t^{-1}(x_t - \sqrt{\bar{\alpha}_t}\mu_0)$$
$$- (I_d - H^\dagger H)\Sigma_{t|y}^{-1}(x_t - \sqrt{\bar{\alpha}_t}(I_d - H^\dagger H)\mu_0 - \sqrt{\bar{\alpha}_t}H^\dagger y).$$

Now, define

$$V_t := (H^\dagger H)\Sigma_0(I_d - H^\dagger H) + (I_d - H^\dagger H)\Sigma_0(H^\dagger H) + (H^\dagger H)\Sigma_0(H^\dagger H)$$
$$A_t := (I_d - H^\dagger H)\Sigma_{t,sig}^{-1}(I_d - H^\dagger H)$$

Thus, we have $\Sigma_t = \Sigma_{t,sig} + \bar{\alpha}_t V_t$ and $\Sigma_{t|y} = \Sigma_{t,sig} + \bar{\alpha}_t \sigma_y^2 H^\dagger (H^\dagger)^\intercal$. By Woodbury matrix identity, for any two matrices $A$ and $B$, their sum can be inversed as $(A + B)^{-1} = A^{-1} - A^{-1}B(A + B)^{-1}$. Thus, we get

$$\Sigma_t^{-1} = (\Sigma_{t,sig} + \bar{\alpha}_t V_t)^{-1} = \Sigma_{t,sig}^{-1} - \bar{\alpha}_t \Sigma_{t,sig}^{-1} V_t \Sigma_t^{-1}$$
$$\Sigma_{t|y}^{-1} = (\Sigma_{t,sig} + \bar{\alpha}_t \sigma_y^2 H^\dagger (H^\dagger)^\intercal)^{-1} = \Sigma_{t,sig}^{-1} - \bar{\alpha}_t \sigma_y^2 \Sigma_{t,sig}^{-1} H^\dagger (H^\dagger)^\intercal \Sigma_{t|y}^{-1}$$
$$\overset{(i)}{=} \Sigma_{t,sig}^{-1} - \bar{\alpha}_t \sigma_y^2 \Sigma_{t,sig}^{-1}(H^\dagger H)\Sigma_{t|y}^{-1}$$

where $(i)$ holds under assumption $H = (I_p \quad 0)$. Thus,

$$\Delta_{t,y} = (I_d - H^\dagger H)\left(\Sigma_t^{-1}(x_t - \sqrt{\bar{\alpha}_t}\mu_0) - \Sigma_{t|y}^{-1}(x_t - \sqrt{\bar{\alpha}_t}\mu_{0|y})\right)$$
$$= (I_d - H^\dagger H)\left(\Sigma_{t,sig}^{-1} - \bar{\alpha}_t \Sigma_{t,sig}^{-1} V_t \Sigma_t^{-1}\right)(x_t - \sqrt{\bar{\alpha}_t}\mu_0)$$
$$- (I_d - H^\dagger H)\left(\Sigma_{t,sig}^{-1} - \bar{\alpha}_t \sigma_y^2 \Sigma_{t,sig}^{-1}(H^\dagger H)\Sigma_{t|y}^{-1}\right)(x_t - \sqrt{\bar{\alpha}_t}(I_d - H^\dagger H)\mu_0 - \sqrt{\bar{\alpha}_t}H^\dagger y)$$
$$\overset{(ii)}{=} (I_d - H^\dagger H)\left(\Sigma_{t,sig}^{-1} - \bar{\alpha}_t \Sigma_{t,sig}^{-1} V_t \Sigma_t^{-1}\right)(x_t - \sqrt{\bar{\alpha}_t}\mu_0)$$
$$- (I_d - H^\dagger H)\Sigma_{t,sig}^{-1}(x_t - \sqrt{\bar{\alpha}_t}(I_d - H^\dagger H)\mu_0 - \sqrt{\bar{\alpha}_t}H^\dagger y)$$
$$= -\bar{\alpha}_t(I_d - H^\dagger H)\Sigma_{t,sig}^{-1} V_t \Sigma_t^{-1}(x_t - \sqrt{\bar{\alpha}_t}\mu_0)$$
$$+ (I_d - H^\dagger H)\Sigma_{t,sig}^{-1}\left((I_d - H^\dagger H)(x_t - \sqrt{\bar{\alpha}_t}\mu_0) + H^\dagger H(x_t - \sqrt{\bar{\alpha}_t}\mu_0)\right)$$
$$- (I_d - H^\dagger H)\Sigma_{t,sig}^{-1}\left((I_d - H^\dagger H)(x_t - \sqrt{\bar{\alpha}_t}\mu_0) + ((H^\dagger H)x_t - \sqrt{\bar{\alpha}_t}H^\dagger y)\right)$$
$$\overset{(iii)}{=} -\bar{\alpha}_t(I_d - H^\dagger H)\Sigma_{t,sig}^{-1} V_t \Sigma_t^{-1}(x_t - \sqrt{\bar{\alpha}_t}\mu_0)$$
$$\overset{(iv)}{=} -\bar{\alpha}_t(I_d - H^\dagger H)\Sigma_{t,sig}^{-1}(I_d - H^\dagger H)\Sigma_0(H^\dagger H)\Sigma_t^{-1}(x_t - \sqrt{\bar{\alpha}_t}\mu_0)$$
$$= -\bar{\alpha}_t A_t \Sigma_0(H^\dagger H)\Sigma_t^{-1}(x_t - \sqrt{\bar{\alpha}_t}\mu_0) \tag{56}$$

where $(ii)$–$(iv)$ hold because $(I_d - H^\dagger H)\Sigma_{t,sig}^{-1}(H^\dagger H) = 0$ by Lemma 9.

Now, since $H^\dagger H = \begin{pmatrix} I_p & 0 \\ 0 & 0 \end{pmatrix}$, we can re-express $A_t$ and $\Delta_{t,y}$ as follows.

$$A_t = (I_d - H^\dagger H)\Sigma_{t,sig}^{-1}(I_d - H^\dagger H)$$
$$= \begin{pmatrix} 0 & 0 \\ 0 & I_{d-p} \end{pmatrix} \begin{pmatrix} (1 - \bar{\alpha}_t)I_p & 0 \\ 0 & \bar{\alpha}_t[\Sigma_0]_{\bar{y}\bar{y}} + (1 - \bar{\alpha}_t)I_{d-p} \end{pmatrix}^{-1} \begin{pmatrix} 0 & 0 \\ 0 & I_{d-p} \end{pmatrix}$$
$$= \begin{pmatrix} 0 & 0 \\ 0 & (\bar{\alpha}_t[\Sigma_0]_{\bar{y}\bar{y}} + (1 - \bar{\alpha}_t)I_{d-p})^{-1} \end{pmatrix}$$
$$\Delta_{t,y} = \begin{pmatrix} 0 \\ -\bar{\alpha}_t(\bar{\alpha}_t[\Sigma_0]_{\bar{y}\bar{y}} + (1 - \bar{\alpha}_t)I_{d-p})^{-1}[\Sigma_0]_{\bar{y}y}[\Sigma_t^{-1}]_{y:}(x_t - \sqrt{\bar{\alpha}_t}\mu_0) \end{pmatrix},$$

and we have

$$\mathbb{E}_{X_t \sim Q_{t|y}}\|\Delta_{t,y}\|^2 = \bar{\alpha}_t^2 \mathbb{E}_{X_t \sim Q_{t|y}}\left\|(\bar{\alpha}_t[\Sigma_0]_{\bar{y}\bar{y}} + (1 - \bar{\alpha}_t)I_{d-p})^{-1}[\Sigma_0]_{\bar{y}y}[\Sigma_t^{-1}]_{y:}(X_t - \sqrt{\bar{\alpha}_t}\mu_0)\right\|^2$$
$$= \bar{\alpha}_t^2 \mathbb{E}_{X_t \sim Q_{t|y}}\mathrm{Tr}\bigg((\bar{\alpha}_t[\Sigma_0]_{\bar{y}\bar{y}} + (1 - \bar{\alpha}_t)I_{d-p})^{-1}[\Sigma_0]_{\bar{y}y}[\Sigma_t^{-1}]_{y:}(X_t - \sqrt{\bar{\alpha}_t}\mu_0)(X_t - \sqrt{\bar{\alpha}_t}\mu_0)^\intercal$$

$$[\Sigma_t^{-1}]_{:y}[\Sigma_0]_{y\bar{y}}(\bar{\alpha}_t[\Sigma_0]_{\bar{y}\bar{y}} + (1-\bar{\alpha}_t)I_{d-p})^{-1}\Big)$$

$$\overset{(v)}{\leq} \bar{\alpha}_t^2 \left\|(\bar{\alpha}_t[\Sigma_0]_{\bar{y}\bar{y}} + (1-\bar{\alpha}_t)I_{d-p})^{-1}\right\|^2 \left\|[\Sigma_t^{-1}]_{y:}\right\|^2 \left\|[\Sigma_0]_{y\bar{y}}[\Sigma_0]_{\bar{y}y}\right\| \mathbb{E}_{X_t \sim Q_{t|y}}\left\|X_t - \sqrt{\bar{\alpha}_t}\mu_0\right\|^2$$
$$(57)$$

where $(v)$ follows from the fact that $|\mathrm{Tr}(UV)| \leq \|U\|\,\mathrm{Tr}(V)$ if $V$ is positive semi-definite. To analyze each norm above, denote $\lambda_1 \geq \cdots \geq \lambda_d > 0$ to be the eigenvalues of $\Sigma_0$, and note that $\|[\Sigma_0]_{\bar{y}y}\| \leq \|\Sigma_0\| = \lambda_1$. The largest eigenvalue of $\Sigma_t^{-1}$ is $(\bar{\alpha}_t\lambda_d + (1-\bar{\alpha}_t))^{-1}$, and note that $\left\|[\Sigma_t^{-1}]_{y:}\right\| \leq \left\|\Sigma_t^{-1}\right\|$. Also, since $[\Sigma_0]_{\bar{y}\bar{y}}$ is positive semi-definite, denote $\tilde{\lambda}_1 \geq \cdots \geq \tilde{\lambda}_{d-p} > 0$ to be its eigenvalues, and thus

$$\left\|(\bar{\alpha}_t[\Sigma_0]_{\bar{y}\bar{y}} + (1-\bar{\alpha}_t)I_{d-p})^{-1}\right\| \leq \frac{1}{\bar{\alpha}_t\tilde{\lambda}_{d-p} + (1-\bar{\alpha}_t)} \leq \frac{1}{\min\{\tilde{\lambda}_{d-p}, 1\}} < \infty.$$

Also, since

$$\mathbb{E}_{X_t \sim Q_{t|y}}(X_t - \sqrt{\bar{\alpha}_t}\mu_0)(X_t - \sqrt{\bar{\alpha}_t}\mu_0)^{\mathsf{T}}$$
$$= \mathbb{E}_{X_t \sim Q_{t|y}}(X_t - \mu_{t|y} + \sqrt{\bar{\alpha}_t}(H^\dagger y - H^\dagger H\mu_0))(X_t - \mu_{t|y} + \sqrt{\bar{\alpha}_t}(H^\dagger y - H^\dagger H\mu_0))^{\mathsf{T}}$$
$$= \Sigma_{t|y} + \bar{\alpha}_t(H^\dagger y - H^\dagger H\mu_0)(H^\dagger y - H^\dagger H\mu_0)^{\mathsf{T}},$$

we have

$$\mathbb{E}_{X_t \sim Q_{t|y}}\left\|X_t - \sqrt{\bar{\alpha}_t}\mu_0\right\|^2 = \mathrm{Tr}(\Sigma_{t|y}) + \bar{\alpha}_t\left\|H^\dagger y - H^\dagger H\mu_0\right\|^2$$
$$\overset{(vi)}{\leq} \bar{\alpha}_t(\lambda_1 + \sigma_y^2) + (1-\bar{\alpha}_t) + \bar{\alpha}_t\left\|H^\dagger y - H^\dagger H\mu_0\right\|^2$$
$$\leq \max\left\{\left\|H^\dagger y - H^\dagger H\mu_0\right\|^2 + d(\lambda_1 + \sigma_y^2), d\right\} \quad (58)$$

where $(vi)$ is because $\Sigma_{t|y}$ is positive definite with

$$\left\|\Sigma_{t|y}\right\| = \left\|\bar{\alpha}_t(I_d - H^\dagger H)\Sigma_0(I_d - H^\dagger H) + \bar{\alpha}_t\sigma_y^2 H^\dagger(H^\dagger)^{\mathsf{T}} + (1-\bar{\alpha}_t)I_d\right\|$$
$$\leq \bar{\alpha}_t\left\|\Sigma_0\right\| + \bar{\alpha}_t\sigma_y^2 + (1-\bar{\alpha}_t)$$
$$= \bar{\alpha}_t(\lambda_1 + \sigma_y^2) + (1-\bar{\alpha}_t).$$

Therefore, for the second moment,

$$\mathbb{E}_{X_t \sim Q_{t|y}}\left\|\Delta_{t,y}\right\|^2 \leq \bar{\alpha}_t^2 \frac{\max\left\{\left\|H^\dagger y - H^\dagger H\mu_0\right\|^2 + d(\lambda_1 + \sigma_y^2), d\right\}}{\min\{\lambda_d, 1\}^2 \min\{\tilde{\lambda}_{d-p}, 1\}^2}\left\|[\Sigma_0]_{y\bar{y}}[\Sigma_0]_{\bar{y}y}\right\|$$
$$\lesssim \bar{\alpha}_t^2 \cdot \max\left\{\left\|H^\dagger y - H^\dagger H\mu_0\right\|^2 + d(\lambda_1 + \sigma_y^2), d\right\}$$

since $\|[\Sigma_0]_{y\bar{y}}[\Sigma_0]_{\bar{y}y}\| \leq \|\Sigma_0\|^2 = \lambda_1^2$. Also, for general moments with $m \geq 2$,

$$\left\|\Delta_{t,y}\right\|^m \leq \bar{\alpha}_t^m \left\|(\bar{\alpha}_t[\Sigma_0]_{\bar{y}\bar{y}} + (1-\bar{\alpha}_t)I_{d-p})^{-1}[\Sigma_0]_{\bar{y}y}[\Sigma_t^{-1}]_{y:}(x_t - \sqrt{\bar{\alpha}_t}\mu_0)\right\|^m$$
$$\leq \bar{\alpha}_t^m \left\|(\bar{\alpha}_t[\Sigma_0]_{\bar{y}\bar{y}} + (1-\bar{\alpha}_t)I_{d-p})^{-1}\right\|^m \left\|\Sigma_0\right\|^m \left\|\Sigma_t^{-1}\right\|^m \left\|x_t - \sqrt{\bar{\alpha}_t}\mu_0\right\|^m$$
$$\lesssim \bar{\alpha}_t^m \left\|x_t - \sqrt{\bar{\alpha}_t}\mu_0\right\|^m$$

and thus

$$\mathbb{E}_{X_t \sim Q_{t|y}}\left\|\Delta_{t,y}\right\|^m \lesssim \bar{\alpha}_t^m \left(\mathbb{E}_{X_t \sim Q_{t|y}}\left\|\Sigma_{t|y}^{-\frac{1}{2}}(X_t - \mu_{t|y})\right\|^m + \left\|\sqrt{\bar{\alpha}_t}(H^\dagger y - H^\dagger H\mu_0)\right\|^m\right)$$
$$\leq \bar{\alpha}_t^m\left((m-1)!! \cdot d^{m/2-1} + \left\|\sqrt{\bar{\alpha}_t}(H^\dagger y - H^\dagger H\mu_0)\right\|^m\right) = O(\bar{\alpha}_t).$$

Therefore, Assumption 4 is satisfied if the $\alpha_t$ satisfies Definition 1. The proof is now complete.

## J.2 PROOF OF LEMMA 8

We continue from the expression of $\Delta_{t,y}$ in (40) when $Q_0$ is Gaussian mixture. For $m \geq 2$, we have

$$\mathbb{E}_{X_t \sim Q_{t|y}} \|\Delta_{t,y}\|^m$$

$$\leq 2^{m-1}(\bar{\alpha}_t)^{m/2} \max_{n \in [N]} \left\|\Sigma_t^{-1}\mu_{0,n}\right\|^m$$

$$+ 2^{m-1}(\bar{\alpha}_t)^m \mathbb{E}_{\substack{X_t \sim Q_{t|y} \\ N \sim \Pi_{\cdot|t,y}(\cdot|X_t)}} \left\|(I_d - H^\dagger H)\Sigma_{t,sig}^{-1}(I_d - H^\dagger H)\Sigma_0(H^\dagger H)\Sigma_t^{-1}(X_t - \sqrt{\bar{\alpha}_t}\mu_{0,N})\right\|^m$$

$$\overset{(ii)}{\lesssim} (\bar{\alpha}_t)^{m/2}d^{m/2} + (\bar{\alpha}_t)^m \mathbb{E}_{\substack{X_t \sim Q_{t|y} \\ N \sim \Pi_{\cdot|t,y}(\cdot|X_t)}} \left\|X_t - \sqrt{\bar{\alpha}_t}\mu_{0,N}\right\|^m$$

$$= (\bar{\alpha}_t)^{m/2}d^{m/2} + (\bar{\alpha}_t)^m \mathbb{E}_{\substack{N \sim \Pi \\ X_t \sim Q_{t,N|y}}} \left\|X_t - \sqrt{\bar{\alpha}_t}\mu_{0,N}\right\|^m$$

$$\overset{(iii)}{\lesssim} (\bar{\alpha}_t)^{m/2}d^{m/2} + (\bar{\alpha}_t)^m \left(d^{m/2} + \sum_{n=1}^N \pi_n \left\|H^\dagger y - H^\dagger H\mu_{0,n}\right\|^m\right) \tag{59}$$

$$= O(\bar{\alpha}_t).$$

Here $(ii)$ follows because

$$\left\|(I_d - H^\dagger H)\Sigma_{t,sig}^{-1}(I_d - H^\dagger H)\Sigma_0(H^\dagger H)\Sigma_t^{-1}\right\| = \left\|(\bar{\alpha}_t[\Sigma_0]_{\bar{y}\bar{y}} + (1 - \bar{\alpha}_t)I_{d-p})^{-1}[\Sigma_0]_{\bar{y}y}[\Sigma_t^{-1}]_{y:}\right\|$$

$$\leq \frac{\lambda_1}{\min\{\tilde{\lambda}_{d-p}, 1\}\min\{\lambda_d, 1\}} = O(1),$$

and $(iii)$ follows from (48). Therefore, this verifies Assumption 4 when the $\alpha_t$ satisfies Definition 1. The proof is now complete.

## J.3 LEMMA 9 AND ITS PROOF

**Lemma 9.** *Given a positive semi-definite matrix $\Sigma$, $\sigma \geq 0$, and $\alpha \in (0, 1)$,*

$$(\alpha\sigma H^\dagger(H^\dagger)^\intercal + (1 - \alpha)I_d)^{-1}(I_d - H^\dagger H) = \frac{1}{1 - \alpha}(I_d - H^\dagger H)$$

$$(I_d - H^\dagger H)(\alpha(I_d - H^\dagger H)\Sigma(I_d - H^\dagger H) + \alpha\sigma H^\dagger H + (1 - \alpha)I_d)^{-1}(H^\dagger H) = 0.$$

*Proof.* The key of the proof is the Woodbury matrix identity, which states that for any matrices $U \in \mathbb{R}^{d \times p}, V \in \mathbb{R}^{p \times d}$,

$$(I_d + UV)^{-1} = I_d - U(I_p + VU)^{-1}V.$$

For the first equality, we apply Woodbury with $U = \sqrt{\frac{\alpha\sigma}{1-\alpha}}H^\dagger$ and $V = \sqrt{\frac{\alpha\sigma}{1-\alpha}}(H^\dagger)^\intercal$ and we get

$$(\alpha\sigma H^\dagger(H^\dagger)^\intercal + (1 - \alpha)I_d)^{-1} = \frac{1}{1 - \alpha}\left(I_d - \frac{\alpha\sigma}{1-\alpha}H^\dagger\left(I_p + \frac{\alpha\sigma}{1-\alpha}(H^\dagger)^\intercal H^\dagger\right)^{-1}(H^\dagger)^\intercal\right).$$

Since $p \leq d$, the pseudo-inverse equals $H^\dagger = H^\intercal(HH^\intercal)^{-1}$, and by the orthogonal property we have

$$(H^\dagger)^\intercal(I_d - H^\dagger H) = (HH^\intercal)^{-1}H(I_d - H^\dagger H) = 0.$$

We have thus shown the first equality.

For the second equality, we first consider the case where $\sigma = 0$. Write $S = (1 - \alpha)I_d + \alpha(I_d - H^\dagger H)\Sigma(I_d - H^\dagger H)$. Since $\Sigma$ is positive semi-definite, there exists matrix $L$ such that $\Sigma = LL^\intercal$. Thus,

$$S^{-1} = \left((1 - \alpha)I_d + \alpha(I_d - H^\dagger H)\Sigma(I_d - H^\dagger H)\right)^{-1}$$

$$= \left((1 - \alpha)I_d + \alpha\left((I_d - H^\dagger H)L\right)\left((I_d - H^\dagger H)^\intercal L\right)^\intercal\right)^{-1}$$

$$= \frac{1}{1-\alpha}I_d - \frac{\alpha}{1-\alpha}((I_d - H^\dagger H)L)\left(I_d + \frac{\alpha}{1-\alpha}L^\mathsf{T}(I_d - H^\dagger H)L\right)^{-1}L^\mathsf{T}(I_d - H^\dagger H),$$

where in the last line we have applied Woodbury with $U = V^\mathsf{T} = \sqrt{\frac{\alpha}{1-\alpha}}(I_d - H^\dagger H)L$. The equality is achieved because

$$(I_d - H^\dagger H)S^{-1}(H^\dagger H)$$
$$= \frac{1}{1-\alpha}\underbrace{(I_d - H^\dagger H)(H^\dagger H)}_{=0}$$
$$- \frac{\alpha}{1-\alpha}(I_d - H^\dagger H)L\left(I_d + \frac{\alpha}{1-\alpha}L^\mathsf{T}(I_d - H^\dagger H)L\right)^{-1}L^\mathsf{T}\underbrace{(I_d - H^\dagger H)(H^\dagger H)}_{=0}$$
$$= 0. \qquad \square$$

When $\sigma > 0$, we can apply Woodbury identity to sum of matrices $A$ and $B$ and get $(A + B)^{-1} = A^{-1} - A^{-1}B(A + B)^{-1}$. Thus,

$$(S + \alpha\sigma H^\dagger H)^{-1} = S^{-1} - \alpha\sigma S^{-1}(H^\dagger H)(S + \alpha\sigma H^\dagger H)^{-1}$$

and

$$(I_d - H^\dagger H)(S + \alpha\sigma H^\dagger H)^{-1}(H^\dagger H)$$
$$= \underbrace{(I_d - H^\dagger H)S^{-1}(H^\dagger H)}_{=0}$$
$$- \alpha\sigma \underbrace{(I_d - H^\dagger H)S^{-1}(H^\dagger H)}_{=0}(S + \alpha\sigma H^\dagger H)^{-1}(H^\dagger H)$$
$$= 0.$$

The proof is now complete.

### J.4 LEMMA 10 AND ITS PROOF

**Lemma 10.** *With $1 - \alpha_t \equiv \frac{\log T}{T}$, $\forall t \geq 0$ (which satisfies Definition 1), given any $p > 0$,*

$$\sum_{t=2}^{T}(1 - \alpha_t)\bar{\alpha}_t^p = \frac{1}{p}\left(1 - \frac{2pc\log T}{T}\right) + \tilde{O}\left(\frac{1}{T^2}\right).$$

*Proof.* Define the sum as $s_T$. Then,

$$s_T = \sum_{t=2}^{T}\frac{c\log T}{T}\left(1 - \frac{c\log T}{T}\right)^{pt} = \frac{c\log T}{T}\left(1 - \frac{c\log T}{T}\right)^{2p}\frac{1 - \left(1 - \frac{c\log T}{T}\right)^{p(T-1)}}{1 - \left(1 - \frac{c\log T}{T}\right)^{p}}$$

$$= \frac{c\log T}{T}\left(1 - \frac{c\log T}{T}\right)^{2p}\frac{1}{1 - \left(1 - \frac{c\log T}{T}\right)^{p}}(1 + O(T^{-cp}))$$

$$= \frac{1}{p}\left(1 - \frac{2pc\log T}{T}\right) + \tilde{O}(T^{-2}). \qquad \square$$

