# OpenReview forum: "Theory on Score-Mismatched Diffusion Models and Zero-Shot Conditional Samplers"
_ICLR.cc/2025/Conference — ICLR 2025 Poster_

### Official Review · Reviewer_n16U · 2024-11-04

**Soundness:** 2
**Presentation:** 2
**Contribution:** 3
**Rating:** 6
**Confidence:** 3

**Summary:**

This paper provides a non-asymptotic convergence theory for diffusion models with a mismatched score function. The results can be applied to zero-shot conditional samplers, especially linear conditional models. The authors also propose the Bias-Optimal (BO) DDNM sampler and provide a comparison with the previous algorithms.

**Strengths:**

- The convergence bound of DDPM under score mismatch applies to a variety of target distributions without restrictive assumptions.
- A bias-optimal zero-shot sampler is proposed, and its convergence rate is analyzed.

**Weaknesses:**

- Assumption 2 requires an upper bound on the score estimation error $\epsilon^2 = \tilde{\mathcal{O}}(T^{-2})$, which could be restrictive compared to the previous literature (e.g., Li et al. 2024b) which applies to any $\epsilon$.
- The in-line equations in Section 2.1 and Section 5 are hard to follow. I suggest the authors re-organize the equations for better readability, especially by highlighting the definitions and differences of $q$, $p$, and $\hat{p}$.

**Questions:**

- The big-O notation in this work is confusing. To name a few, in Assumptions 3 and 4, should $(1-\alpha_t)^m \mathbb{E}[\cdot] = O((1-\alpha_t)^m)$ be equivalent to $\mathbb{E}[\cdot] = O(1)$? Or do the hidden constants in these assumptions implicitly depend on $(1-\alpha_t)^m$? In Definition 1, the noise schedule needs to satisfy $\bar{\alpha}\_T = o (1/T)$ which is defined as an asymptotic bound $\lim\sup_{T\to\infty} |\bar{\alpha}_T / (1/T)| \to 0$, while Theorem 1 presents a non-asymptotic analysis. How does the asymptotic assumption apply to the non-asymptotic analysis? I suggest the authors clarify the notations or even explicitly write the constants' dependency on the hyperparameters if possible.
- Does $1-\alpha_1=\delta$ in equation (8) contradict with $1-\alpha_1 \lesssim \frac{\log T}{T}$ in Definition 1? What can we obtain from Theorem 1 if $\alpha_t$ is chosen as (8)?

---

> ### Author Response · Authors · 2024-11-21
>
> We thank the reviewer for providing the highly inspiring feedback. In the revised paper, we have made changes based on your review comments and have highlighted all revisions in blue fonts.
>
> **Q1:** Assumption 2 requires an upper bound on the score estimation error $\epsilon = \tilde{O}(T^{-2})$, which could be restrictive compared to the previous literature (e.g., Li et al. 2024b) which applies to any $\epsilon$.
>
> **A1:** We clarify that for score-**mismatched** scenarios (which our paper considers),  the score estimation error needs to be achieved at higher accuracy than the score-**matched** DDPM (e.g., [1]), because an extra term proportional to $\epsilon$ arises when there is score mismatch (see Lemma 2). Such a higher level of estimation accuracy also occurs in theoretical studies for accelerated DDPM samplers (e.g., [1, Theorem 3.2]). We also made the clarification in our revised draft.
>
> **Q2:** The in-line equations in Section 2.1 and Section 5 are hard to follow. I suggest the authors re-organize the equations for better readability, especially by highlighting the definitions and differences of $q$, $p$, and $\hat{p}$.
>
> **A2:** We have clarified the difference of notations in Section 2.2 and before Theorem 1. In particular, $Q$ is the distribution used for training the score, $\tilde{Q}$ is the target distribution, and $P$ and $\widehat{P}$ represent the sampling distribution with and without estimation error, respectively. We have also reduced the number of inline equations in Section E (prev. Section 5).
>
> **Q3:** The big-O notation in this work is confusing. To name a few, in Assumptions 3 and 4, should $(1-\alpha_t)^m \mathbb{E}[\cdot] = O((1-\alpha_t)^m)$ be equivalent to $\mathbb{E}[\cdot] = O(1)$? Or do the hidden constants in these assumptions implicitly depend on $(1-\alpha_t)^m$?
>
> **A3:** Thank you for your suggestion. We have made our Assumptions 3 and 4 more clear by removing the factors. In particular, Assumptions 3 and 4 can still be satisfied under two cases: (i) where $Q_0$ has bounded support for any $H$, with a special $\alpha_t$ in Equation (8) (see the proof of Theorem 4); or (ii) where $Q_0$ is Gaussian mixture and $H = [I_p, 0]$ (see Lemma 8). We have made more clarifications after stating the assumptions. Please check out the revised draft.
>
> **Q4:** In Definition 1, the noise schedule needs to satisfy $\bar{\alpha}_T = o(1/T)$ which is defined as an asymptotic bound $\lim \sup  | \bar{\alpha}_T / (1/T) | = 0$, while Theorem 1 presents a non-asymptotic analysis. How does the asymptotic assumption apply to the non-asymptotic analysis?
>
> **A4:** Such an analysis trick has been used widely in prior literature (say, [3]); namely “non-asymptotic analysis” is applied to the regime where $T$ is large enough but finite. When $T$ is large enough, $\bar{\alpha}_T = o(1/T)$ implies that for all constant $\epsilon<\infty$ we have $\bar{\alpha}_T < \epsilon/T$. Thus, this guarantees that the initialization error does not contribute to the rate-determining terms in Theorem 1 (see Line 1057).
>
> **Q5:** I suggest the authors clarify the notations or even explicitly write the constants' dependency on the hyperparameters if possible.
>
> **A5:** Thank you for your suggestions! We have made our Assumptions 3 and 4 more clear by removing the factors. Please check out our revised draft. Also note that in Theorem 1, the $\lesssim$ only omits terms that decay faster than the rate-determining terms being displayed; there are no other hyperparameter-related coefficients/constants hidden in these rate-determining terms themselves.
>
> **Q6:** Does $1-\alpha_1 = \delta$ in equation (8) contradict with $1-\alpha_1 \lesssim \log T / T$ in Definition 1? What can we obtain from Theorem 1 if $\alpha_t$ is chosen as (8)?
>
> **A6:** Thank you for pointing this out! We agree that the $\alpha_t$ in Equation (8) does not satisfy $1-\alpha_1 \lesssim \log T / T$ at time $t=1$. However, Corollary 1 only requires such $\alpha_t$ where Definition 1 is satisfied at $t \geq 2$. We have clarified the requirement on $\alpha_t$ in Corollary 1 in the appendix.
>
> We sincerely thank the reviewer once again for your highly inspiring comments. We hope our responses have effectively addressed your concerns. If so, we kindly ask if you might consider raising your score. Of course, we would be happy to address any additional questions or comments you may have.

---

> > ### Author Response · Authors · 2024-11-21
> >
> > [1] G. Li, Y. Huang, T. Efimov, Y. Wei, Y. Chi, and Y. Chen. Accelerating Convergence of Score-Based Diffusion Models, Provably. Proceedings of the 41st International Conference on Machine Learning. 2024.
> >
> > [2] Y. Liang, P. Ju, Y. Liang, and N. Shroff. Broadening Target Distributions for Accelerated Diffusion Models via a Novel Analysis Approach. arXiv preprint arXiv:2402:13901.
> >
> > [3] G. Li, Y. Wei, Y. Chen, and Y. Chi. Towards Non-Asymptotic Convergence for Diffusion-Based Generative Models. The Twelfth International Conference on Learning Representations. 2024.
> >
> > [4] J. Benton, V. D. Bortoli, A. Doucet, and G. Deligiannidis. Nearly $d$-linear convergence bounds for diffusion models via stochastic localization. The Twelfth International Conference on Learning Representations. 2024.
> >
> > [5] H. Chen, H. Lee, and J. Lu. Improved analysis of score-based generative modeling: user-friendly bounds under minimal smoothness assumptions. Proceedings of the 40th International Conference on Machine Learning. 2023.

---

> > > ### Comment · Reviewer_n16U · 2024-11-24
> > > **Thank you for your response**
> > >
> > > Thank you for your response, which clarified most of my concerns. I have updated my rating.

---

### Official Review · Reviewer_oUs6 · 2024-11-07

**Soundness:** 3
**Presentation:** 3
**Contribution:** 2
**Rating:** 8
**Confidence:** 2

**Summary:**

This work provides the first guarantees for conditional sampling (within the inverse problems setting) via diffusion models, which approximate the guidance term via techniques that do not require additional supervision (e.g., x,y pairs such as in CG and CFG), which the authors coin zero-shot as they do not require to observe any additional data to fine-tune.

The authors then provide a preconditioned linear-approximation-based approach to their proposed decomposition of the posterior score, which minimises the mismatch; the proposed estimator is very cheap and could be useful to the community. Furthermore, the authors provide a very modest numerical benchmark showing significant improvements to denoising diffusion null space models. In general, the strategy/sketch used in Theorem 3 can be of great use as a template to practitioners looking to derive quick yet accurate estimators.

**Strengths:**

1. The paper is clear and well written
2. The short summary sketches of the proofs allow us to for a high-level verification in the soundness of the results
3. The work is novel and clearly relevant to the community; with some small modifications and clarifications, this work can be impactful.

**Weaknesses:**

1. The reviewer found the presentation of the paper a bit confusing. Calling methodologies such as reconstruction guidance (Chung 2023 in the paper) zero-shot (whilst potentially accurate) can be confusing and misleading for the following reasons:


      *  Whilst many of these methodologies do not require additional supervision (e.g. observing x,y pairs such as in CFG or classifier guidance) they do often still approximate the same quantity e.g. the h-transform / guidance term [1], ultimately with the goal of approximating the posterior score.

     *  Many of these zero shot approximations induce costs that are comparable to training when one considers (see wallclock times in [1]) , furthermore with methods like DPS / reconstruction guidance we can see that they are effectively fine-tuning/minimising a reconstruction loss across timesteps, whilst there's no extra data (thus arguably zero shot) , there is certainly a significant extra cost arising from this "online-finetuning".

In short, I think the paper would improve its reach/impact if it revised its presentation of what it considers to be zero-shot, or make it much more precise from earlier in the manuscript.


2. Assumption 4 seems quite strong and rather non-practical for some of these zero-shot methodologies, which make Gaussian approximations on the reverse process and thus do not induce a mismatch that can be controlled.


[1] Denker, A., Vargas, F., Padhy, S., Didi, K., Mathis, S., Dutordoir, V., Barbano, R., Mathieu, E., Komorowska, U.J. and Lio, P., 2024. DEFT: Efficient Finetuning of Conditional Diffusion Models by Learning the Generalised $ h $-transform. arXiv preprint arXiv:2406.01781.

**Questions:**

1. I am slightly confused by theorem 2 following from theorem 1 (more specifically, Corollary 1 following from theorem 1).  Theorem 2 bounds the KL between Q_1 (the marginal distribution of the zero-shot sampler at time 1) and P_1 (e.g. the time 1 marginal of the VP-SDE at time 1 where P_0 is the conditional distribution). Typically for all practical purposes in diffusion models Q_1 is simply a Gaussian and this bound can be obtained from exploring the ergodicity of the OU-process and prior work on the convergence of diffusion models. So two questions

     * Why is this bound for a general Q_1 relevant? We typically fix/choose Q_1 , so Im a bit confused about what specific aspect of the zero-shot diffusion models considered in this work induces/requires a particular Q_1 which needs to be analysed
     * How does corollary 1 follow from theorem 2 , do you simply just preform the same analysis in reverse time ?  (e.g. flip the time indices) more detail here would be useful

2. How realistic is assumption 4 for zero-shot models such as Chung 2022a-b, Chung 2023 where the score mismatch cant really be controlled as it makes rather unrealistic Gaussian approximations of the transition density in the backwards process.  Assumption 4 is a rather specific bound on the mismatch it seems its form is required to make everything else work so discussion on its relevance outside of a Gaussian setting for the prior (data distribution) would be extremely helpful.

---

> ### Author Response · Authors · 2024-11-21
>
> We thank the reviewer for providing the highly inspiring feedback. In the revised paper, we have made changes based on your review comments and have highlighted all revisions in blue fonts.
>
> **Q1:** The reviewer found the presentation of the paper a bit confusing. Calling methodologies such as reconstruction guidance (Chung 2023 in the paper) zero-shot (whilst potentially accurate) can be confusing and misleading for the following reasons:
>
> - Whilst many of these methodologies do not require additional supervision (e.g. observing x,y pairs such as in CFG or classifier guidance) they do often still approximate the same quantity e.g. the h-transform / guidance term [1], ultimately with the goal of approximating the posterior score.
>
> - Many of these zero shot approximations induce costs that are comparable to training when one considers (see wallclock times in [1]) , furthermore with methods like DPS / reconstruction guidance we can see that they are effectively fine-tuning/minimising a reconstruction loss across timesteps, whilst there's no extra data (thus arguably zero shot) , there is certainly a significant extra cost arising from this "online-finetuning".
>
> In short, I think the paper would improve its reach/impact if it revised its presentation of what it considers to be zero-shot, or make it much more precise from earlier in the manuscript.
>
> **A1:** We greatly appreciate your detailed discussion of the pros and cons of different conditional guidance methods. We employ the name “zero-shot” to highlight the fact that the approximated guidance term in our paper does *not* require extra training based on conditional information, which is different from other popular methods such as classifier guidance and classifier-free guidance (CFG). This terminology also aligns with some previous surveys in image editing (see [2, Section 7.1.2] and [3, Section V.E]). We have clarified our definition for “zero-shot” in our revised draft with the remark that some zero-shot methods might have additional computational costs during sampling.
>
> While it is true that methods like DPS require “online-finetuning” that increases computation cost, there are some other zero-shot methods, including the DDNM [4] which is of primary focus in this paper, that take advantage of the linear conditional model and thus have similar computation complexity as the vanilla (unconditional) DDPM [5] (see [4, Alg. 1]). We agree that a theoretical convergence analysis of the samplers using finetuning as in [1] is an interesting future direction.
>
> **Q2:** Assumption 4 seems quite strong and rather non-practical for some of these zero-shot methodologies, which make Gaussian approximations on the reverse process and thus do not induce a mismatch that can be controlled.
>
> **A2:** We respectfully disagree with the reviewer’s claim that Assumption 4 is strong. In our paper, Assumption 4 has been established in two cases of zero-shot conditional sampling: (1) where $Q_0$ has bounded support for any $H$, using a special $\alpha_t$ in Equation (8) (see the proof of Theorem 4); or (2) where $Q_0$ is Gaussian mixture and $H = [I_p, 0]$ (see Lemma 8). Here the first case has wide applicability in practice (e.g., images [2-5]) and is commonly assumed in many theoretical investigations (e.g., [6]). We have made more clarifications in the revised draft after stating the assumptions.

---

> ### Author Response · Authors · 2024-11-21
>
> **Q3:** I am slightly confused by theorem 2 following from theorem 1 (more specifically, Corollary 1 following from theorem 1). Theorem 2 bounds the KL between $Q_1$ (the marginal distribution of the zero-shot sampler at time 1) and $P_1$ (e.g. the time 1 marginal of the VP-SDE at time 1 where $P_0$ is the conditional distribution). Typically for all practical purposes in diffusion models $Q_1$ is simply a Gaussian and this bound can be obtained from exploring the ergodicity of the OU-process and prior work on the convergence of diffusion models. So two questions:
>
> **Question 1**: Why is this bound for a general $Q_1$ relevant? We typically fix/choose $Q_1$ , so I’m a bit confused about what specific aspect of the zero-shot diffusion models considered in this work induces/requires a particular $Q_1$ which needs to be analysed.
>
> **Question 2**: How does corollary 1 follow from theorem 2, do you simply just perform the same analysis in reverse time? (e.g. flip the time indices) more detail here would be useful
>
> **A3:** We first clarify our notations, which largely follows from [5]. In particular, $Q$ is the distribution used for training the score, $\tilde{Q}$ is the target distribution, and $P$ and $\widehat{P}$ represent the sampling distribution with and without estimation error, respectively. In the forward process, $Q_t$ is the Gaussian perturbed version of the training data distribution $Q_0$, and $Q_T$ is very close to a standard Gaussian. In the reverse sampling process, we start from $x_T \sim \widehat{P}_T = \mathcal{N}(0,I_d)$ and goes *backward* in time $t$ to sequentially obtain $x_T,\dots,x_1$. Also note that $t$ is the index on the number of steps, which is different from the SDE diffusion time.
>
> **Regarding Question 1**: Since we go backward in time in the reverse process, we fix/choose $\widehat{P}_T = \mathcal{N}(0,I_d)$ and provide a convergence upper bound for $\mathrm{KL}(\tilde{Q}_0||\widehat{P}_0)$ in Theorem 1. Here Theorem 1 characterizes how close the final output distribution $\widehat{P}_0$ is to the target distribution $\tilde{Q}_0$. However, for general target distributions, $\mathrm{KL}(\tilde{Q}_0||\widehat{P}_0)$ might be undefined for non-smooth $\tilde{Q}_0$ (e.g., images). Thus, in Corollary 1 and Theorem 2, we provide an upper bound using the early-stopped version $\widehat{P}_1$. In words, Corollary 1 and Theorem 2 say that the early-stopped output distribution ($\widehat{P}_1$) will be close to some distribution ($\tilde{Q}_1$), which itself is a slight perturbation of the true target distribution ($\tilde{Q}_0$). Note that such KL-Wasserstein guarantees are common for early-stopped procedures in previous theoretical investigations (e.g., [7]).
>
> **Regarding Question 2**: First, our Corollary 1 follows from Theorem 1 because the Gaussian perturbed distribution $\tilde{Q}_1$ both exists and is analytic. Then, Theorem 2, whose proof is based on Corollary 1, shows the explicit dimensional dependency with a particular choice of noise schedule in Equation (8). We will provide more details in our revised draft.
>
> **Q4:** How realistic is assumption 4 for zero-shot models such as Chung 2022a-b, Chung 2023 where the score mismatch can’t really be controlled as it makes rather unrealistic Gaussian approximations of the transition density in the backwards process. Assumption 4 is a rather specific bound on the mismatch; it seems its form is required to make everything else work so discussion on its relevance outside of a Gaussian setting for the prior (data distribution) would be extremely helpful.
>
> **A4:** Please see our answer in A2.
>
> We deeply appreciate the reviewer’s thoughtful and inspiring feedback. We hope our responses have satisfactorily addressed your concerns. If so, we kindly ask you to consider raising your score. Please don’t hesitate to reach out with any additional questions or comments—we would be delighted to address them.

---

> > ### Author Response · Authors · 2024-11-21
> >
> > [1] A. Denker, F. Vargas, S. Padhy, K. Didi, S. Mathis, V. Dutordoir, R. Barbano, E. Mathieu, U. J. Komorowska, and P. Lio. DEFT: Efficient Finetuning of Conditional Diffusion Models by Learning the Generalised $h$-transform. arXiv preprint arXiv:2406.01781.
> >
> > [2] Y. Huang, J. Huang, Y. Liu, M. Yan, J. Lv, J. Liu, W. Xiong, H. Zhang, S. Chen, and L. Cao. Diffusion Model-Based Image Editing: A Survey. arXiv preprint arXiv:2402.17525.
> >
> > [3] B. B. Moser, A. S. Shanbhag, F. Raue, S. Frolov, S. Palacio, and A. Dengel. Diffusion Models, Image Super-Resolution, and Everything: A Survey. IEEE Transactions on Neural Networks and Learning Systems. 2024.
> >
> > [4] Y. Wang, J. Yu, and J. Zhang. Zero-Shot Image Restoration Using Denoising Diffusion Null-Space Model. The Eleventh International Conference on Learning Representations. 2023.
> >
> > [5] J. Ho, A. Jain, and P. Abbeel. Denoising diffusion probabilistic models. Advances in
> > Neural Information Processing Systems. 2020.
> >
> > [6] G. Li, Y. Wei, Y. Chen, and Y. Chi. Towards Non-Asymptotic Convergence for Diffusion-Based Generative Models. The Twelfth International Conference on Learning Representations. 2024.
> >
> > [7] H. Chen, H. Lee, and J. Lu. Improved analysis of score-based generative modeling: user-friendly bounds under minimal smoothness assumptions. Proceedings of the 40th International Conference on Machine Learning. 2023.

---

> > > ### Comment · Reviewer_oUs6 · 2024-11-25
> > > **followup question**
> > >
> > > >  We respectfully disagree with the reviewer’s claim that Assumption 4 is strong. In our paper, Assumption 4 has been established in two cases of zero-shot conditional sampling: ...
> > >
> > > What do the reviewers mean by established? Unfortunately, I don't quite follow their answer; I can't understand how it addresses my question. This may be my fault. I will spend some more time going over this, but if the reviewers can, I would appreciate a bit more detail in their response (don't be afraid to spell it out, so to say).
> > >
> > > Let me quickly repeat my question, my concern with this assumption is that I don't understand how Assumption 4 is practical for a non-Gaussian target distribution (I appreciate the Gaussian mixture example I will review it more closely). More precisely  these zero-shot / linear approximations of the guidance term typically make a Gaussian assumption on the transition density of the reverse/backwards SDE see equation 8-9 here https://arxiv.org/pdf/2306.07526 , only with approximations of the sort can one arrive at these linear / zero-shot corrections (there is a true guidance term and that is not linear).
> > >
> > > Such Gaussian approximation cannot control the error due to its lack of expressivity so Im still at odds in understanding how assumption 4 which looks rather controllable is practical.
> > >
> > > Thanks to the reviewers for their time and I look forward to understanding this detail better.

---

> > > > ### Author Response · Authors · 2024-11-26
> > > >
> > > > **Q1:** What do the reviewers mean by established? Unfortunately, I don't quite follow their answer; I can't understand how it addresses my question. This may be my fault. I will spend some more time going over this, but if the reviewers can, I would appreciate a bit more detail in their response (don't be afraid to spell it out, so to say).
> > > >
> > > > **A1:** We apologize if there is any unclarity in our explanation, and we greatly appreciate the reviewer’s follow-up question. We hope that our answers below could alleviate your concern.
> > > >
> > > > **Q2:** Let me quickly repeat my question. My concern with this assumption is that I don't understand how Assumption 4 is practical for a non-Gaussian target distribution (I appreciate the Gaussian mixture example; I will review it more closely).
> > > >
> > > > **A2:** Assumption 4 roughly says that the norm of the mismatch between the posterior (i.e., the target) and the approximate score at each time is *bounded* in $T$ in expectation (see our definition of $\Delta_t$ in Equation (7)). For linear conditional models, the posterior (or target) score at time $t$ can be decomposed into a linear term (in $y$ and $x_t$) and an expectation term of $X_0$ under the posterior (or target) distribution (see our Theorem 3). Here the linear term is obtained in a zero-shot way for our sampler and thus does not contribute to the total mismatch. To approximate the expectation term, we simply replace the expected value of $X_0$ under the posterior (or target) distribution with that under the marginal (or training) distribution. Now, for the two cases considered ($Q_0$ bounded support and Gaussian mixture), all norms of $X_0$ are bounded in expectation, which guarantee that Assumption 4 holds. Notably, we do **not** need $Q_0$ to be distributed as a Gaussian.
> > > >
> > > > **Q3:** More precisely these zero-shot / linear approximations of the guidance term typically make a Gaussian assumption on the transition density of the reverse/backwards SDE see equation 8-9 here https://arxiv.org/pdf/2306.07526 , only with approximations of the sort can one arrive at these linear / zero-shot corrections (there is a true guidance term and that is not linear).
> > > >
> > > > **A3:** To clarify, in order to approximate the posterior score, we are not following the approach taken in https://arxiv.org/pdf/2306.07526, which decomposes the score into the unconditional score plus some “guidance” term (in particular, see Equation (4) in that paper), where this “guidance” term is approximated with some Gaussian approximation. Instead, for *linear* conditional models, we first decompose the posterior score into a linear term (in $y$ and $x_t$) and an expectation term under the posterior through direct evaluation (see Theorem 3). Then, the approximation is made only on the posterior expectation. This term does not have a one-to-one correspondence to the suggested “guidance” term. Therefore, since the way in which the decomposition is done is different, **we do not require any Gaussian approximation** for the distribution of $X_0$ given $X_t$.
> > > >
> > > > The decomposition in our paper follows the idea of DDNM [4]. There, instead of using some “guidance” term, [4] modifies the sampling rule of the vanilla DDPM [5] by projecting $\hat{x}_0$ according to the range and null space of $H$. We have provided a detailed introduction of [4] and showed that this range-null space decomposition is equivalent to ours in Appendix D.2.
> > > >
> > > > **Q4:** Such a Gaussian approximation cannot control the error due to its lack of expressivity so I'm still at odds in understanding how assumption 4 which looks rather controllable is practical.
> > > >
> > > > **A4:** Please see our response in A3. Since we do not decompose the posterior score into some “guidance” term, there is no need to perform Gaussian approximation to obtain the guidance expression.
> > > >
> > > > We hope that our response above has sufficiently addressed your concern. If so, we kindly ask you to consider raising your score. Also, feel free to ask any other follow-up question if you have any.

---

> ### Comment · Reviewer_oUs6 · 2024-11-27
> **Thank you for the detailed explantion**
>
> The authors have clarified my concerns and initial misunderstanding about their posterior score decomposition (and approximation) and made the paper more accessible, so I have updated my score.
>
> I have also re-calibrated my confidence to account for my initial misunderstanding of their work. While I believe I have a better understanding now, the initial confusion impacted my first review, and thus, I should have assigned lower confidence.
>
> I would like to thank the authors for their patience and careful explanation of their work during the rebuttal process.

---

### Official Review · Reviewer_n4bv · 2024-11-08

**Soundness:** 3
**Presentation:** 2
**Contribution:** 2
**Rating:** 6
**Confidence:** 2

**Summary:**

This paper mainly focuses on the convergence analysis of zero-shot samplers when training scores are based on the unconditional distribution while inference samples for a conditional distribution. Under these conditions, the inference required scores vary from the predicted one, which is considered as the score-mismatched diffusion.
Although various practical works have appeared, from a theoretical perspective, this is the first work to analyze the inference convergence under score-mismatched settings. The core step of this paper is to recognize and minimize (Theorem 3) the mismatch scores (Eq.(7)) by considering the conditional reverse process (Eq.(5)). The conditional reverse process is deduced by a known connection (e.g., linear connection) between the random variables and the conditions (Eq.(4)).

**Strengths:**

1. The problem setting considered in this paper is interesting, especially regarding the effectiveness of the zero-shot sampler in the score-mismatched diffusion.
2. Authors denote the gap between ground truth and generated samples lies in the accumulative mismatch between the target and training distributions. This qualitative result is useful and aligns with their sampler design.

**Weaknesses:**

1. I am a bit afraid that this paper overclaims the problem settings it actually solved. Specifically, this paper claims to consider a general score-mismatched setting. Although it provides a convergence analysis (Theorem 1) for such a setting, the assumption, e.g., (Assumption 4) required, is highly artificial. Besides, it can hardly find an algorithm to achieve Theorem 1, since authors do not provide any approximation for $\grad \ln \tilde{q}_t(x_t)$ under a general score-mismatched setting ($y=h(x_0)$). Instead, I think this paper more focused on the linear setting, i.e., $y=Hx_0+n$ (Eq.(4))
2. I am curious about whether the problem setting is really useful beyond the super-resolution problem. Since the conditioning (y) is usually correlated with the random variable (x) implicitly. I suggest authors provide more examples of utilizing zero-shot samplers to solve score-mismatch diffusions.
3. The establishment of Assumption 4 requires strict assumptions. It not only constrains the unconditional distribution $Q_0$ but also constrains the $H=[I_p, 0]$. I hope the authors can demonstrate whether the results can extend to a general $H$ and how.
4. The final KL divergence bound between ground truth and generated samples is unsatisfactory. Although it is an upper bound, it contains a factor O(d). Such a result will make me concern whether a zero-shot sampler is really effective for mismatch-score settings.

**Questions:**

1. Theorem 4 is somewhat confusing. Specifically, it claims the establishment of Theorem 2 under the specific choice of $f_{t,y}$, while Theorem 2 has additional assumptions for $\E[\|\Delta_t(X_t)\|]$. The problem is whether the upper bound of the mismatch-score, i.e., $\E[\|\Delta_t(X_t)\|]$, is required for Theorem 4 when the conditional target distribution is general.
2. The BO-DDNM is too dependent on the linear structure of $y=Hx+n$. Considering a general mismatch-score setting, i.e., $y=h(x_0)$. How can we adapt BO-DDNM to the general case except for a Taylor extension?

I am willing to raise my score if the authors solve my concerns.

---

> ### Author Response · Authors · 2024-11-21
>
> We thank the reviewer for providing the highly inspiring feedback. In the revised paper, we have made changes based on your review comments and have highlighted all revisions in blue fonts.
>
> **Q1:** I am a bit afraid that this paper overclaims the problem settings it actually solved. Specifically, this paper claims to consider a general score-mismatched setting. Although it provides a convergence analysis (Theorem 1) for such a setting, the assumption, e.g., (Assumption 4) required, is highly artificial. Besides, it can hardly find an algorithm to achieve Theorem 1, since authors do not provide any approximation for $\nabla \ln \tilde{q}_t$ under a general score-mismatched setting ($y=h(x_0)$). Instead, I think this paper more focused on the linear setting, i.e., $y = Hx_0 + n$ (Eq.(4)).
>
> **A1:** We respectfully disagree with the reviewer. (i) Our score-mismatched analysis is indeed applicable to general conditional models, including non-linear ones, as long as there is some mismatch in the scores. In particular, Assumption 4 naturally captures to what extend the mismatch is, which needs to be bounded (in the $L^\ell(\tilde{Q}_t)$ space) so that accumulating $t=1,\dots,T$ the bias will not diverge. Note that Assumption 4 does not assume a linear conditional model. (ii) We acknowledge that it is hard to find an algorithm to achieve Theorem 1 for the general score-mismatched setting ($y=h(x_0)$). But our paper did not overclaim that we designed such an algorithm.
>
> We re-emphasize that our contribution lies in convergence analysis, starting from a general scenario and specializing to the linear setting. And further, we provide a bias-optimal sampler design for the linear case. We believe what we claim in the paper matches the results we develop.
>
> **Q2:** I am curious about whether the problem setting is really useful beyond the super-resolution problem. Since the conditioning (y) is usually correlated with the random variable (x) implicitly. I suggest authors provide more examples of utilizing zero-shot samplers to solve score-mismatch diffusions.
>
> **A2:** The linear model is useful for a number of computer vision applications, including image super-resolution, image inpainting (where $H$ is 0-1 diagonal in both cases), image colorization (where $H$ is a pixel-wise operator with $[1/3, 1/3, 1/3]$), image deblurring (where $H$ is the blurring filter) (see [1, Section D] and [2, Section 3.2]). The linear model is also useful for medical image reconstruction [3], where $H$ corresponds to the phases from Fourier Transform. The linear model is also the main focus in many empirical zero-shot diffusion samplers, such as DDRM [1], DDNM [2], MCG [4], and $\Pi$GDM [5]. These facts inspire us to focus primarily on linear conditional models.
>
> **Q3:** The establishment of Assumption 4 requires strict assumptions. It not only constrains the unconditional distribution $Q_0$ but also constrains the $H = [I_p, 0]$. I hope the authors can demonstrate whether the results can extend to a general $H$ and how.
>
> **A3:** Assumption 4 is not restrictive but rather soft. In our paper, Assumption 4 has be established in two cases: (1) where $Q_0$ has bounded support with a special $\alpha_t$ in Equation (8) (see the proof of Theorem 4); or (2) where $Q_0$ is Gaussian mixture and $H = [I_p, 0]$ (see Lemma 8). In the first case, we do not require any particular form for $H$. Also here, the assumption that $Q_0$ has bounded support has wide applicability in practice (e.g., images [1-5]) and is commonly assumed in many theoretical investigations (e.g., [6]). For the second case, the proof of Lemma 8 can be easily extended for general $H$ (since $||H^\dagger H|| = ||I_d - H^\dagger H|| = 1$). We have added this clarification in the revised draft.
>
> **Q4:** The final KL divergence bound between ground truth and generated samples is unsatisfactory. Although it is an upper bound, it contains a factor $O(d)$. Such a result will make me concern whether a zero-shot sampler is really effective for mismatch-score settings.
>
> **A4:** On the theory side, linear dimensional dependency is established in many state-of-the-art analyses of score-matched DDPMs (say, [7,8]). While there are some works to further shrink $d$ dependency by assuming an underlying low-dimensional structure, the current factor of $O(d)$ is considered to be satisfactory when such a low-dimensional structure is not assumed. Also, in empirical studies, the performance of zero-shot samplers, in particular that of DDNM [2], has been shown to be very effective. Along this line of research, our BO-DDNM further reduces the bias of DDNM both theoretically and empirically.

---

> > ### Author Response · Authors · 2024-11-21
> >
> > **Q5:** Theorem 4 is somewhat confusing. Specifically, it claims the establishment of Theorem 2 under the specific choice of $f_{t,y}$, while Theorem 2 has additional assumptions for $\mathbb{E} ||\Delta_t(X_t)||^2$. The problem is whether the upper bound of the mismatch-score, i.e., $\mathbb{E} ||\Delta_t(X_t)||^2$, is required for Theorem 4 when the conditional target distribution is general.
> >
> > **A5:** To clarify, the first part of the proof of Theorem 4 is to establish Assumption 4 in order to invoke Theorem 2. There, we established Assumption 4 for two different $f_{t,y}$’s when $Q_0$ has bounded support and using a particular choice of $\alpha_t$. In the first scenario, Assumption 4 can be satisfied with $f^*_{t,y}$ (see Equation (35)) for general $H$. In the second scenario, Assumption 4 can be satisfied with $f^N_{t,y}$ when $H^\dagger \lesssim 1$. Note that this latter condition holds in many applications, such as in image super-resolution (where $H^\dagger$ is a diagonal matrix with 0 or 1's with $||H^\dagger|| = 1$) and in image deblurring and colorization (where $H^\dagger$ is a block-diagonal matrix, whose spectral norm is independent of $d$).
> >
> > **Q6:** The BO-DDNM is too dependent on the linear structure of $y = Hx_0 + n$. Considering a general mismatch-score setting, i.e., $y=h(x_0)$. How can we adapt BO-DDNM to the general case except for a Taylor extension?
> >
> > **A6:** Good question! First note that recovery from general non-linear models is still an extremely challenging open problem, and even empirical studies are quite limited for this regime. One recent attempt in the empirical studies is to propose pseudo-inverse based samplers for some examples of non-linear models (e.g., in [5, Section 3.2]). There, an $h^\dagger$ function is used for applications like quantization (where $h(x) = \lfloor x \rfloor$ and $h^\dagger(x) = x$) and JPEG encoding (where $h^\dagger(x)$ corresponds to the JPEG decoding algorithm). With our analyzing approach, we can first identify the asymptotic bias of the algorithms using Theorem 1, which is applicable to general non-linear models. Then, we can similarly minimize the bias to obtain some bias-optimal sampler. This is a potential solution to adapt the BO-DDNM sampler to non-linearity, and will be an exciting problem to study in the future.
> >
> > We sincerely thank the reviewer once again for your insightful and inspiring comments. We hope our responses have adequately addressed your concerns. If so, we kindly ask if you might consider increasing your score. Of course, we would be more than happy to address any further questions or concerns you may have.
> >
> > [1] B. Kawar, M. Elad, S. Ermon, and J. Song. Denoising diffusion restoration models. Proceedings of the 36th International Conference on Neural Information Processing Systems. 2024.
> >
> > [2] Y. Wang, J. Yu, and J. Zhang. Zero-Shot Image Restoration Using Denoising Diffusion Null-Space Model. The Eleventh International Conference on Learning Representations. 2023.
> >
> > [3] A. Jalal, M. Arvinte, G. Daras, E. Price, A. G. Dimakis, and J. I. Tamir. Robust Compressed Sensing {MRI} with Deep Generative Priors. Advances in Neural Information Processing Systems. 2021.
> >
> > [4] H. Chung, B. Sim, D. Ryu, and J. C. Ye. Improving Diffusion Models for Inverse Problems using Manifold Constraints. Advances in Neural Information Processing Systems. 2022.
> >
> > [5] J. Song, A. Vahdat, M. Mardani, and J. Kautz. Pseudoinverse-Guided Diffusion Models for Inverse Problems. The Eleventh International Conference on Learning Representations. 2023.
> >
> > [6] G. Li, Y. Wei, Y. Chen, and Y. Chi. Towards Non-Asymptotic Convergence for Diffusion-Based Generative Models. The Twelfth International Conference on Learning Representations. 2024.
> >
> > [7] J. Benton, V. D. Bortoli, A. Doucet, and G. Deligiannidis. Nearly $d$-linear convergence bounds for diffusion models via stochastic localization. The Twelfth International Conference on Learning Representations. 2024.
> >
> > [8] G. Conforti, A. Durmus, and M. G. Silveri. KL Convergence Guarantees for Score diffusion models under minimal data assumptions. arXiv preprint arXiv:2308.12240.

---

> > > ### Comment · Reviewer_n4bv · 2024-11-27
> > >
> > > Thank you for your reply. Some of my concerns have been resolved. I have updated my rating.

---

### Official Review · Reviewer_nM2R · 2024-11-09

**Soundness:** 2
**Presentation:** 2
**Contribution:** 2
**Rating:** 6
**Confidence:** 2

**Summary:**

The paper derives KL convergence bounds for diffusion models when the model is not trained on the target distribution but on a mismatched one. Under some assumptions, the authors exhibit a bound with a term depending on the mismatch. Building on this result, the authors propose an algorithm for conditional sampling that minimizes the mismatch term.

**Strengths:**

This paper participates in the effort to provide theoretical characterizations of guidance and conditional sampling in diffusion models, an area which has not yet been well explored.

**Weaknesses:**

1. Assumption 2 is not realistic: Assumption 2 measures how much the approximated conditional expectation deviates from the true $\mu_t$. The expectation in assumption 2 is taken with respect to $\tilde{Q}$ the target distribution, which is unknown during training. Assumption 2 can thus not be guaranteed to hold. Moreover, the authors state mistakenly, as far as I understand, that Assumption 5 is the same as Assumption 4 when is taken under different distributions. Assumption 5, under $Q_t$, is a classic assumption and, as it relates closely to the training loss, is a realistic assumption. Assumption 4 on the other hand needs more justification.
2. A minor modification of prior work with assumptions to control the additional terms: Theorem 1 is obtained through modification Liang et al 2024 by assuming the target distribution is different from the training one. The simple additional terms that appear due to the modification are controlled using a series of assumptions. This limits the technical novelty over Liang et al 2024.
3. The paper is not easy to read. This can be fixed by reducing the large number of inline equations (paragraphs line 513-523 or paragraph A in the appendix). Unnecessary factors $(1-\alpha_t)$ appear on both sides of an equation in crucial assumptions 3 and 4 which I found to be confusing.

**Questions:**

What does the final line in theorem 2 signify: "Here $W_2(\tilde{Q}_1, \tilde{Q}_0) < \delta d$"?  Is it an assumption or a result of the theorem ? It is unclear from the proof as well.

---

> ### Author Response · Authors · 2024-11-21
>
> We thank the reviewer for providing the highly inspiring feedback. In the revised paper, we have made changes based on your review comments and have highlighted all revisions in blue fonts.
>
> **Q1:** Assumption 2 is not realistic: Assumption 2 measures how much the approximated conditional expectation deviates from the true $\mu_t$. The expectation in assumption 2 is taken with respect to $\tilde{Q}$ the target distribution, which is unknown during training. Assumption 2 can thus not be guaranteed to hold.
>
> **A1:** In learning theory, all performance guarantees are obtained by assuming that the ground-truth scenarios satisfy the claimed assumptions. This does not require that those assumptions can be checked during training, as long as the execution of the algorithm does not need such information. Thus, in almost all literature on theoretical studies of DDPMs, the assumption involves the target distribution, which is unknown during training.
>
> In practice, Assumption 2 is very likely to hold as long as $\tilde{Q}$ is close to $Q$. Also note that in the past, as a first attempt to investigate score-matched DDPMs, [1] assumes that the score function is $L^\infty$-accurate. This assumption is sufficient for our Assumption 2 to hold. In particular, in case of zero-shot conditional sampling, we only consider those $y$’s such that Assumption 2 (and 5) holds.
>
> **Q2:** Moreover, the authors state mistakenly, as far as I understand, that Assumption 5 is the same as Assumption 4 when taken under different distributions. Assumption 5, under $Q_t$, is a classic assumption and, as it relates closely to the training loss, is a realistic assumption. Assumption 4 on the other hand needs more justification.
>
> **A2:** We did not find the suggested claim in our paper that “Assumption 5 is the same as Assumption 4”. Instead, we claim that “Assumption 5 directly implies Assumption 2,” which is proven (straightforwardly) in Equation 9.
>
> Assumption 4 can been verified in two cases of zero-shot conditional sampling: (1) where $Q_0$ has bounded support with a special $\alpha_t$ in Equation (8) (see the proof of Theorem 4); or (2) where $Q_0$ is Gaussian mixture and $H = [I_p, 0]$ (see Lemma 8). Here the first case has wide applicability in practice (e.g., images [2]) and is commonly assumed in many theoretical investigations (e.g., [3]). We have added this piece of clarification in the revised draft.
>
> **Q3:** A minor modification of prior work with assumptions to control the additional terms: Theorem 1 is obtained through modification Liang et al 2024 by assuming the target distribution is different from the training one. The simple additional terms that appear due to the modification are controlled using a series of assumptions. This limits the technical novelty over Liang et al 2024.
>
> **A3:** We respectfully disagree with the reviewer. (i) Theorem 1 identifies the bias terms due to score mismatch via a non-trivial analysis. As we outlined in Section E (prev. Section 5), each of the four steps in the proof of Theorem 1 requires us to establish non-trivial bounds due to mismatched scores. For example, the mismatched scores require us to provide an upper bound for the expected tilting-factor difference in terms of two different conditionals: $P$ and $\tilde{P}$. In comparison, we directly have $P = \tilde{P}$ for score-matched scenarios. Because of this difference, different from the score-matched case where we have centralized moments, here we need to upper-bound all higher-order **non-centralized** moments, for which we employ non-trivial combinatorial techniques (Lemma 4). (ii) Those additional bias terms, which do not exist for the score-matched models analyzed in [4], are not simple, and analyzing them does not come readily from the assumptions. In general, the bias in Theorem 1 may not have a constant upper bound. We show the existence of such a constant upper bound for two noise schedules: the constant noise schedule (Lemma 10) and that in Equation (8) (Lemma 7). Here the latter schedule corresponds to the commonly used exponential-decay-then-constant noise schedule (e.g., [5]). (iii) We also note that the novelty of our paper goes beyond Theorem 1. For example, for the zero-shot conditional sampling problem, we provide a novel design of a zero-shot sampler, as long as explicit parameter dependencies of the sampler (see Section 1.1). Both of these contributions are not captured by Theorem 1 nor by [4].

---

> > ### Author Response · Authors · 2024-11-21
> >
> > **Q4:** The paper is not easy to read. This can be fixed by reducing the large number of inline equations (paragraphs line 513-523 or paragraph A in the appendix). Unnecessary factors
> >  appear on both sides of an equation in crucial assumptions 3 and 4 which I found to be confusing.
> >
> > **A4:** Thank you for your suggestion. We have reduced the number of inline equations in Section E (prev. Section 5). We have also made our Assumptions 3 and 4 more clear by suppressing the factors. In particular, Assumptions 3 and 4 can still be satisfied under two cases: (i) where $Q_0$ has bounded support with a special $\alpha_t$ in Equation (8) (see the proof of Theorem 4); or (ii) where $Q_0$ is Gaussian mixture and $H = [I_p, 0]$ (see Lemma 8). Please check out the revised draft.
> >
> > **Q5:** What does the final line in theorem 2 signify: "Here $W_2(\tilde{Q}_1, \tilde{Q}_0)^2 \lesssim \delta d$"? Is it an assumption or a result of the theorem ? It is unclear from the proof as well.
> >
> > **A5:** This is part of the result in Theorem 2. In words, it means that the output distribution ($\widehat{P}_1$) of the algorithm will be close to some distribution ($\tilde{Q}_1$), which itself is a slight perturbation of the true target ($\tilde{Q}_0$). Such KL-Wasserstein guarantees are common for early-stopped procedures in previous literature (e.g., [5]). We have modified the statement of Theorem 2 in our revised draft to clarify this.
> >
> > We thank the reviewer again for your comments. We hope that our responses resolved your concerns. If so, we wonder if the reviewer could kindly consider raising your score. Certainly, we are more than happy to answer your further questions if any.
> >
> > [1] V. De Bortoli, J. Thornton, J. Heng, and A. Doucet. “Diffusion Schrodinger bridge with applications to score-based generative modeling”. Advances in Neural Information Processing Systems. 2021.
> >
> > [2] J. Ho, A. Jain, and P. Abbeel. Denoising diffusion probabilistic models. Advances in Neural Information Processing Systems. 2020.
> >
> > [3] G. Li, Y. Wei, Y. Chen, and Y. Chi. Towards Non-Asymptotic Convergence for Diffusion-Based Generative Models. The Twelfth International Conference on Learning Representations. 2024.
> >
> > [4] Y. Liang, P. Ju, Y. Liang, and N. Shroff. Broadening Target Distributions for Accelerated Diffusion Models via a Novel Analysis Approach. arXiv preprint arXiv:2402:13901.
> >
> > [5] H. Chen, H. Lee, and J. Lu. Improved analysis of score-based generative modeling: user-friendly bounds under minimal smoothness assumptions. Proceedings of the 40th International Conference on Machine Learning. 2023.

---

> ### Author Response · Authors · 2024-11-27
>
> Dear Reviewer nM2R,
>
> We've taken your initial feedback into careful consideration in our response. Could you please check whether our responses have properly addressed your concerns? If so, could you please kindly consider increasing your initial score accordingly? Certainly, we are more than happy to answer your further questions.
>
> Thank you for your time and effort in reviewing our work!

---

> > ### Comment · Reviewer_nM2R · 2024-11-30
> >
> > Dear authors,
> >
> > Thank you for your clarifications on Assumptions 2, 4 and 5. The following sentence was important for my understanding: **Assumption 2 is very likely to hold as long as $\tilde{Q}$ is close to $Q$.** This claim should be made more clearly in the paper, i.e that the score mismatches are "moderate". I have updated my score.

---

> > > ### Author Response · Authors · 2024-12-01
> > >
> > > Thank you for your feedback! Since the review deadline of PDF revision has passed, we will definitely add this line of clarification under Assumption 2 in our final draft.

---

### Official Review · Reviewer_w6Vp · 2024-11-09

**Soundness:** 3
**Presentation:** 2
**Contribution:** 2
**Rating:** 6
**Confidence:** 3

**Summary:**

This paper studies the convergence guarantees of zero-shot conditional diffusion models under certain regularity assumptions on the target distributions and assuming small score estimation errors. The authors provide distribution estimation error bounds in KL divergence for score-mismatched DDPM and establish convergence rates under several settings. They also introduce a new zero-shot conditional sampler designed to accelerate the convergence rates by reducing the asymptotic distributional bias in the obtained bounds.

**Strengths:**

S1. The paper is well-organized and easy to follow.

S2. The convergence results for zero-shot conditional diffusion models are new and address an area that remains largely unexplored in the literature.

S3. The authors propose a new zero-shot conditional sampler that accelerates the convergence rates of the zero-shot conditional DDPM sampler by reducing the asymptotic distributional bias in the obtained bounds.

**Weaknesses:**

My main concern revolves around Assumption 3, which appears to be relatively strong and may limit the applicability of the results. Specifically:

C1. **Smoothness Requirements:** The assumption requires the existence and control of higher-order derivatives of the log densities. Distributions that are not sufficiently smooth (e.g., those with discontinuities or singularities) may not satisfy this assumption.

C2. **Scaling with $(1−\alpha_t)^m$:** When $\alpha_t$ approaches 1, the factor $(1-\alpha_t)^m$  becomes very small. If the expectations do not decrease at the same rate, the assumption may not hold. This can occur in practice when $\alpha_t$ is close to 1, as in the noise schedule Equation (8) in the paper.

C3. **High Dimensionality Issues:** In high-dimensional settings (large $d$), the behavior of the derivatives can be more complex. The curse of dimensionality may make the assumption unrealistic because the number of derivatives grows exponentially with $d$.

C4. **Applicability to other distributions**: The assumption may not be valid for distributions outside certain classes, such as those with multimodal densities where modes are not well-separated, or distributions with skewness and kurtosis that affect the derivatives.

Additional concerns include:

C5. **Avoiding overstatements of contributions:** In Section 1.1, the paper claims to provide convergence bounds for general target distributions with finite second moments (e.g., Line 92-93). However, their analysis requires regular high-order derivatives for the score functions, which significantly restricts their results to smooth distributions. Adjusting the claim accordingly would provide a more accurate representation of the scope of the contributions.

C6. **Insufficient Discussion on the obtained bounds:** The paper lacks sufficient discussions on the practical implications of the bounds. For example, how do the obtained bounds compare to conventional conditional sampling methods, such as Langevin dynamics? How can the bounds explain the success of conditional diffusion models over other sampling methods?

C7. **Clarification on early stopping in Theorem 2:** The bound in Theorem 2 is given in terms of the KL divergence between the target and learned distribution at time step 1 rather than at time step 0, implying that early stopping has been imposed. Clarification on this point would be helpful.

**Questions:**

Q1. How does Assumption 3 relate to the standard smoothness and regularity conditions typically required for Langevin dynamics sampling, such as Lipschitz continuity and bounds on the gradient and Hessian of the log-density?

Q2. Considering high-dimensional data, how does Assumption 3 ensure that the expectations involving higher-order derivatives remain well-controlled?

Q3. Are there any connections between Assumption 3 and functional inequalities like log-Sobolev or Poincaré inequalities?

---

> ### Author Response · Authors · 2024-11-21
>
> We thank the reviewer for providing the highly inspiring feedback. In the revised paper, we have made changes based on your review comments and have highlighted all revisions in blue fonts.
>
> **Q1:** Smoothness Requirements: The assumption requires the existence and control of higher-order derivatives of the log densities. Distributions that are not sufficiently smooth (e.g., those with discontinuities or singularities) may not satisfy this assumption.
>
> **A1:** Note that Assumption 3 is not restrictive. When $\tilde{Q}_0$ is non-smooth, we can show that Assumption 3 holds as long as $\tilde{Q}_0$ has finite variance (not only bounded support, see Lines 1219-1229), for a particular choice of $\alpha_t$ in Equation (8). Note that this $\alpha_t$ corresponds to the early-stopped procedures in the literature, which is a common practice for non-smooth target distributions [1-2].
>
> **Q2:** Scaling with $(1-\alpha_t)^m$: When $\alpha_t$ approaches 1, the factor $(1-\alpha_t)^m$ becomes very small. If the expectations do not decrease at the same rate, the assumption may not hold. This can occur in practice when $\alpha_t$ is close to 1, as in the noise schedule Equation (8) in the paper.
>
> **A2:** Note that this scaling factor appears on both sides of the equations in Assumption 3. Thus, a sufficient condition for Assumption 3 is that the two expected partial derivatives are *bounded* in $L^\ell$ space. Indeed, with the noise schedule in (8), we can verify Assumption 3 for any $\tilde{Q}_0$ with finite variance.
>
> **Q3:** High Dimensionality Issues: In high-dimensional settings (large $d$), the behavior of the derivatives can be more complex. The curse of dimensionality may make the assumption unrealistic because the number of derivatives grows exponentially with $d$.
>
> **A3:** In [3, Lemmas 15 and 17], an upper bound is shown for all expected values of higher-order derivatives with only *polynomial* dependency on $d$ (rather than “exponential” as suggested). Such polynomial dependency is due to the nice Gaussianity of the perturbation in the forward process.
>
> **Q4:** Applicability to other distributions: The assumption may not be valid for distributions outside certain classes, such as those with multimodal densities where modes are not well-separated, or distributions with skewness and kurtosis that affect the derivatives.
>
> **A4:** Assumption 3 can be satisfied for any $\tilde{Q}_0$ has finite variance with the $\alpha_t$ in Equation (8) (see Lines 1228-1238). Here even though $\log q_0$ might not be smooth, all Gaussian perturbed $\log q_t$’s are still infinitely smooth. Such $\tilde{Q}_0$ constitutes a large class of distributions, including multimodal densities and those with non-smooth derivatives.
>
> **Q5:** Avoiding overstatements of contributions: In Section 1.1, the paper claims to provide convergence bounds for general target distributions with finite second moments (e.g., Line 92-93). However, their analysis requires regular high-order derivatives for the score functions, which significantly restricts their results to smooth distributions. Adjusting the claim accordingly would provide a more accurate representation of the scope of the contributions.
>
> **A5:** See our response in A4. While Assumption 3 is necessary for the proof of Theorem 1, this assumption can be satisfied for a fairly large class of distributions.
>
> **Q6:** Insufficient Discussion on the obtained bounds: The paper lacks sufficient discussions on the practical implications of the bounds. For example, how do the obtained bounds compare to conventional conditional sampling methods, such as Langevin dynamics? How can the bounds explain the success of conditional diffusion models over other sampling methods?
>
> **A6:** First, to the best of our knowledge, there are no theoretical bounds for conditional samplers using Langevin dynamics to compare with, in particular for the inverse problem. Also, to be clear, the goal of this paper is not to compare diffusion-model based conditional samplers with other conditional samplers, say those using Langevin dynamics. The goal, instead, is to investigate the performance of *zero-shot* conditional diffusion samplers when there is score-mismatch, and to compare zero-shot samplers with those *diffusion* samplers knowing full conditional scores. With our result in Theorem 1, we show that, different from these samplers knowing full scores, zero-shot samplers introduce an asymptotic bias which is a function of the score-mismatch. Then, we further minimize the bias with the proposed BO-DDNM sampler.

---

> > ### Author Response · Authors · 2024-11-21
> >
> > **Q7:** Clarification on early stopping in Theorem 2: The bound in Theorem 2 is given in terms of the KL divergence between the target and learned distribution at time step 1 rather than at time step 0, implying that early stopping has been imposed. Clarification on this point would be helpful.
> >
> > **A7:** Thank you for this suggestion. We have made clarifications after Equation (8).
> >
> > **Q8:** How does Assumption 3 relate to the standard smoothness and regularity conditions typically required for Langevin dynamics sampling, such as Lipschitz continuity and bounds on the gradient and Hessian of the log-density?
> >
> > **A8:** Assumption 3 addresses a slightly different class of distributions from traditional assumptions of Lipschitz continuity. To be specific, Assumption 3 can be shown for Gaussian mixtures [3, Lemmas 13-14], which in general do not have a smoothness constant for its derivatives of log-pdf. On the other hand, this assumption becomes unnecessary for the same tilting-factor analysis when there is a uniform smoothness constant (see the proof of [3, Theorem 4]).
> >
> > **Q9:** Considering high-dimensional data, how does Assumption 3 ensure that the expectations involving higher-order derivatives remain well-controlled?
> >
> > **A9:** Please see our answer A3 to Q3. In particular, with the $\alpha_t$ in Equation (8), Assumption 3 can be shown for any $\tilde{Q}_0$ with finite variance (see Lines 1219-1229). Also, the dependency on $d$ is not exponential but only polynomial.
> >
> > **Q10:** Are there any connections between Assumption 3 and functional inequalities like log-Sobolev or Poincaré inequalities?
> >
> > **A10:** We could not think of an explicit relationship between the two. We are happy to hear any insight that the reviewer has. In the convergence analysis, these inequalities are useful to establish convergence rates when the target is Lipschitz-smooth. However, as we explained in A8, Assumption 3 is used to address a different class of target distributions.
> >
> > We deeply thank the reviewer once again for your highly inspiring and thoughtful comments. We sincerely hope our responses have fully addressed your concerns and clarified any uncertainties. If so, we would be truly grateful if you could kindly consider raising your score. We remain more than happy to address any further questions or provide additional clarifications.
> >
> > [1] Joe Benton, Valentin De Bortoli, Arnaud Doucet, and George Deligiannidis. Nearly $d$-linear convergence bounds for diffusion models via stochastic localization. The Twelfth International Conference on Learning Representations. 2024.
> >
> > [2] H. Chen, H. Lee, and J. Lu. Improved analysis of score-based generative modeling: user-friendly bounds under minimal smoothness assumptions. Proceedings of the 40th International Conference on Machine Learning. 2023.
> >
> > [3] Y. Liang, P. Ju, Y. Liang, and N. Shroff. Broadening Target Distributions for Accelerated Diffusion Models via a Novel Analysis Approach. arXiv preprint arXiv:2402:13901.

---

> ### Author Response · Authors · 2024-11-27
>
> Dear Reviewer w6Vp,
>
> We've taken your initial feedback into careful consideration in our response. Could you please check whether our responses have properly addressed your concerns? If so, could you please kindly consider increasing your initial score accordingly? Certainly, we are more than happy to answer your further questions.
>
> Thank you for your time and effort in reviewing our work!

---

### Meta-Review · Area_Chair_oyN5 · 2024-12-11

**Metareview:**

The reviews are all generally on the positive side, though usually not enthusiastically so.  The contributions are generally deemed to be valuable, giving convergence analyses for  zero-shot conditional diffusion models under suitable regularity conditions, and introducing a zero-shot sampler to accelerate convergence.  The opinions on the paper writing are mixed (some positive and some negative).  Maybe most significantly, several of the assumptions were raised into question or clarifications sought, but the rebuttal seems to have addressed these sufficiently.

**Additional Comments On Reviewer Discussion:**

As noted above, several of the assumptions were raised into question or clarifications sought, but the rebuttal seems to have addressed these sufficiently.

---

### Decision · Program_Chairs · 2025-01-22

Accept (Poster)